**Multiple-point statistical simulation for hydrogeological models: 3D training image development and conditioning strategies**

Anne-Sophie Høyer[1], Giulio Vignoli[1,2], Thomas Mejer Hansen[3], Le Thanh Vu[4], Donald A. Keefer[5], Flemming Jørgensen[1]

[1]Groundwater and Quaternary Geology Mapping Department, GEUS, Aarhus, 8000, Denmark
[2]Department of Civil, Environmental Engineering and Architecture (DICAAR), University of Cagliari, Cagliari, 09123, Italy
[3]Niels-Bohr Institute, University of Copenhagen, Copenhagen, 2100, Denmark
[4]I-GIS, Risskov, 8240, Denmark
[5]Illinois State Geological Survey, Champaign, IL 61820, USA

*Correspondence to*: Anne-Sophie Høyer (ahc@geus.dk) and Giulio Vignoli (gvignoli@unica.it)

**Abstract.** Most studies about the application of geostatistical simulations based on multiple-point statistics (MPS) to hydrogeological modelling focus on relatively fine-scale models and concentrate on the estimation of facies-level structural uncertainty. Much less attention is paid to the use of input data and optimal construction of training images. For instance, even though the training image should capture a set of spatial geological characteristics to guide the simulations, the majority of the research still relies on 2D or quasi-3D training images. In the present study, we demonstrate a novel strategy for 3D MPS modelling characterized by: (i) realistic 3D training images, and (ii) an effective workflow for incorporating a diverse group of geological and geophysical data sets. The study covers an area of $2\,810$ km$^2$ in the southern part of Denmark. MPS simulations are performed on a subset of the geological succession (the lower to middle Miocene sediments) which is characterized by relatively uniform structures and dominated by sand and clay. The simulated domain is large and each of the geostatistical realizations contains approximately 45 million voxels with size 100 m x 100 m x 5 m. Data used for the modelling include water well logs, high-resolution seismic data, and a previously published 3D geological model. We apply a series of different strategies for the simulations based on data quality, and develop a novel method to effectively create observed spatial trends. The training image is constructed as a relatively small 3D voxel model covering an area of 90 km$^2$. We use an iterative training image development strategy and find that even slight modifications in the training image create significant changes in simulations. Thus, this research shows how to include both the geological environment, and the type and quality of input information in order to achieve optimal results from MPS modelling. We present a practical workflow to build the training image and effectively handle different types of input information to perform large-scale geostatistical modelling.

# 1 Introduction

Simulation of groundwater flow and solute transport requires representative hydrogeological models of the subsurface. While many studies focus on estimating the spatial distribution of hydraulic properties (i.e. hydraulic conductivity, porosity),

the reliable delineation of the underlying geological structural or hydrostatrigraphic model is far more important to ensure the reliability of groundwater flow predictions (e.g. Carrera, 1993; Liu et al., 2004a; Renard and Allard, 2013).

Early geostatistical methods (e.g.: Chiles and Delfiner, 1999; Deutsch, 2002) provided tools to model uncertainty in smoothly varying Gaussian properties, like porosity or permeability within targeted rock formations (e.g., oil reservoirs).

These produced petrophysical models that generally improved mineral resource development, but the models were poor when abrupt changes in rock type, and hence petrophysical properties were important in defining reservoir structure. Later generations of geostatistical methods supplied tools for modelling rock types (i.e., lithology or facies), that provided incremental improvement (Stafleu et al., 2011), but still did not effectively model non-stationary patterns in rock type, and poorly captured patterns in stratigraphical units that are typically complex combinations of rock type.

Without suitable tools for developing realistic uncertainty-informed models of either non-stationary rock type or hydrostratigraphy, groundwater models are typically based on single, deterministic geological framework models that are manually constructed. Research has continued to explore methods for uncertainty-based simulation of structural heterogeneities (Huysmans and Dassargues, 2009; Kessler et al., 2013) to better estimate uncertainties in groundwater flow (Feyen and Caers, 2006; He et al., 2013; Poeter and Anderson, 2005; Refsgaard et al., 2012).

The multiple-point statistics (MPS, Guardiano and Srivastava, 1993; Strebelle, 2002) was developed with the objective to better reproduce complex geological patterns than other available geostatistical modelling techniques. The method combines the ability to condition the realizations to hard data (information without uncertainty) and soft data (information with uncertainty) with the ability to reproduce geological features characterized by statistical properties described through a so-called Training Image (TI).

Hence, the TI plays a crucial role and should be constructed to represent the structural patterns of interest (Hu and Chugunova, 2008; Maharaja, 2008; Pickel et al., 2015). However, a fundamental challenge for the use of MPS for 3D hydrogeological modelling lies in the difficulty to produce realistic 3D TIs (e.g., Pérez et al., 2014, Ronayne et al, 2008), and most of the studies in the literature are therefore based on 2D or quasi-3D TIs (Jha et al., 2014; Comunian et al., 2012; Feyen and Caers, 2006). Several researches have demonstrated the potential of the MPS to simulate 2D patterns (Hu and

Chugunova, 2008; Liu et al., 2004b; Strebelle, 2002), but less attention has been given to the construction of real three-dimensional TIs.

In the present study, we describe a strategy to develop effective and realistic 3D TIs based on iterative modifications of an initial training image. The TI's updates are based on the consistency between the general features of the corresponding unconstrained realizations and the actual geological expectations. This approach allows taking into account the coupled

effects of the considered TI and the used implementation of the MPS algorithm. Moreover, the tests performed in this research show that few unconstrained realizations per tested TI are sufficient to obtain a correct assessment of the general features common to all the realizations.

Concerning the use of conditioning data, it is extremely important to address the fact that the data are characterized by different uncertainty levels and scales (He et al., 2014; McCarty and Curtis, 1997). In the example discussed in this study,

available sources of information are seismic lines, boreholes, and a pre-existing, manually constructed geo-model. Depending on their scale and uncertainty, we developed a practical way to incorporate all of them into the Single Normal Equation Simulation (SNESIM) workflow as it is implemented in SGEMS (Remy et al., 2009). For example, the data from boreholes are used in a non-standard way as soft conditioning and only after an appropriate pre-processing aiming at: (i)

removing the effects of scale mismatches, (ii) properly accounting for the data uncertainty, and (iii) effectively migrating the information between the borehole locations. The results of this novel approach are compared against more traditional strategies along the article. The proposed workflow is general and can be readily extended to other data types (for example, other kinds of geophysical data).

In summary, the main objectives of the present study concern: (i) the iterative development of effective 3D TIs, and (ii) the

optimal strategy for the simulation conditioning.

The paper is organized as follows: (i) Section "2 The methodological framework" summarizes the strategies for the optimal TI construction and best conditioning utilizing the available data and prior knowledge. These strategies are actually discussed along the entire paper and exemplified through their application to a real case. (ii) Section "3 Study area" deals with the overall geological framework of the area considered to test the novel approach. In particular, its subsection "3.1

Establishing framework-model constraints" goes into details about the specific geological unit targeted during the stochastic simulations. (iii) After the description of the test area, section "4 Data" describes the amount and characteristics of the different kinds of available data. (iv) Section "5 Defining MPS input information" consists of the presentation of the application of different approaches whose outputs are then compared in the subsequent section "6 Results". (v) The second last section, "7 Discussion", is about the assumptions/choices made across the paper and their possible limitations and future

developments; (v) "8 Conclusion" is a concise section where we recap our results.

Moreover, in "Appendix A", for sake of completeness, we provide further details regarding the variability within each probabilistic model and analyse additional realizations with respect to those necessary for the implementation of the proposed workflow.

## 2 The methodological framework

This research is about the development and testing of a methodological workflow for 3D MPS modelling aiming at: (i) the construction of truly effective (3D) training images that considers also the coupled effects of the specific MPS algorithm actually used, (ii) the inclusion of different geological/geophysical data sets that properly takes into account their varying uncertainty, (iii) the possibility of spreading the influence of the soft conditioning beyond the neighbourhood of the available observations.

## 2.1 Optimal training image development

In order to apply any MPS strategy, it is necessary to build a TI (possibly 3D). The construction of the TI is based on the prior geological knowledge concerning the geological unit to be simulated. However, the actual realizations produced by using a particular TI are also influenced by the specific implementation, and simulation settings, used for the stochastic simulation. Hence, it is necessary to consider the coupled effects of the TI together with the algorithm implementation as a whole. The proposed approach to practically tackle this is based on an analysis of the unconditioned realizations. Thus, in the present framework (Fig. 1a), it is not merely the TI that needs to be representative of the investigated geology, but it is the unconditioned simulation result that needs to match the geological expectations. So, our strategy for the development of a truly effective training image consists in iteratively adjusting the initial TI until the associated unconditioned realization shows the behaviour we desire. In the following section "5.4 Training image", we demonstrate that even tiny modifications have dramatic impacts on the features in the unconditioned simulations. Clearly, during the trial-and-error adjustment of the TI, only features in the unconstrained realizations that are general and not realization-dependent can be considered. For example, in the present article, a single realization per proposed TI is visually analysed and the corresponding TI's updates are based on the evolution in size, elongation, and compactness of the resulting sand bodies. In general, the differences between the subsequent unconditioned realizations are very clear to the geomodeller developing the optimal TI (on this respect, see the section devoted to the TI construction: "5.4 Training image"). However, the same differences can be easily made more quantitatively evident (as it is exemplified in the practical test discussed along the present paper).

## 2.2 Accounting for variable data uncertainty and scale mismatch

Too often borehole data are treated as ground-truth and naïvely considered with no uncertainty. However, as any other observation, they are affected by some sort of noise and are prone to errors and misinterpretations. Moreover, borehole data can be at a resolution that is not compatible with the scale of the geostatistical simulation. On the other hand, observations that are usually considered "indirect" can be extremely reliable and precise, and at the correct scale when compared with the features to be simulated. In the rest of the paper, for example, seismic data and pre-existing adjoining geological models are considered very reliable sources of information at the right scale. Our approach invites to reconsider the conditioning data both in terms of uncertainty and scale mismatch (Fig. 1b). And it offers a way to effectively take into account both these aspects: data can be efficiently translated into soft probability weighted accordingly to the varying reliability. For example, in the test discussed along the paper (see, Table 1, strategies (c) and (d)), borehole information is turned into soft probability weighted by the distance from the centre of each of the formations (this is the essence of what the moving window does; see sections "5.3 Borehole data" and "7 Discussion"). In addition, to take into account the generally poor drilling quality, a flat 20% weight has been associated to every transformed borehole. In an even more sophisticated implementation of our approach, as it is further described in the section "7 Discussion", the weighting can, in principle, be varying between boreholes and changing based on their available quality rate. Similar weighting strategy can be applied to any kind of data

(including the geophysical measurements). However, in the example in this research, the seismic observations (and the pre-existing Tønder model) are considered highly reliable (and with a characteristic scale compatible with the stochastic simulation); for this reason, a constant weight equal 1 has been used and no averaging has been applied. This makes them suitable for hard conditioning.

**2.3 Enforcing the data-driven spatial trend in the soft conditioning**

Very often, especially if the size of the simulation domain is significant (as in the example discussed later in the present article), some characteristics (e.g.: the sand/clay ratio) are spatially varying at large scale and it would be highly desirable to include this spatial trend in the simulation process. Unfortunately soft data have a limited spatial influence as it is shown in the sections "5.3 Borehole data" and "7 Discussion" and further demonstrated in Hansen at al. (in review). With the same rationale of the previous subsection "2.2 Accounting for data uncertainty and scale mismatch", within the framework of our approach (Fig. 1b), the information is migrated far from the source of soft probability by means of a weighted conditioning. In the specific example described in the rest of the paper, the weight is linked to the distance from the boreholes. Specifically, by kriging the soft probability derived from the borehole observations, we are able to enforce the trend clearly visible in the data. In general, the applicability of this approach is not limited to borehole information, but can be extended to all the soft data with a limited spatial support whose influence needs to be spread across the simulation domain.

**3 Study area**

The study area covers $2\,810$ km$^2$, from coast to coast in the southern part of Jutland, Denmark, and northern portion of Germany (Fig. 2). A part of this area ($625$ km$^2$) has formerly undergone 3D geological modelling (the Tønder model, Fig. 2) which is presented in Jørgensen et al. (2015). In this paper, we will concentrate on the Miocene sediments within the model domain. The Miocene sediments were selected because: (i) they are primarily composed of two main hydrogeological facies: clay and sand, and (ii) the unit is composed of rather uniform structures throughout the area and therefore possible to describe in a 3D TI.

A conceptual sketch of the geology in the area is shown in Fig. 3. The base of the regional groundwater flow system corresponds with the top of the very fine-grained Paleogene clays that have a gentle dip towards the southwest. The overlying Miocene sediments consist of marine clay with sandy deltaic lobate layers. The Miocene deposits in Jutland has been thoroughly studied by Rasmussen et al. (2010) who based the studies on sedimentological and palynological investigations of outcrops and cores, combined with interpretation of high-resolution seismic data. During the Miocene, the coastline fluctuated generally northeast-southwest across Jutland, resulting in a deltaic depositional environment. Rasmussen et al. (2010) divided the Miocene deposits into three sand-rich deltaic units that are interfingered with prodeltaic clayey units. In periods with low water levels, the coastline was situated far to the southwest and coarse-grained sediments were consequently deposited in the study area. When the water level was high, the coastline was situated further to the northeast,

resulting in deposition of marine clays. The delta lobes dominate in the northeast and show smooth dips towards southwest. In middle and upper Miocene, very fine-grained marine clays belonging to the Måde Group were deposited in the western part of the model domain.

The unconformable boundary between the Quaternary and the Miocene is often scoured by buried valleys (Fig. 3). The
Quaternary deposits mainly consist of till and various meltwater deposits, but, in some places, also of interglacial and postglacial deposits. In some areas, the Quaternary deposits are heavily deformed due to glaciotectonism. In the western portion of the study area, the geology generally consists of outwash sandurs surrounding old pre-Weichselian moraine landscapes. The outwash sandurs show a low relief with a gentle dip towards west, whereas the old moraine landscape shows a more irregular relief up to 60 meters above sea-level (m.a.s.l). In the east, the younger morainic landscape from the
Weichselian shows topographic changes between 40 - 90 m.a.s.l. The model area is influenced by a large fault-bounded structure (the Tønder graben structure, Fig. 2b) that offsets the deeper Top Chalk surface of about hundred meters (Ter-Borch, 1991). The faults clearly offset both the top of the Paleogene and the bottom of the Quaternary.

### 3.1 Establishing framework-model constraints

The MPS model domain discussed in the present research corresponds to the lower and middle Miocene sediments which are
positioned below the Måde Group and above the Top Paleogene surface (Fig. 4). Inside the Tønder model area (Fig. 2), however, the results of the already existing model (Jørgensen et al., 2015) are used and therefore are not included in the MPS simulations. The Tønder model results are used because the model was constructed based on very thorough geological analyses and so considered as the best geological model obtainable in that area. The surfaces of the relevant geological boundaries: Top Paleogene, Bottom Måde and Top pre-Quaternary (Fig. 4) are constructed in the geo-modelling software
package Geoscene3D (I-GIS, 2014) by using interpretation points and interpolating these data to create the corresponding stratigraphical surfaces. The top of the MPS model domain is obtained by merging the surfaces of the Bottom Måde and the Top pre-Quaternary surface (Figs. 4 and 5a). The lower boundary of the model domain corresponds to the Top Paleogene surface (Figs. 4 and 5b).

The top surface of the Miocene sediments shows a regional dip towards west with significant depressions along the Tønder
graben structure and at the buried valleys. The surface elevation ranges from 20 m.a.s.l. to -460 m.a.s.l. (Fig. 5a), corresponding to depths varying between 30 m and 80 m, in the east, and of about 160 m, in the west. In the Tønder graben, the depth to the top of the Miocene ranges approximately from 260 m to 320 m and locally (associated with buried valleys) up to 460 m. The bottom surface of the Miocene sediments (Fig. 5b) varies more smoothly, and also has a small regional dip from east (near elevation -100 m.a.s.l.) to west (approximately elevation -320 m.a.s.l.), punctuated primarily by the Tønder
graben structure (elevation down to -560 m.a.s.l.). The Miocene sediments typically have a thickness varying between 50 m and 250 m, but are thinner and locally absent below the buried valleys (Fig. 3).

## 4 Data

### 4.1 Borehole data

The borehole data in the Danish portion of the study area are extracted from the Danish national borehole database, the 'Jupiter' database (http://jupiter.geus.dk), and comprise about 9 000 boreholes with lithological information from driller's logs and sample set descriptions. The Jupiter database provides information on a number of parameters about each borehole, which enables an evaluation of borehole data quality. Borehole data, like any other kinds of observations, are affected by uncertainty. The level of uncertainty determines the quality/reliability of the measurements (i.e., the quality/reliability of the boreholes). Many factors impact the quality of boreholes. Just to mention a few of them: (i) the drilling methods (e.g.: rotary drilling and air lift drillings. In some cases, for example, the finer sediments can be flushed out and the driller could potentially misinterpret a clay layer as more sandy); (ii) the drilling purpose (sometimes, if the goal is to reach a specific target, the lithological description can be poor since it is not a priority); (iii) the age of the boreholes (for example, nowadays, in Denmark, samples are collected systematically every meter, and this was not the case few years ago); (iv) the presence of simultaneous wireline logging data (these kinds of ancillary information make the geological interpretation definitely more certain). During the geological modelling phase, a skilled geologist should go through all the borehole records and verify all these different pieces of information and check for inconsistencies. This is (or should be) the standard procedure to assess the quality of the boreholes and prepare the data for the subsequent geo-modelling phases. In the study area, many borehole records are of low quality.

Together with the Danish boreholes, geological information from about 500 boreholes located within the German portion of the study area and deeper than 10 m is included in the analysis. The majority of the boreholes in both the Danish and the German area are quite shallow and, in total, only 2 % of all the boreholes are deeper than 100 m (Fig. 2).

### 4.2 High-resolution seismic data

The seismic data are extracted from the Danish geophysical database 'Gerda' (Møller et al., 2009). Most of the seismic lines were conducted with the purpose to investigate the groundwater resources and examine the Miocene sediments (Rasmussen et al., 2007). In the study area, seismic lines for a total length of around 170 km (Fig. 2) were collected by several contractors with two slightly different acquisition systems that, for sake of clarity, we conventionally indicate here with SYS_COWI and SYS_RAMBØLL. In both cases, the lines were acquired as landstreamer high-resolution seismic data (Vangkilde-Pedersen et al., 2006) by using seismic vibrators as energy source. The frequency ranges spanned: from 50 Hz to 350 Hz, for SYS_COWI, and from 50 Hz to 400 Hz, for SYS_RAMBØLL; for both systems, the sweep was 5 s long. The receiver arrays consisted of: 95 not-equally spaced geophones (1.25 m between the first 32 geophones, and 2.5 m for the others), for SYS_COWI, and 102 geophones (1.25 m between the first 50 geophones, and 2.5 m for the others), for SYS_RAMBØLL. In both acquisition settings, the sources were fired every 10 m.

Under normal circumstances, the data are high quality between approximately 30 m to 800 m in depth. Prior to the import of the seismic data to the geological interpretation software, the elevation values are adjusted, based on an assumed constant seismic velocity of 1 800 m/s (Kristensen et al., 2015), which is a common velocity for Miocene and Quaternary deposits in Denmark (Høyer et al., 2011; Jørgensen et al., 2003). Elevation corrections are important because of the considerable effect

of the topography that, even if mildly varying, along extended profile can significantly affect the quality of the final results.

## 5 Defining MPS input information

### 5.1 Seismic data

Rasmussen et al. (2010) proposed a lithostratigraphy for the Miocene succession, a "Miocene model" (Kristensen et al., 2015), based on interpretations of seismic data that was correlated with borehole interpretations and outcrops. The

10 observational data points used in the Miocene model were utilized in our study to define the top of individual Miocene stratigraphical formations along the seismic lines (Fig. 6a). In order to use this information in the MPS modelling, the stratigraphical interpretations from the Miocene model are translated into a binary sand/clay voxel model of one cell width along each seismic line (Fig. 6b and 7).

The sand/clay distribution along the seismic lines is used as hard data in the MPS simulations. The information is used as

hard conditioning since it is considered highly reliable, and since the scale of structures delineated by the seismics is comparable to the scale of the simulated output.

### 5.2 Existing 3D model

In order to ensure edge matching between the stochastic realizations and the deterministically constructed Tønder model, a buffer zone along the pre-existing model's edges is created within GeoScene3D and used as hard conditioning data for the

20 MPS simulations. The Miocene formations interpreted in the existing Tønder model (Jørgensen et al., 2015) are translated into sand and clay. In Fig. 7, the hard conditioning data from the pre-existing model (the buffer zone) are shown together with the hard information from the seismic lines.

### 5.3 Borehole data

All lithological categories contained in Jupiter are simplified and divided into three groups: sand, clay and 'other'. In this

study, the conditioning based on the boreholes has been conducted through the use of three different strategies. The different ways of considering the borehole information are illustrated and summarized in Fig. 8.

The simplest and most common approach for exploiting the information from the boreholes is to include them as hard conditioning data. This is the first test performed in the present study (Fig. 8a). However, this simple approach does not take into consideration the uncertainty in borehole data associated with: (i) inaccuracies in the recorded information, and (ii) the

30 resolution, or scale, differences between the information in the borehole records and the geological model cell size.

In order to address these issues, an alternative conditioning strategy is tested. In this case, the borehole data are considered as soft (Fig. 8b): a 20-meter-long moving window is applied to each original borehole in order to average the densely sampled lithological data into probabilities defined on the coarser simulation grid. Since the typical vertical dimensions of the structures are ~20 m, the size of the averaging window is chosen accordingly. This procedure is consistent with the reasonable assumption that, in the middle of the geological formation, we are relatively certain of the lithology, while this certainty decreases as we get closer to the boundaries. Moreover, in order to take the general uncertainty in the borehole information into account, we map the resulting averaged values into a sand probability interval ranging from 80 to 20 %. In this way, voxels with the highest chances of having sand are characterized by a maximum sand probability value of 80 % (in pink, Fig. 8b), whereas voxels with the lowest sand probability are associated to a value as low as 20 % (in green, Fig. 8b). In Fig. 8b, it is clear that the transition zones between sand and clay correspond to a band with sand probabilities of 40 %. This value is the marginal distribution value for sand occurrence within the investigated model volume, as it is calculated from the borehole data and consistently formalized in the TIs.

When borehole information is too distant, the sand/clay ratios in the stochastic realization are solely derived from the TI and the sand marginal distribution. Unfortunately, borehole information at the relevant depths is relatively sparse (due to shallow boreholes), and, at the same time, the SNESIM algorithm has a tendency to ignore such "localized" soft data (Hansen et al., in review). Moreover, it is known that the model domain shows a slight spatial lithological variation in which the overall proportion of sand is higher in the northeast compared to the southwest (that is also clear from the borehole data). To enforce this real, general trend in the realisations, the "localized" borehole probabilities are kriged into a 3D grid to be used as "diffuse" soft conditioning (Fig. 8c and Fig. 9). This would be unnecessary if SNESIM could handle soft data as local conditional probability and, here, we are suggesting a practical strategy to effectively overcome this limitation. Clearly, the kriged sand probability distribution is seen to correspond to the boreholes close to their locations (Fig. 8c), whereas areas far from boreholes (e.g., at km 6 along the section in Fig. 8c) appear white (so characterized by a sand probability equal to 40 %). However, in a few cases (e.g. the clay layer in the deep part of the borehole at profile distance 10 km, Fig. 8c), the borehole information does not seem to migrate into the surrounding grid. This can be explained by the influence of other boreholes in the 3D domain. The desired spatial trend, characterized by a higher sand probability in the north-east compared to the south-west, is evident in the sand probability grid (Fig. 9).

### 5.4 Training image

The training images (TIs) are constructed as 3D voxel models with the same discretisation as the entire model (100 m by 100 m laterally, and 5 m vertically) and consist in approximately 500 000 voxels, covering an area of 90 km$^2$. The sizes of the TIs are significantly smaller than the simulation domain, but they are large enough to cover the size of the typical structures in the simulated Miocene unit. The TIs are used by the MPS simulation algorithm to represent the basic spatial and proportion relationships of the sand and clay facies within the Miocene unit. Hence, the TIs are modelled to show deltaic sand layers

building out towards southwest within a larger clayey unit. The sand content in the TIs is 40 %, in accordance with the proportion of sand observed in the borehole data.

The TIs are built in Geoscene3D using the voxel modelling tools described in Jørgensen et al. (2013). In practice, each sand and clay layer is defined by so called "interpretation" points. These points are subsequently interpolated into surface grids, defining the volumes, which are then populated with sand and clay voxels. Thus, by manually changing the locations of these interpretation points and/or creating/deleting some of them, we can have a full control over the adjustments of the TIs.

In the present study, several different TIs were tested, and, for sake of simplicity, only two of them (the first attempt and the final TI) are explicitly showed in this paper (Fig. 10). The first TI (TI1 - Fig. 10a) is based on the existing 3D geological model covering an adjacent area (the Tønder model; Jørgensen et al., 2015). The first TI has been manually adjusted, during several iterations, based on the unconditional outputs. This iterative process stopped when the corresponding unconstrained realization was found satisfactory in terms of its ability to mimic the geological features we expect in the Miocene across the study area. Those expectations about the geology are based on our prior geological understanding of the area, the available seismic lines, and the few existing deep boreholes. For example, the unconditioned realizations, associated with the TIs in Fig. 10, are shown in Fig. 11. The main difference is that TI1 has more layers than the second one (TI2 - Fig. 10b). This is clearly reflected in the unconditioned simulations, in which, the realization based on TI1 (Fig. 11a) shows significantly more layers than the corresponding realization based on TI2 (Fig. 11b). The results are evaluated and compared against the structures expected from the Miocene model (Kristensen et al., 2015). Hence, to adhere to what we know about the Miocene geology, the unconstrained realization associated with the selected TI was supposed to show fewer, larger, and more compact sand structures. These characteristics are evident if we directly compare the two realizations in Fig. 11, and clearly confirmed by the study of the associated distributions in Fig.s 12, 13, 14 (see, for instance, Haralick and Shapiro, 1992). In particular, Fig. 12 quantitatively demonstrates that larger sand bodies are more frequent in the realization corresponding to TI2 (black histogram), while the realization generated by TI1 (green histogram) is characterized by a significantly higher presence of relatively small sand layers.  Regarding the shape of the features of the realizations in Fig. 11, Fig. 13 highlights that elongated sand structures are more probable for the realization in Fig. 11a (in fact, eccentricity equal 1 corresponds to the degenerate case of a straight line). Jaggedness, defined as the ratio between the surface and the size of bodies, can provide a useful estimation of the compactness of the sand bodies (Fig. 14). Not surprisingly, the sand lenses in the realization associated to TI1 are more jagged than those in the other realization obtained by using TI2. Because of the higher accordance between our geological expectation of the Miocene in the area and the unconstrained realization in Fig. 11b, TI2 is selected for all MPS simulations discussed in the rest of the paper. These conclusions are drawn on a single unconstrained realization per TI. Nevertheless, their validity is general and they do not depend on the specific realization considered. In fact, only the features induced by the specific choice of the TI (coupled with the actually used implementation of the MPS algorithm) are taken into account in the TI selection process. An in depth discussion of multiple realizations and their mutual coherence in terms of the proposed analysis is presented in the Appendix A (and in the associated Fig.s 20-22). Naturally, after the full model has been set up conditional to all the data and the selected TI, and during its use for, e.g., risk analysis or

as input for hydrological modelling, a large (as large as possible) collection of realizations of this model would be useful. However, during the construction of the optimal TI to be utilized as input to the geostatistical model (as we do here) a few realizations suffice.

## 6 Results

In the following, we present and compare the structures of the single realizations generated by each of the different conditional strategies analysed in this study (Table 1). All the realizations are produced by using the same random seed to better appreciate the differences. For comparison, an unconditioned realization (a) is presented in the first panel of each of the figures (Figs. 15 - 17). The second panel (b) shows a realization generated by using exclusively hard conditioning. In the third panel (c) a realization is presented, in which the borehole information is directly treated as soft conditioning data.

Finally, in the fourth panel (d), a realization with the sand probability borehole grid used for soft conditioning is shown.

Figure 15 shows a horizontal slice through a realization representing each of the four conditional strategies listed in Table 1. As expected, the overall size and form of the structures are comparable between the realizations in (a) – (d) since the spatial characteristics of the structures are primarily determined by the TI. In (b), (c) and (d), the realizations within the Tønder buffer zone (delimited by the dash lines, Fig. 15) are completely defined by the hard data. It can be observed that the buffer

zone cannot be identified in the realizations, which further supports our final choice for the TI2. In fact, this indicates that the same kind of spatial variability and geological patterns are seamlessly present within the pre-existing Tønder model and the simulation results. In this horizontal slice, it is possible to appreciate the effect of the soft probability grid (Fig. 15d) compared to the case of "localized" borehole information (Fig. 15b and 15c): when using the probability grid, there is a significantly higher sand content and more interconnected sand body occurrence in the middle part of the study area

(compare with Fig. 9).

Figure 15 shows vertical profiles through each of the tested realizations (Table 1) along a transect with several deep boreholes (for location, see Fig. 15). Like in Fig. 15, the overall structural patterns of the individual realizations appear similar. The outcome of the unconditioned realization does not agree with the borehole information, since this has not been used. For instance, this is confirmed by the borehole at profile distance 10 km in Fig. 16a. On the contrary, in the realization

where the borehole information is considered as hard data (Fig. 16b), almost perfect consistency is observed between the realization and the boreholes. However, mismatches naturally occur when the individual layers in the borehole records are thinner than the simulated voxel thicknesses (e.g., by the borehole at profile distance of about 15.5 km). Another seeming mismatch in Fig. 16b occurs every time the influence of the borehole information is extremely local, sometimes limited to a single voxel corresponding to the one actually holding the borehole information. This is, for instance, seen for the borehole

at profile distance of about 9 km, where the fit between the realization and the deeper part of the borehole only appears when observed at a very detailed scale.

Also in the realization where the borehole information is considered as soft data (Fig. 16c), there is a generally good fit between the realization and the borehole data, but this match is clearly less pronounced in areas where a high uncertainty is associated with the borehole data (Fig. 8b). For instance, a poorer fit compared to Fig. 16b is observed at profile distance of about 10 km. Generally, the fit to the boreholes is better when the soft conditioning information does not conflict with other

statistical properties as defined by the TI or neighbouring conditioning information. An example of this conflict can be observed by the borehole at about 9 km. Here, the test with the hard data (Fig. 16b) resulted in a highly local match concentrated in a single voxel column, implying that the borehole information did not comply with neighbouring data or statistical parameters. For the same reason, the realization does not strictly fit the borehole when the borehole information is treated as soft conditioning data (Fig. 16c). The difference between (c), where the borehole data are treated as soft

probability, and (d), where the soft probability 3D grid is used (Fig. 8c), is most pronounced when considering the realization results on a larger scale than the shown in the profile in Fig. 16.

A further example highlighting the effects of the probability grid is shown in Fig. 17, where a long SW-NE profile through each of the tested realizations (Table 1) is inspected (for location, see Fig. 15). Borehole data are located further away than the voxel size (100 m) and is therefore not shown on the profile. Again, the overall structural pattern is comparable between

the different realizations. As also observed in the horizontal view (Fig. 15), and as it must be, the results in (b), (c) and (d) are fixed within the buffer zone around the Tønder model, where the hard conditioning information is used. For exactly the same reason, the realizations in (b), (c) and (d) perfectly match also where the profile crosses the seismic lines. Since the differences in the constraining strategy are related to the way the borehole data are handled, only little structural dissimilarities are seen when considering this specific profile with no borehole data. Those differences are therefore mainly

controlled by the trend imposed by the soft probability grid in (d). Hence, the most pronounced difference is the higher clay content in the south-western part of (d) compared to the northeast. This spatial variation in the clay content from the west to the east is even clearer in the 3D view of the realization shown in Fig. 18. The kriged sand probability is effective in enforcing the proper spatial trend on the realization. This is evident, not only from the comparison between Fig. 18b and Fig. 9, which allows verifying, voxel-by-voxel, the accordance between the soft conditioning distribution and the final

corresponding realization, but also from the results in Fig. 19. In fact, Fig. 19 shows the cross-correlations (see, for instance, Stoica and Moses, 2005) between the soft probability distribution (Fig. 9) and each of the realizations visible, respectively, for example, in Fig. 15c and Fig. 15d (see, also, Table 1, cases (c) and (d)). As expected, the correlation with the realization (d) has a much higher and more pronounced maximum.

It is probably important to stress that our conclusions are general as all the differences highlighted for each conditioning set-

up are not realization-dependent. This means that they would appear consistently for every realization obtained with the same conditioning setting. For clarity, this aspect, together with the variability within each probabilistic model, are further discussed and analysed in Appendix A.

## 7 Discussion

If MPS modelling is applied to large study areas, it is typically necessary to divide the area into different regions or domains and use different TIs to properly describe the geology in each individual region. In the present study, the entire investigated area was divided into four main, vertically subdivided, units: the Quaternary, the Måde Group, the Miocene and the Paleogene. The Miocene sequence was chosen for our study since it is relatively stationary from a statistic point of view and can be easily divided into two main facies: sand and clay.

One of the challenges was the presence of the significant graben structure, which offsets the Miocene layers. In the current study, the graben structure is only visible through the morphological shape of the top and bottom of the model domain, but in the MPS realizations this has been ignored. In future studies, faults should be handled, such that the MPS simulation results are affected across the faults. A possible solution could be to use the geochron formalism (Mallet, 2004), in which the simulation could be performed in a regular grid, with geo-time as y-axis. Realizations should then be converted into depth using a geo-time to depth conversion.

The generation of 3D TIs that produce desired patterns is time consuming and difficult, which may be why many of the previous studies are based on 2D or quasi-3D TIs (Comunian et al., 2012; Cordua et al., 2016; Feyen and Caers, 2006; Strebelle, 2002). In this study, the 3D TI was created in the modelling software GeoScene3D. Since GeoScene3D is specifically designed for 3D geological modelling and hosts tools for manual voxel modelling (Jørgensen et al., 2013), the creation of the TI was relatively easy and straightforward. When creating the TI in this manner, the main focus was to represent the expected geological structures and not to make it stationary. In theory, MPS implementations assume stationary TIs (Liu et al., 2004b) and MPS application to strongly non-stationary systems is currently an area of research (e.g. Honarkhah and Caers, 2012;  Straubhaar et al., 2011; de Vries et al., 2009; Cuhgunova et al. 2008). The TI used for this study was generated by following the unique criterion that the unconditioned simulation (obtained by using SNESIM through its implementation in SGEMS) could satisfactorily reproduce the expected geological structures. Ideally, the TI should be constructed independently from the choice of MPS algorithm, and the realizations obtained by using a specific TI should have the same spatial variability as formalized by the training image. In practice though, this is rarely the case. While a specific part of the spatial statistics may be accurately reproduced, the realizations may lack geological features that can be crucial for subsequent modelling and interpretation. This is why the choice of MPS algorithm, and the parameters used to run the MPS algorithm have significant impact on the spatial structures seen on generated realizations. Hence, in practice, structural modelling should consist of choosing a TI together with a specific MPS algorithm (and the associated modelling parameters) to generate realizations capable to reflect the spatial variability that appears to be realistic from a geological perspective (Liu, 2006). Thus, the development of an effective TI involves an iterative procedure, where the realizations should be tested and evaluated. It is worth noting that even if the developed TI is not stationary, it is interpreted as stationary by the algorithm we have used (SNESIM; Strebelle, 2000). The realizations showed a significant sensitivity to the actual

choice of the 3D TI (Fig. 11). Thus, we highly recommend a careful evaluation of the unconditioned simulation results and subsequent, consistent TI optimisation.

A strategy to include available information into the stochastic simulation is via hard conditioning. Through hard conditioning, the realizations are forced to perfectly match the provided data. In this study, hard conditioning was used to

ensure a perfect correspondence between the simulation results and a former geological model available for the Tønder area. The results illustrate that this goal can be successfully reached even though the influence of the hard conditioning data remains quite local.

Also the seismic data, represented by interpretations along the seismic lines, have been treated as hard conditioning data. This can be debated since seismics is an "indirect" geophysical method that is inherently affected by uncertainty. The

reflections observed on seismic data represent changes in seismic velocity and/or density, but they are not necessarily related to lithological variations. Furthermore, the quality of the data can be highly varying as the resolution capability depends on the depth and, due to the uncertainty of seismic velocities used for depth conversion, also depths in the seismic sections are uncertain. In the present case, it was decided to use the interpretations along the seismics as hard conditioning data as their level of uncertainty is assumed to be much lower than the other available data (Kristensen et al., 2015), especially at the

scale required for the simulation.

Because of technical and economic limitations, the application of reflection seismics for shallow hydrostatgraphic studies is relatively recent (probably the first examples can be traced back to the '80s). However, as a consequence of the increasing and general awareness concerning the use and protection of water resources, during the last 40 years, seismics - together with several other geophysical techniques (e.g., the airborne electromagnetic methodologies) -  has been applied to many,

diverse hydrogeological characterizations with varying results (e.g., Francese et al. 2005; Giustiniani et al, 2008). Several of the problems in this kind of surveys lie: (i) in the presence of anthropic noise; (ii) the fact that a possible shallow water table may reflect the majority of the energy and, at the same time, mask low velocity geological features, preventing the effective reconstruction of deeper structures. Actually, in the attempt to overcome these difficulties, it has become more and more common to process and invert what is still often considered noise: the ground-roll (Strobbia, 2009). In fact, ground-roll

contains valuable information about the share velocity distribution in the subsurface and is generally characterized by high-amplitude. Recently high-resolution, shallow techniques based on surface waves have been developed and tested successfully for hydrogeological investigations (Vignoli et al., 2012; Vignoli et al., 2016). A possible limitation of techniques based on surface waves concerns the availability of low frequency sources: in presence of slow sediments, low frequencies are required to reach the desired depths, but generating them is very demanding and potentially detrimental to

the seismic vibrator.

In the present research, the use of the borehole data as hard conditioning data was considered suboptimal due to the low quality of many of the wells and the different discretization of the borehole data (1 m) compared to the size of the realization grid (5 m). The borehole data were therefore translated into soft probabilities by using a moving window strategy that, in one shot, takes into account the borehole information uncertainty and the different scale issue. A uniform uncertainty of 20 % has

been assumed for the boreholes across the entire domain; however, in future studies, it would be straightforward to extend the present approach and locally rescale the soft probabilities according to a quality rate of each individual borehole as, e.g., the one presented in He et al. (2014). In this way, poor quality borehole would influence the simulation less than more reliable data.

One of the difficulties in the development of a proper MPS realization of the Miocene sequence was that the sand content varied spatially across the model domain making the sequence non-stationary. A 3D sand probability grid (Fig. 9) was generated in order to migrate the information further from the boreholes and constrain the geostatistical simulation to follow the spatial sand/clay trend characterizing the study area. The soft probability grid can consequently be seen as a shortcut to address the non-stationarity, such that the sand/clay content derived from the TI was overruled by the probability grid.

Another possible solution could consist in the creation of different TIs to represent the end-members, and then interpolate these to obtain gradual changes dependent on the positions as discussed in Mariethoz and Caers (2014). Actually, if SNESIM could correctly handle soft data in the form of localized conditional probability, kriging the borehole probabilities would be unnecessary (Hansen et al., in review).

In addition to the data already incorporated as (soft and hard) conditioning data in this study, dense electromagnetic (EM)

surveys, on approximately half of the area, are available in the Danish geophysical database Gerda (Møller et al., 2009). An option would be to use these resistivity data in the MPS modelling as soft conditioning data, such that high electric resistivities indicate sand, while clay corresponds to low resistivity values (see, for instance, He et al., 2014; He et al. 2016). However, EM data have a limited resolution capability towards thin layers (Ley-Cooper et al., 2014; Vignoli et al, 2017), especially in the deeper parts of the investigated sequences. The modelled unit is generally present at great depths (from a

range between 30 m and 80 m, in the east, to a range between 150 m and 170 m, in the west) and the resolution of the EM data is consequently quite low in most of the area. This is especially the case in the west, where the Miocene unit is located below the depth of investigation (Christiansen and Auken, 2012). Another challenge regarding the use of resistivity data for detecting sand and clay within the Miocene deposits is the common occurrence of silt (Rasmussen et al., 2010). Some of the clayey formations are very silty and, while silt has small grain sizes and hydraulic conductivities, it has high resistivities.

Thus, clayey formations with high silt contents might show higher resistivities than expected, leading to potentially wrong interpretations. In addition to that, during the borehole description phase, silt is often recognised as clay, and this clearly causes a mismatch between the borehole and resistivity information.

Geostatistical simulation methods are most commonly used to create multiple realizations whose variability represents the combined (uncertain) information; for instance, with the purpose to estimate uncertainties of structural variability (Feyen and

Caers, 2006; He et al., 2013; Poeter and Anderson, 2005; Refsgaard et al., 2012) or to make probability calculations to be used for various forecasts (Stafleu et al., 2011, Christensen et al., 2017). In the majority of this study (see Appendix A), only one realization of each of the conditional strategies was presented. The reason is two-fold. Firstly, the primary goal of this paper is to describe a workflow for choosing a training image, and to combine all available information into one consistent probabilistic model. And each individual realization (e.g., in Fig. 18) is, by construction, compatible with all simulation

inputs (thus, the statistics from the TI, the hard data, and the soft conditioning). So, to set up the proposed workflow, only the analysis of single realizations is necessary and, in the study of the performances of the different conditioning strategy, only the features present, by definition, in each realization have been taken into consideration. Secondly, the realization generated in the final test (Fig. 18) was, at the end, incorporated in an overall geological model (Meyer et al., 2016): whereas, in this case, using a single realization is considered acceptable since the objective is to study large-scale groundwater flow and saltwater intrusion, the use of multiple realizations for the propagation of uncertainty into, for example, hydrological models would be outside the scope. To make groundwater predictions on a large scale, the overall distribution and connectivity of the overall structures are crucial, while the precise location of the individual structures is less important. In contrast, detailed studies, like catchment analyses, would not make sense based on a single realization (He et al., 2013) and assessments based on a significant number of different realizations should be conducted to investigate the variability of the outcome.

The validity of the presented workflow is demonstrated for the Miocene unit characterized by relative simplicity and presence of only two categories. This does not mean that the applicability of this approach should be limited to simple situations. This research can be considered a proof of concept and its intent is to clearly show the relevance and effectiveness of our strategy in addressing the difficulties frequently encountered, even in simple cases. We do not see any particular difficulty in extending the proposed strategy to more complex settings characterized, for example, by a larger number of categories. In fact, if this is the case, dealing with three or more categories makes the preparation of the hard conditioning data (e.g., the interpretations of seismic lines) clearly more laborious, but conceptually not more difficult. The same is true for the way we handle the borehole data as soft conditioning: definitely, the implementation of the sliding window and the following kriging procedure are not different if a larger number of categories are involved. Finally, MPS approaches (together with the associated TIs) are already routinely used in situations with more than two categories (e.g., Jones et al. 2013), and, in our approach, special emphasis is placed uniquely on the strategy for the development of effective TIs via careful analyses of the unconstrained realizations.

## 8 Conclusions

This study investigates strategies for MPS simulations in large 3D model domains consistent with different types of input data. The strategies were tested within an area of $2\,810$ km$^2$ in which the Miocene unit was modelled using MPS simulation. This part of the model was chosen since the Miocene can be effectively subdivided into few categories (i.e.: sand and clay) and is relatively stationary in the investigated area. An already existing and detailed geological model (the Tønder model) was present in a part of the study area, and was used in the final comprehensive geological model. A 3D TI was constructed based on the well-known geology of the unit. The stochastic simulations were conducted using the SNESIM algorithm as it is implemented in SGeMS. The final TI was developed iteratively by checking the outcomes of the corresponding unconditioned simulations, and adjusting it in order to obtain the most geologically meaningful structures in the final realizations. The previously published Tønder model and reliable seismic interpretations were used as hard conditioning data

in order to preserve the associated information during the simulation. On the other hand, the borehole data were incorporated into the simulation workflow through different conditioning strategies. The first approach - the most traditional one - consisted of using the borehole data as hard conditioning. This quite standard approach was not satisfactory as borehole data can have a high degree of uncertainty. Hence, via a moving window strategy, the lithological information in the borehole was translated into a probability distribution that could address uncertainties in borehole data both from inaccuracies and scale mismatch. Unfortunately, SNESIM limits the influence of soft conditioning data to local neighbourhoods around each data value and is unable to effectively regionalize any trends that might be captured. To better address this problem, we kriged the sand probability derived from the boreholes into a 3D voxel model and used that kriged sand probability as soft conditioning. By using this last approach, we managed to successfully reproduce the sand/clay trend across the simulation domain evaluated based on a visual inspection. The study shows a practical workflow to properly build TI and effectively handle input information to be successfully used for large-scale geostatistical modelling.

## 9 Acknowledgements

The study is a part of the research project ERGO: "Effective high-resolution geological modelling", which is funded by the Innovation Fund Denmark. In the ERGO project, 3D geological modelling tools have been developed and integrated into the software package GeoScene 3D. The study is also connected to the Geocenter project SaltCoast: "Large-scale hydrogeological modelling of saltwater intrusion in a coastal groundwater system".

We are grateful to Wolfgang Scheer, who kindly provided us the borehole information from the German part of the study area and to Torben Bach, I-GIS, and Ingelise Møller, GEUS, for fruitful discussions.

We would like to thank Anders Juhl Kallesøe, who contributed to the project by modelling the Top Pre-Quaternary surface.

Finally, we would like to express our gratitude to DICAAR at University of Cagliari for the computational resources used for the preparation of Appendix A.

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

**Appendix A**

Despite the discussion of the rest of the paper is realization-independent, it might be worth highlighting some of the effects of our approach on different realizations of the obtained probabilistic models. On this respect, Fig. 20 confirms what is in Fig. 12. Thus, the use of TI2, instead of TI1, consistently promotes the presence of larger sand bodies also when the conditioning strategy (d) in Table 1 is adopted. In addition, the two panels of Fig. 20 demonstrate that similar size distributions are common for all the realizations generated with a specific TI. Analogous behaviours clearly appear also when we compare the other two properties analysed previously: eccentricity (Fig. 21) and jaggedness (Fig. 22). If we compare TI1's with TI2's results for each of the Fig.s 21 and 22, we can draw the same conclusion as for Fig.s 13-14. Hence, elongated and jagged sand structures are more probable in the realizations obtained by using TI1. And, again, within the same probabilistic model, the distributions of eccentricity and jaggedness are very consistent between the realizations. Of course, this should not lead us to the wrong deduction that there is no variability between the elements of the probabilistic model. For example, Fig. 23 compares the conditioning approaches (c) and (d) in Table 1 in terms of e-type and variance maps: Fig.s 23b-c concern the conditioning strategy (c), while Fig.s 23e-f result from the application of approach (d). To make this comparison easier, also the corresponding (soft and hard) conditionings are shown in the same figure: in the panel (a), for the strategy (c) and, in the panel (d), for the approach (d). Fig. 23 makes evident the role of the seismic lines and the buffer zone from the Tønder model (used as hard data in both conditioning strategies) in constraining all the realizations of both probabilistic models. This is particularly clear when we observe the vertical W-E sections (Fig. 23c-f): the only place characterized by an extremely limited variance is where the intersection with the seismic line occurs. Fig. 23 highlights also the crucial role of the 3D kriged grid of the sand probability distribution in migrating the information far from the boreholes. This confirms, once more, the validity of the suggested way to handle the well information.

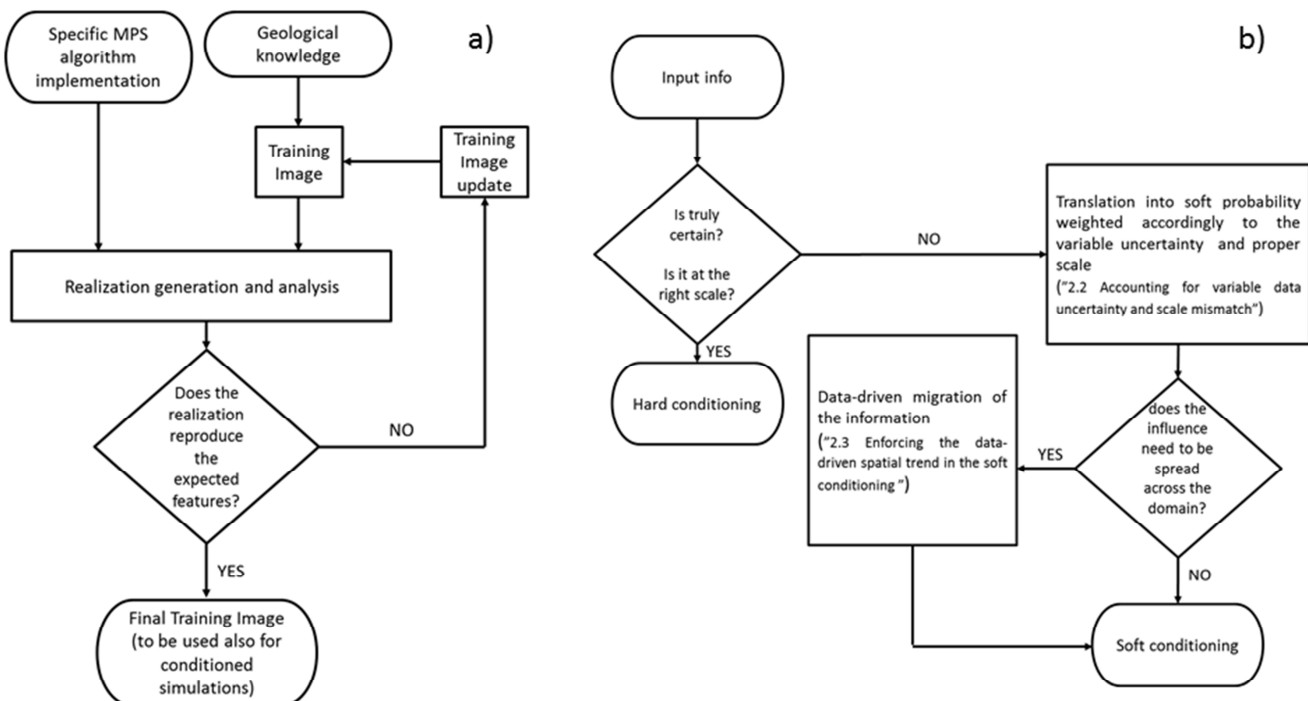

**Figure 1: a) flowchart of the iterative TI construction (see "2.1 Optimal training image development"), b) scheme of the proposed conditioning strategy.**

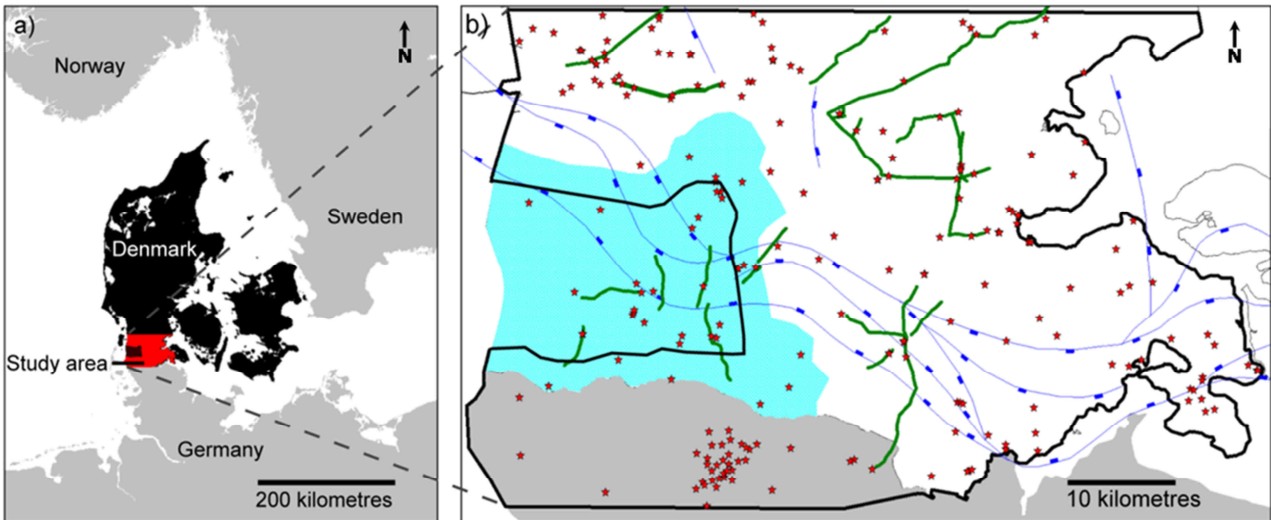

**Figure 2: a) Location of the study area, b) Map of the model area (the solid black line) shown together with the data and the fault structures delimiting the Tønder graben: red stars indicate boreholes deeper than 100 m, green lines the position of the seismic lines; the Tønder graben structure is marked in blue (Ter-Borch, 1991). The turquois shading marks the model area of the Tønder model (Jørgensen et al., 2015); the grey zone in map (b) represents Germany.**

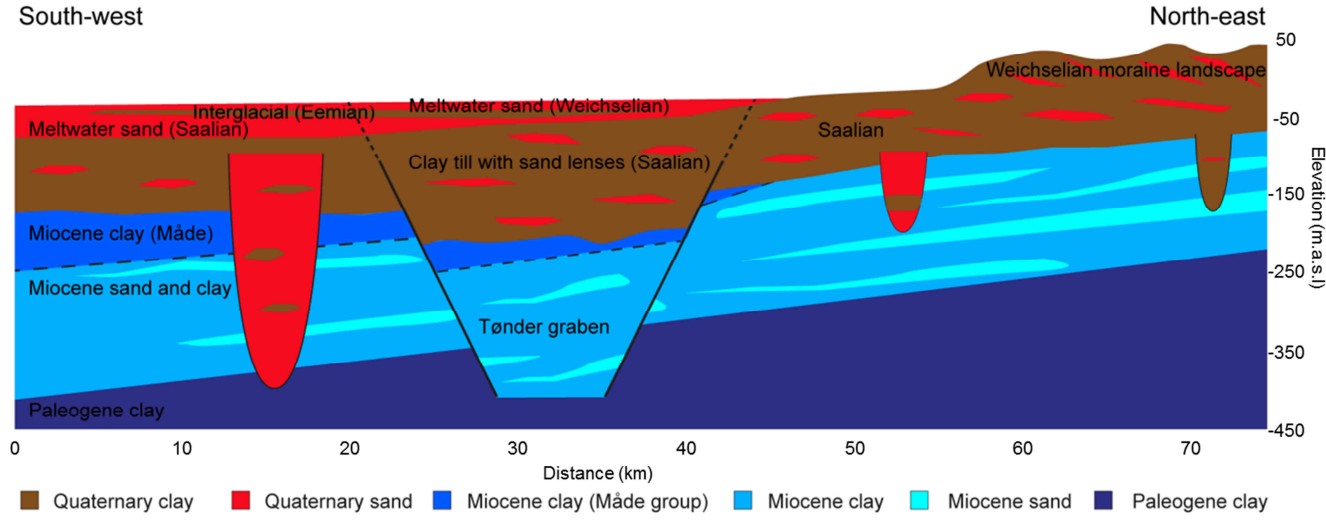

**Figure 3: Conceptual sketch of the geology in the study area.**

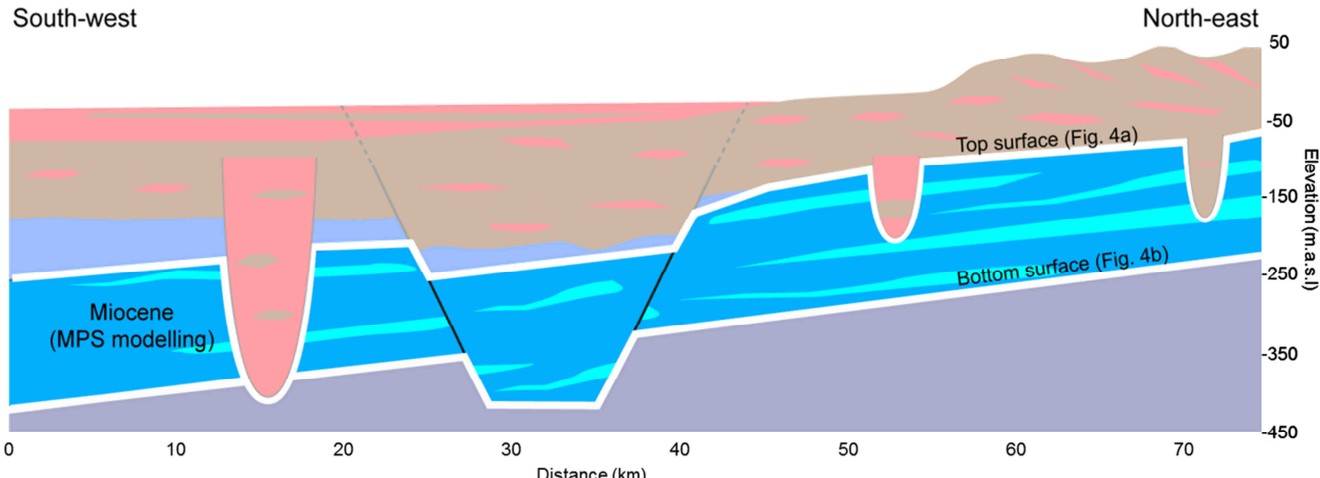

**Figure 4: Sketch of the MPS model domain based on the conceptual sketch of the geology in Fig. 3: Top and Bottom of the Miocene unit is outlined by thick white lines. The surfaces are shown in Fig. 5a and 5b.**

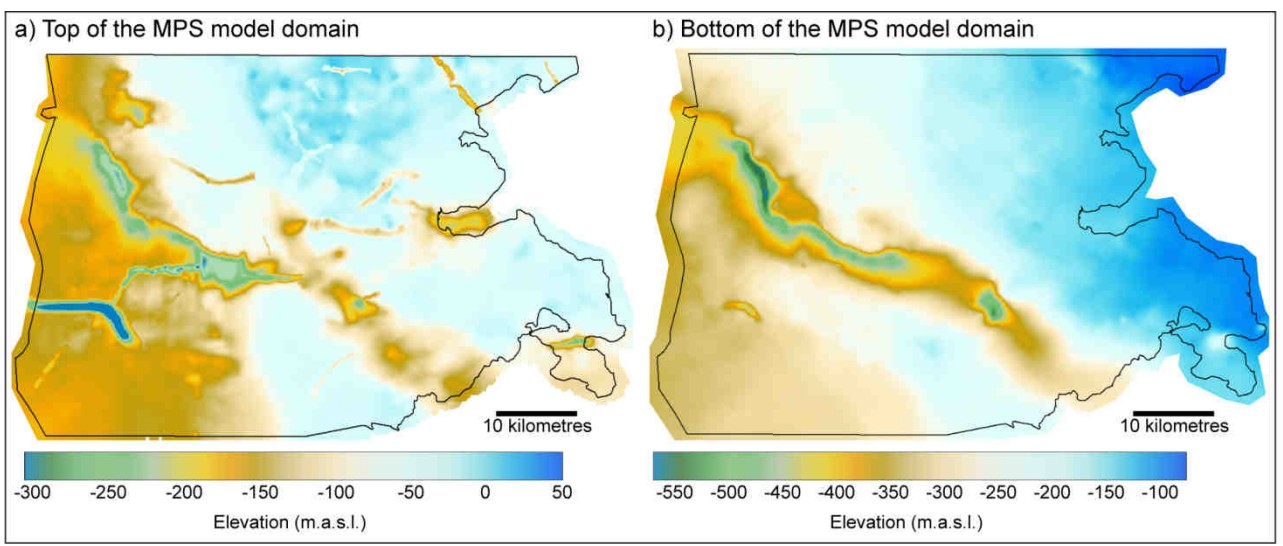

**Figure 5: Elevation of the top and bottom of the MPS model domain (the Miocene unit). a) The modelled surface defining the top of the MPS model domain. The surface is obtained by merging the bottom of the Måde Group and the top pre-Quaternary (see Fig. 4). b) The modelled surface of the bottom of the MPS model domain, which consists of the Top Paleogene clay. Note: Different colour scales.**

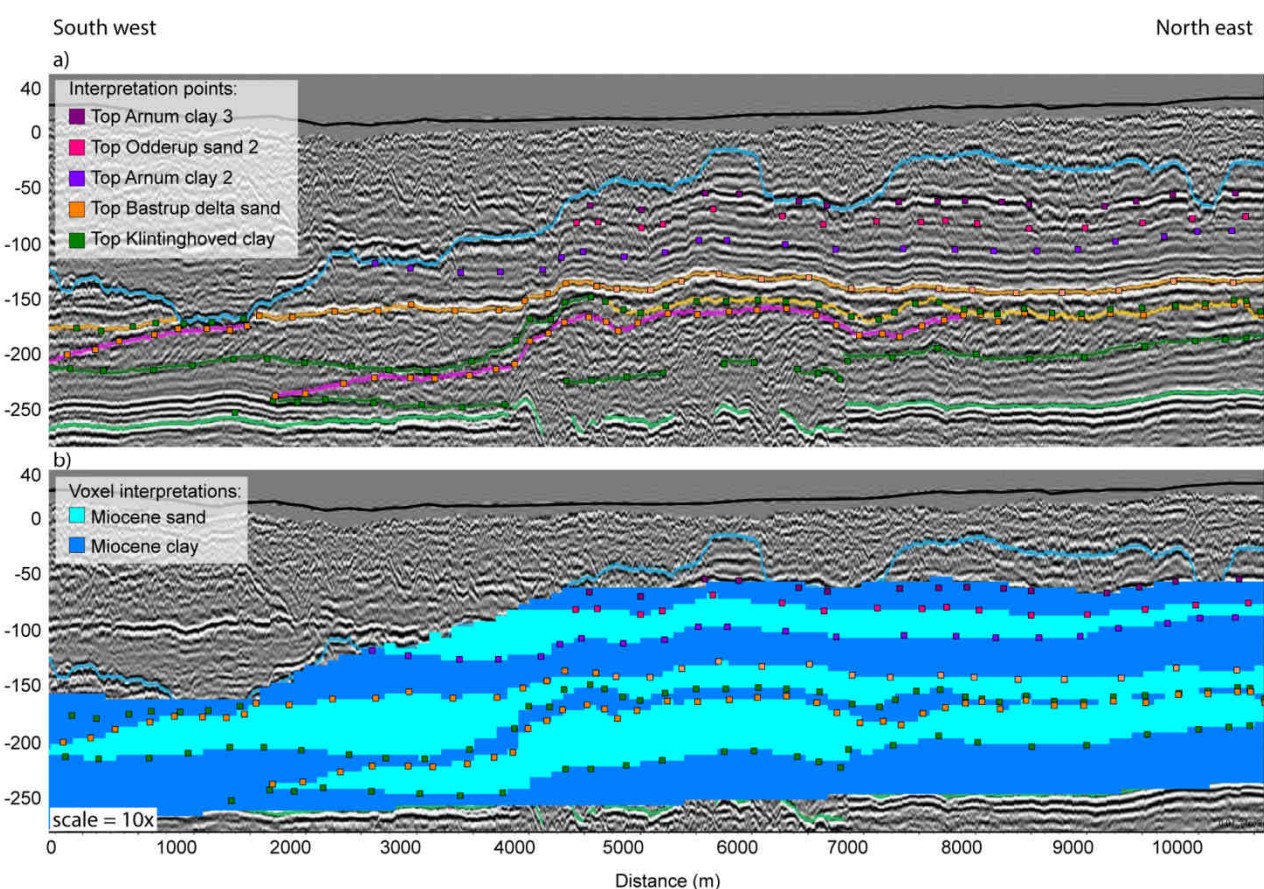

**Figure 6: Preparation of seismic data input exemplified by a selected seismic section (for position, see Fig. 7). a) The seismic section shown together with coloured horizons and interpretation points derived from the Miocene model (Kristensen et al., 2015). b) The seismic section shown together with the generated "2D voxel model". Vertical exaggeration = 10x.**

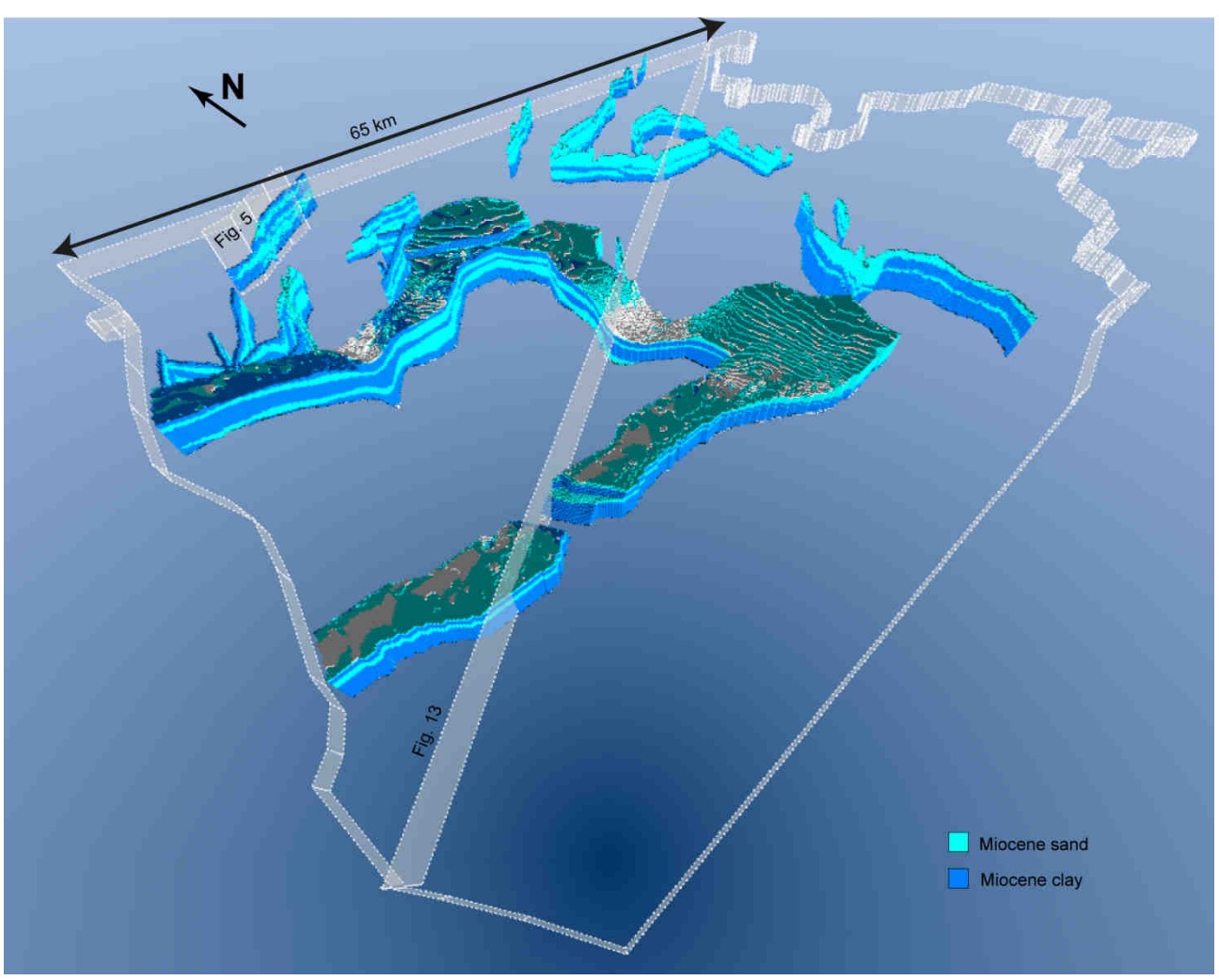

**Figure 7: 3D view of the hard conditioning data (the buffer zone around the Tønder model and the seismic lines), seen from south-west. Positions of the seismic line in Fig. 6 and the profile in Fig. 17 are marked on the figure. Vertical exaggeration = 10x.**

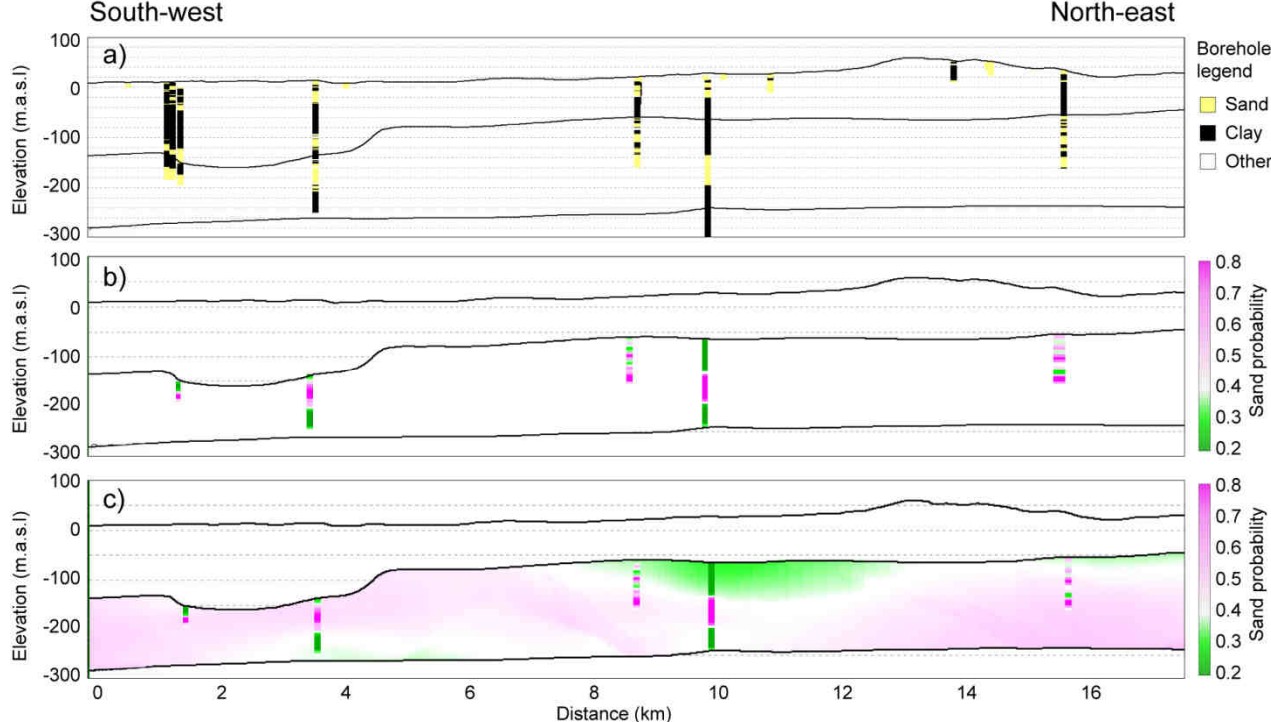

**Figure 8: Three different ways to handle borehole information illustrated along a profile in the study area (for position, see Fig. 15). The resulting realizations are shown in Fig. 13. a) The sand/clay occurrence in the borehole is considered as hard data. b) The sand/clay occurrence is translated into sand/clay probability and then considered as soft data. c) The sand probabilities from the boreholes are kriged into a 3D grid to be used as soft conditioning. Vertical exaggeration 8x.**

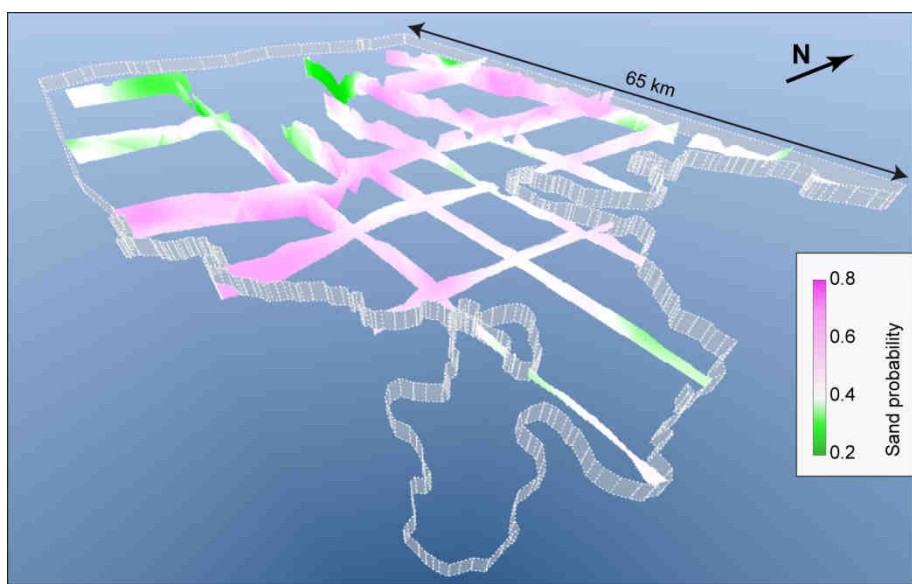

**Figure 9: 3D fence view into the sand probability grid, seen from south-east. Vertical exaggeration = 10x.**

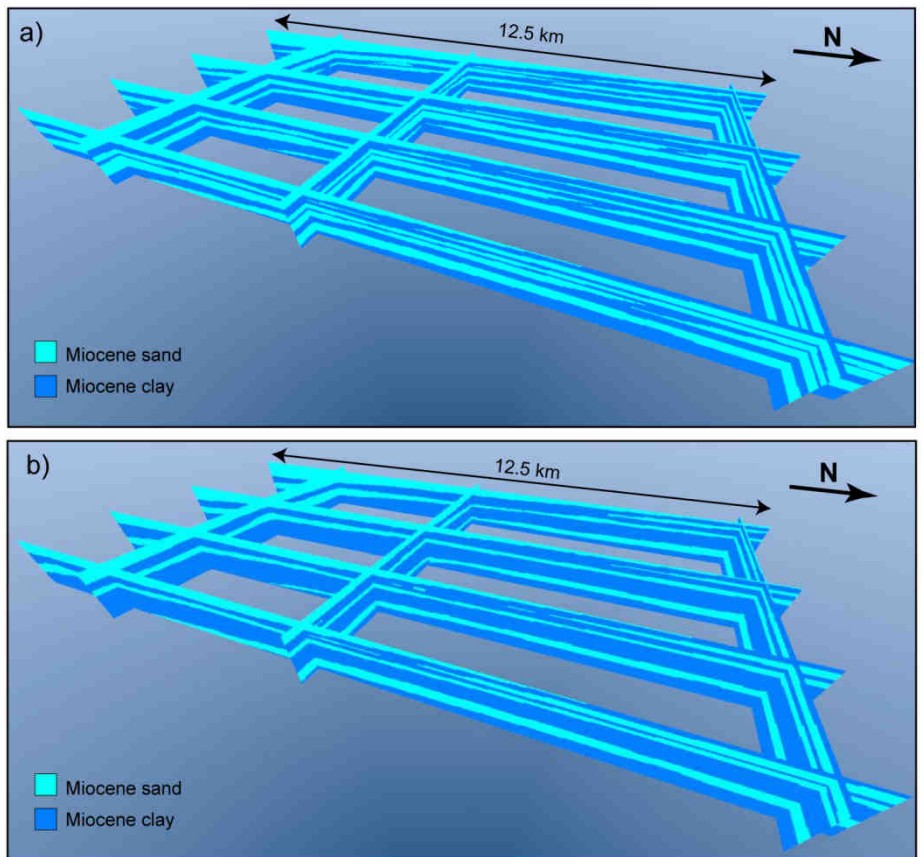

**Figure 10: N-S and E-W slices through the two tested 3D training images seen from north-east. a) The first TI (TI1), used to generate the unconditioned realization in Fig. 11a. b) The second TI (TI2), used for the unconditioned realization in Fig. 11b. Vertical exaggeration = 3x.**

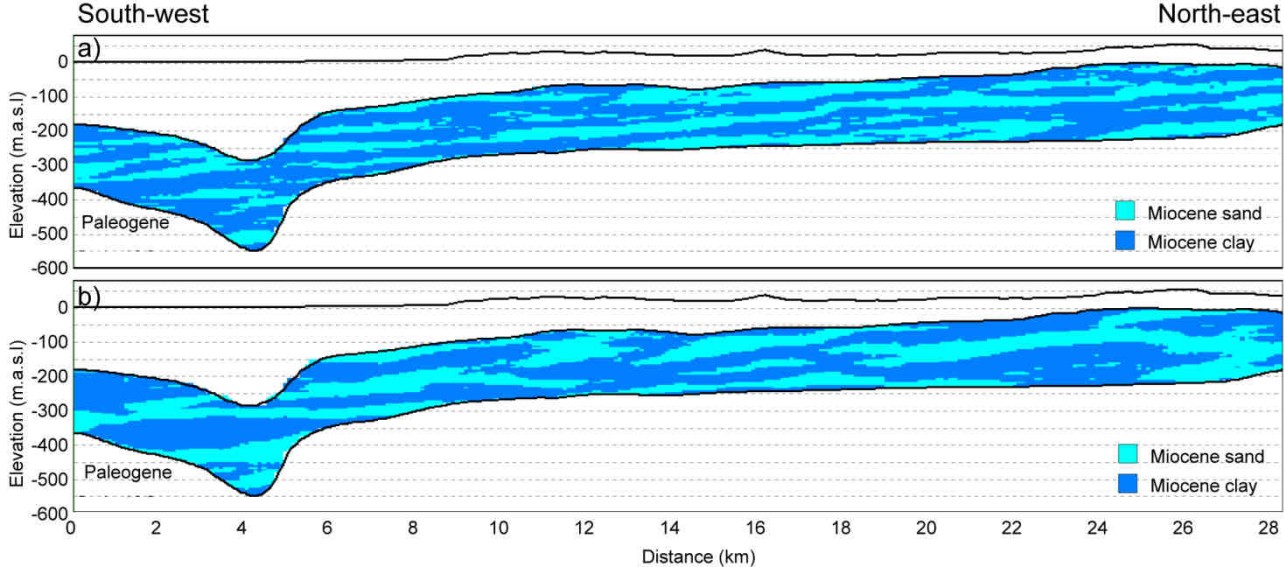

**Figure 11: Profile through the unconditioned realizations obtained by using the two different TIs shown in Fig. 5. a) Realization based on TI1 (Fig. 10a). b) Realization based on TI2 (Fig. 10b). Vertical exaggeration = 8x. For location of the profile, see Fig. 15.**

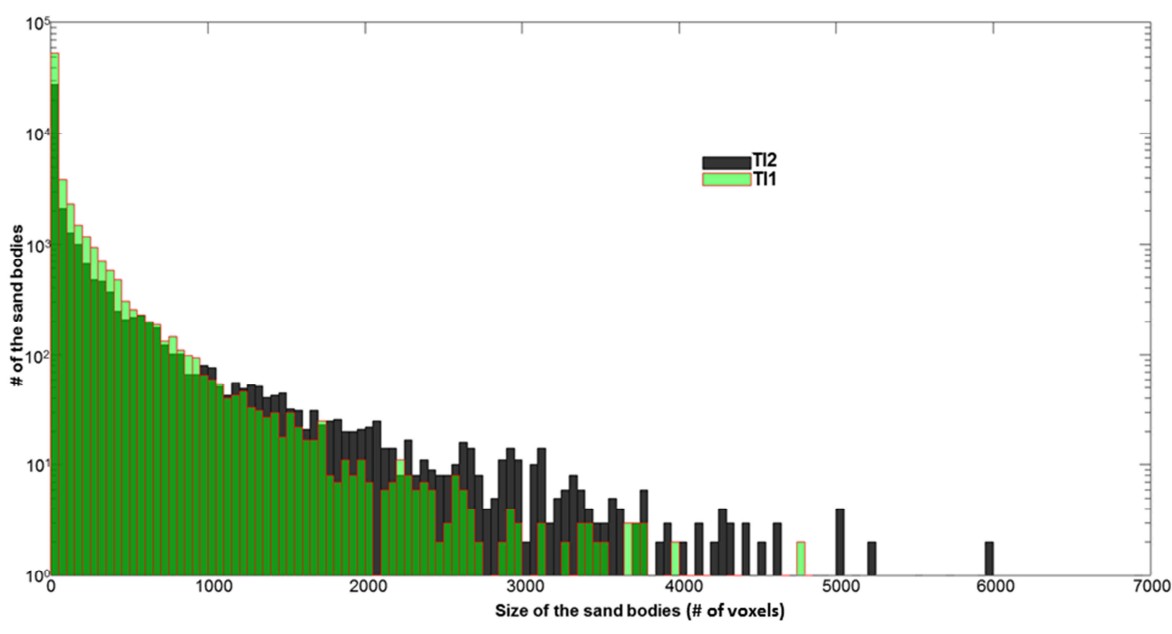

**Figure 12: Sizes of the sand bodies of the unconditioned realizations obtained by using the two different TIs shown in Fig. 10.**

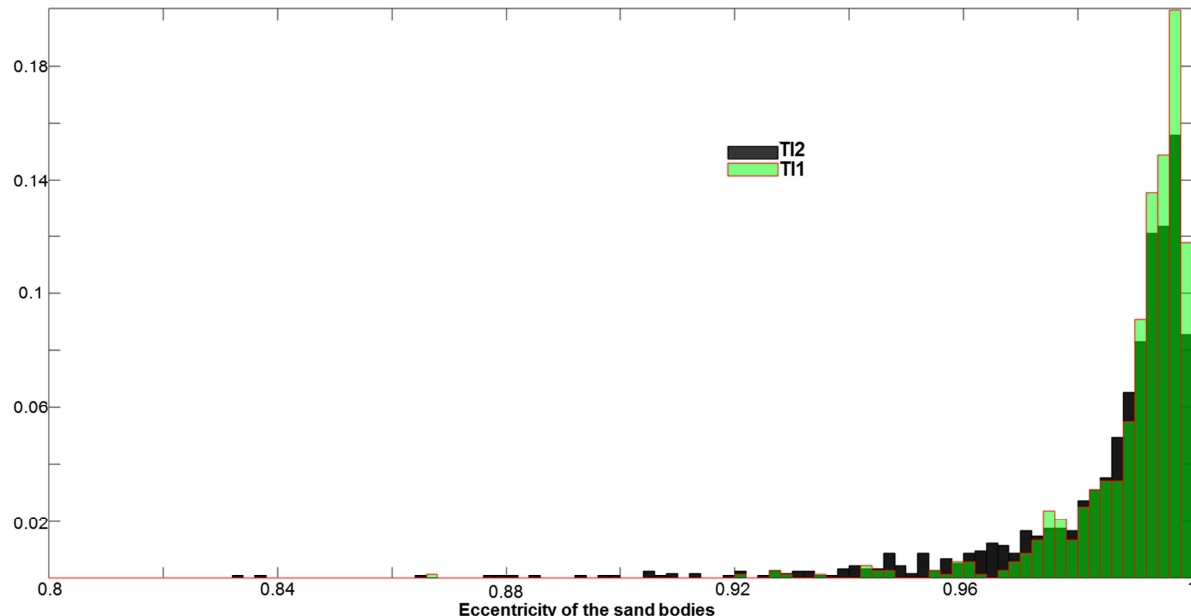

**Figure 13: Probability of the eccentricity of the sand bodies (with a size larger than 1000) for the unconditioned realizations obtained by using the two TIs in Fig. 10.**

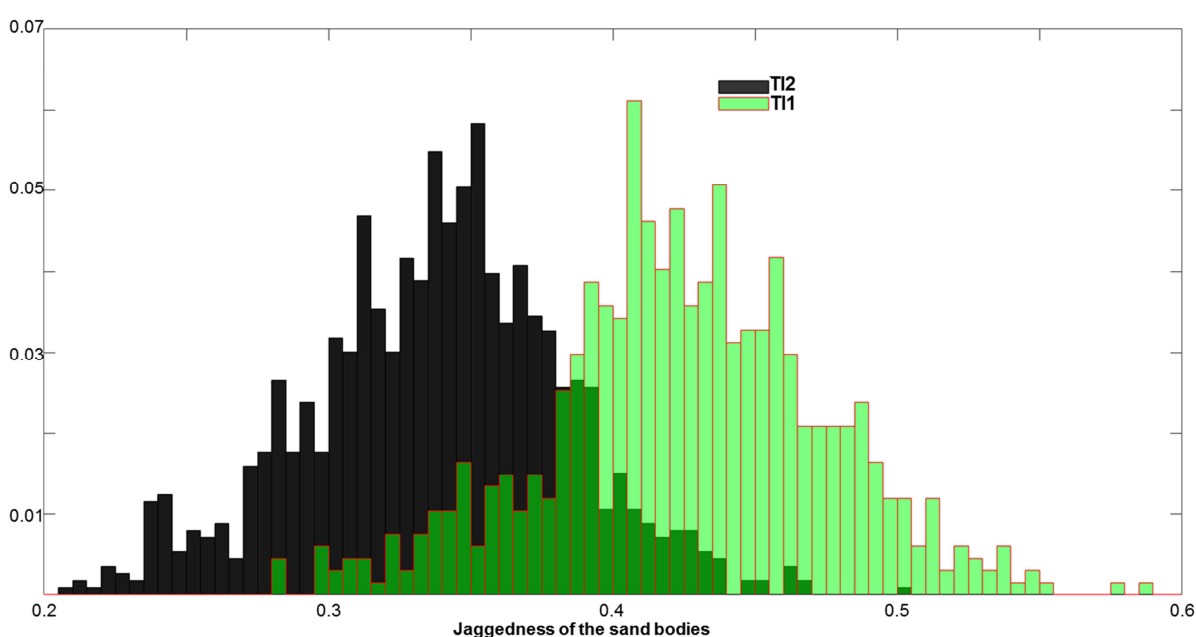

**Figure 14: Probability of the jaggedness of the sand bodies (with a size larger than 1000) for the unconditioned realizations obtained by using the TI1 and TI2 in Fig. 10.**

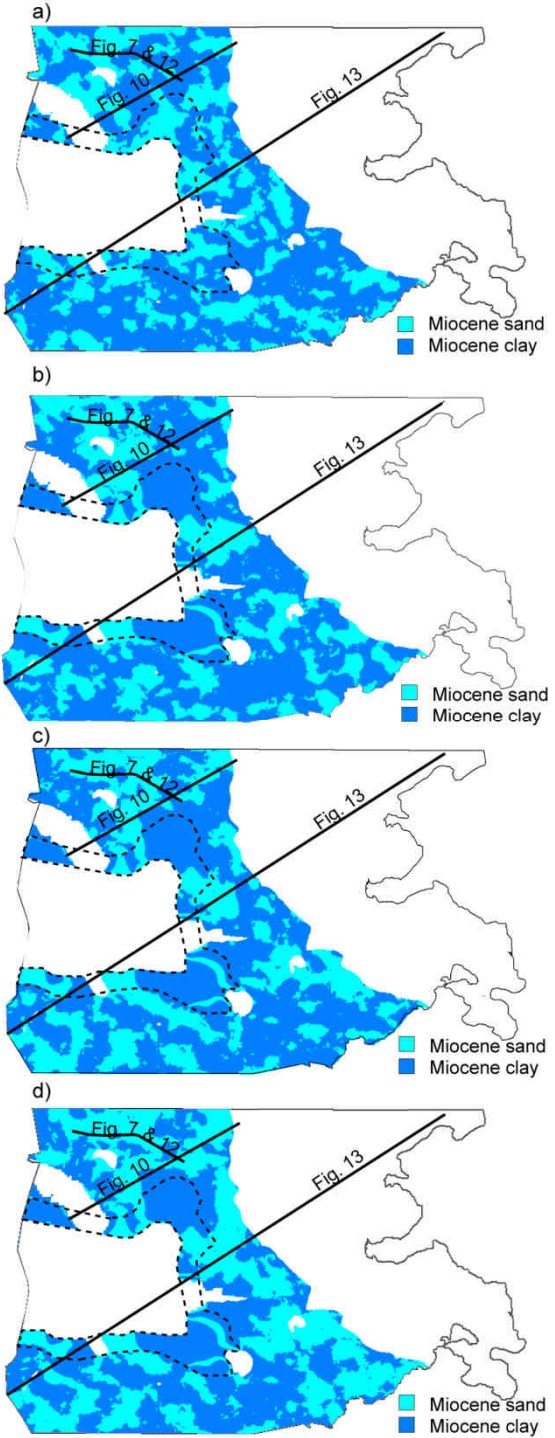

**Figure 15: Horizontal slice at elevation -188 m through the realizations generated by using different conditional strategies (see Table 1). Position of the profiles in Figs. 8, 11 & 13 and 14 are showed. The dashed region encapsulates the buffer around the Tønder model, which has been used as hard conditioning information in (b), (c) and (d).**

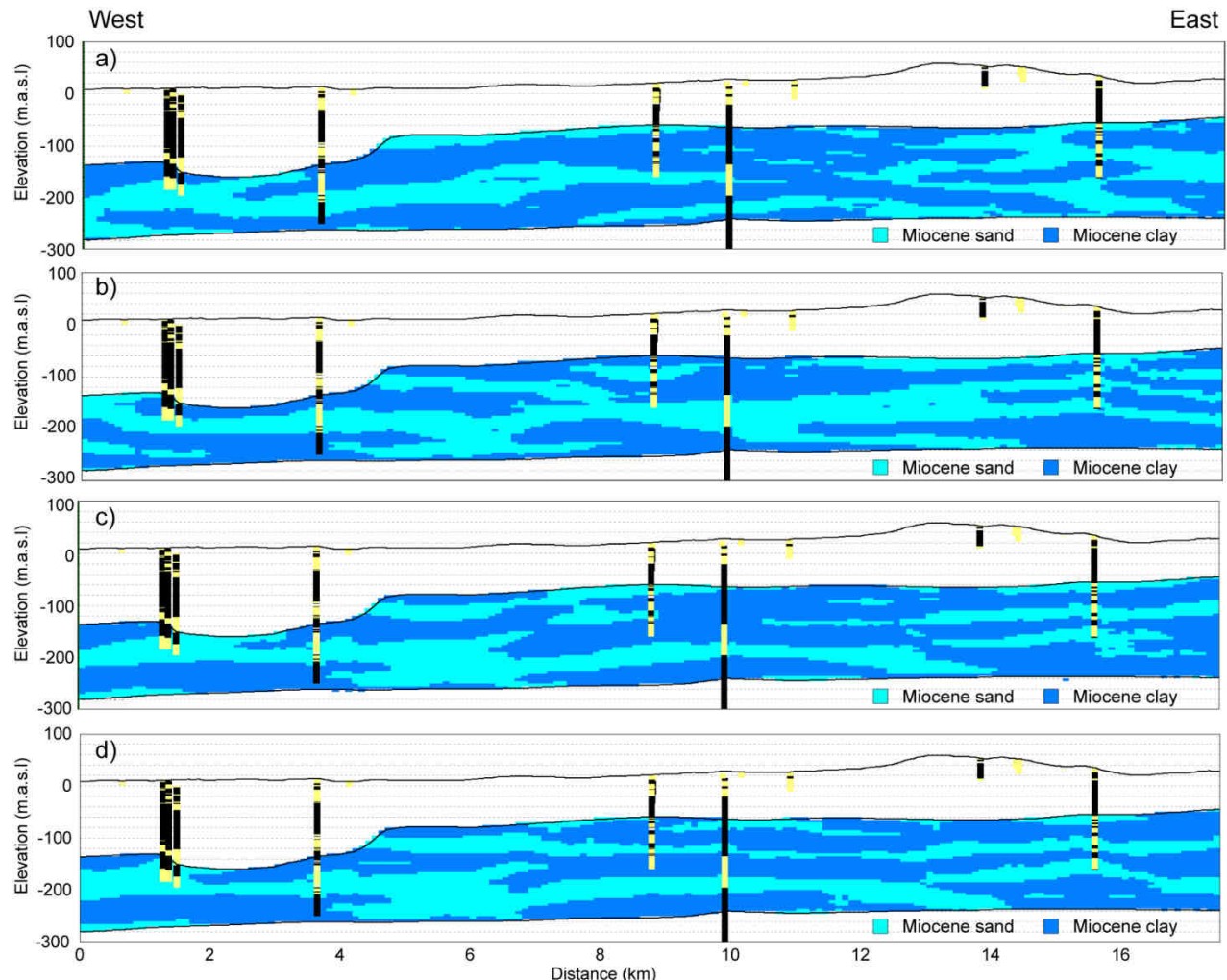

**Figure 16: Profile through the realizations generated by using different conditional strategies (see Table 1). The borehole information used for conditioning is illustrated in Fig. 8: a) unconditioned; b) borehole as hard data (Fig. 8a); c) borehole as soft data (Fig. 8b); d) sand probability grid derived by boreholes as soft data (Fig. 8c). Sand/clay information from the original boreholes are showed within a buffer of 100 m (yellow = sand, black =clay, white = 'other'). For location of the profile, see Fig. 15. Vertical exaggeration = 8x.**

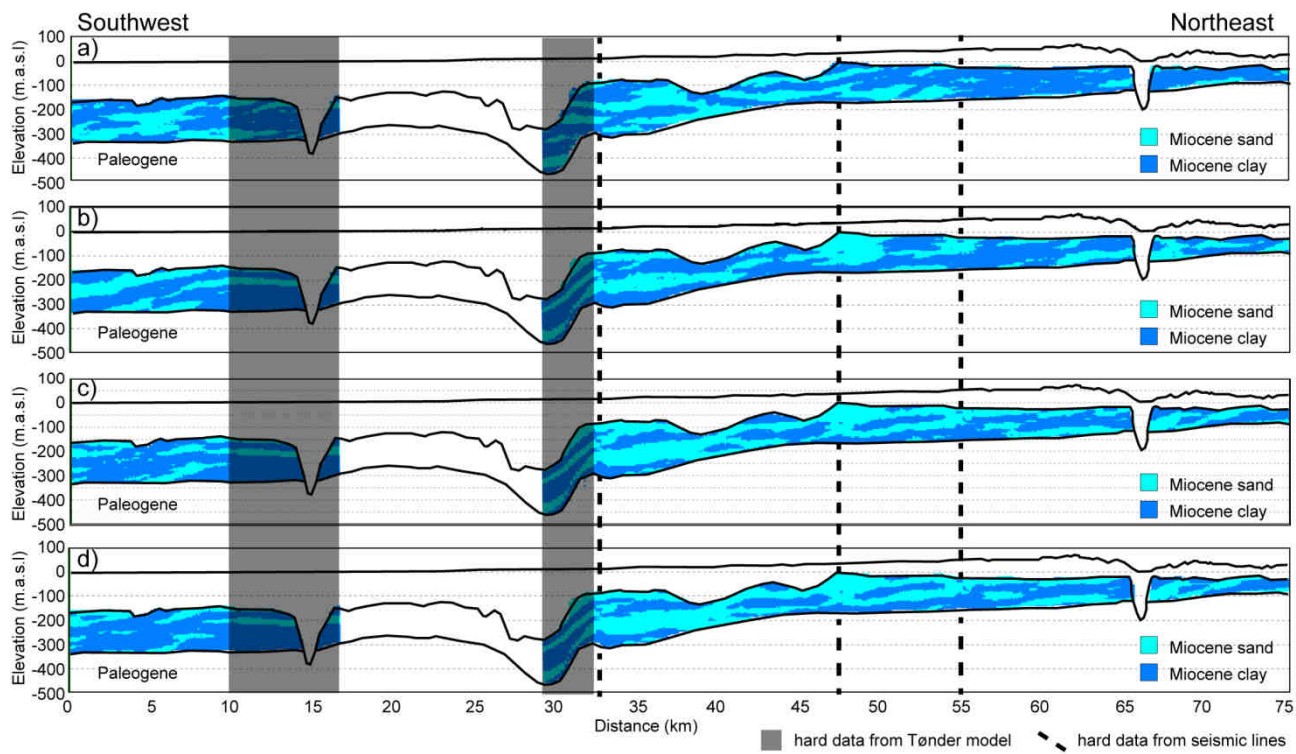

**Figure 17: Long SW-NE profile through the realizations generated by using different conditional strategies (see Table 1). For location of the profile, see Fig. 15. Vertical exaggeration = 15x.**

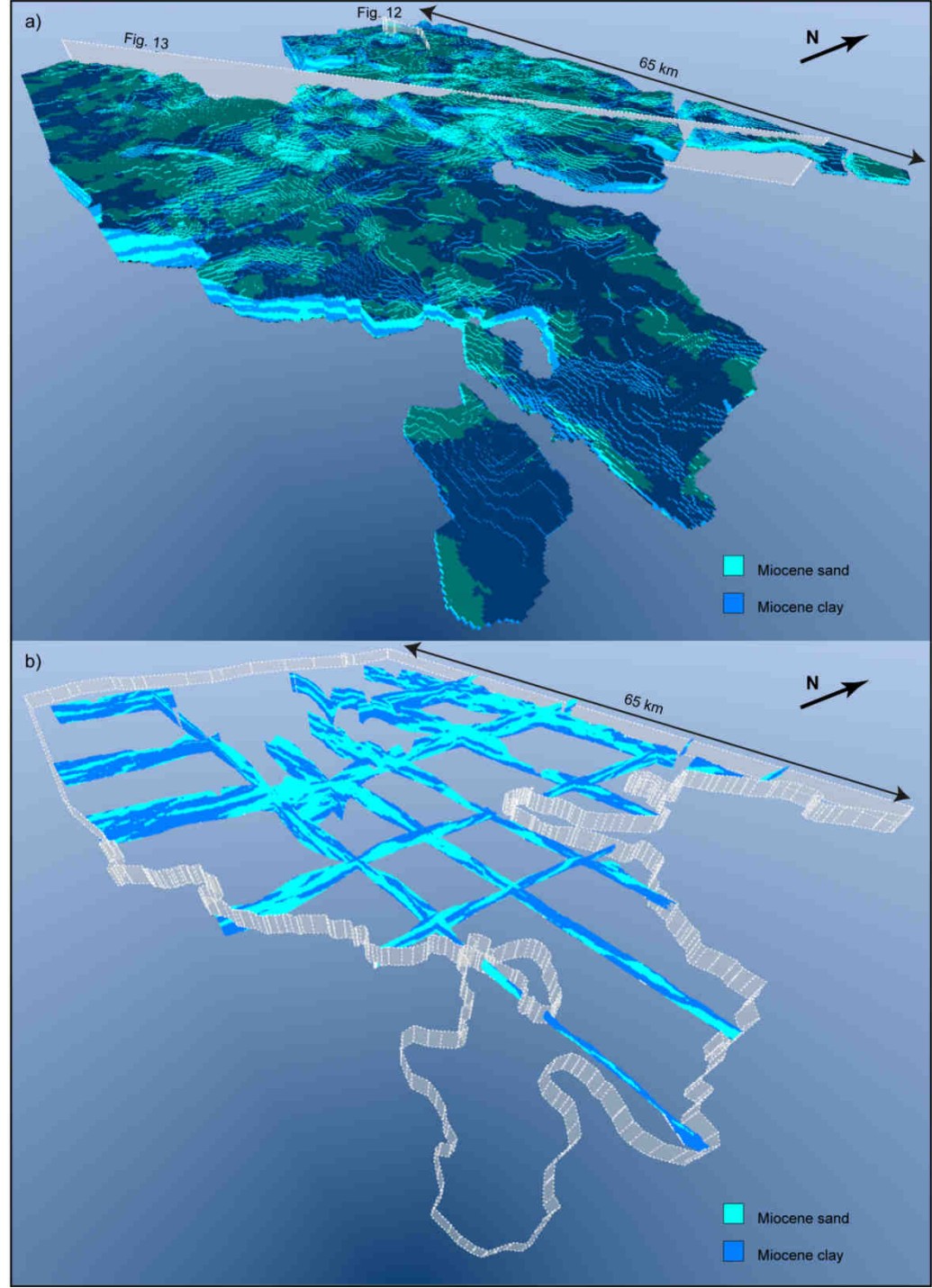

**Figure 18: 3D view of the final realization (test (d), Table 1). The associated 3D probability grid is shown in Fig. 9. a) All voxels are plotted and the location of the profile in Figs. 13 and 14 are shown. b) The associated fence view. Vertical exaggeration = 10x.**

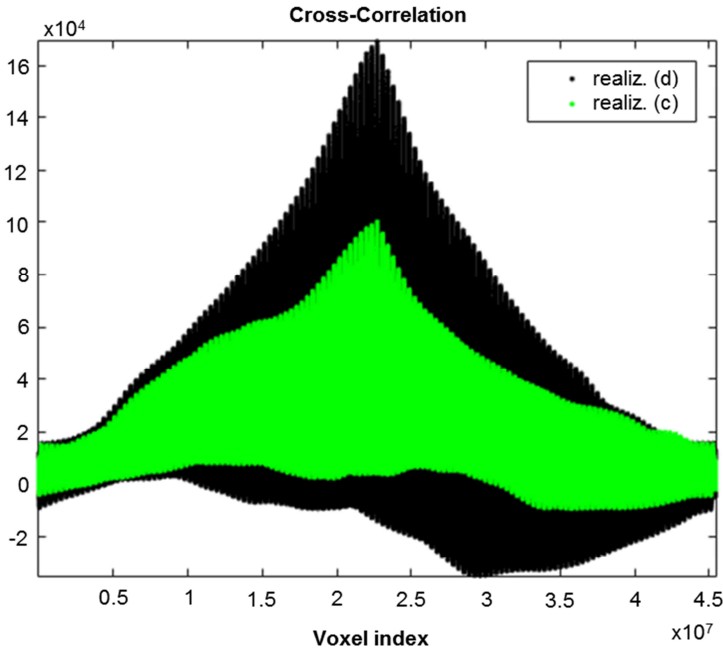

**Figure 19: Cross-correlation between the soft probability distribution (Fig. 9) and each of the realizations showed, for example, in Fig. 15c (real. (c)) and Fig. 15d (real. (d)), and described in Table 1 (cases (c) and (d)).**

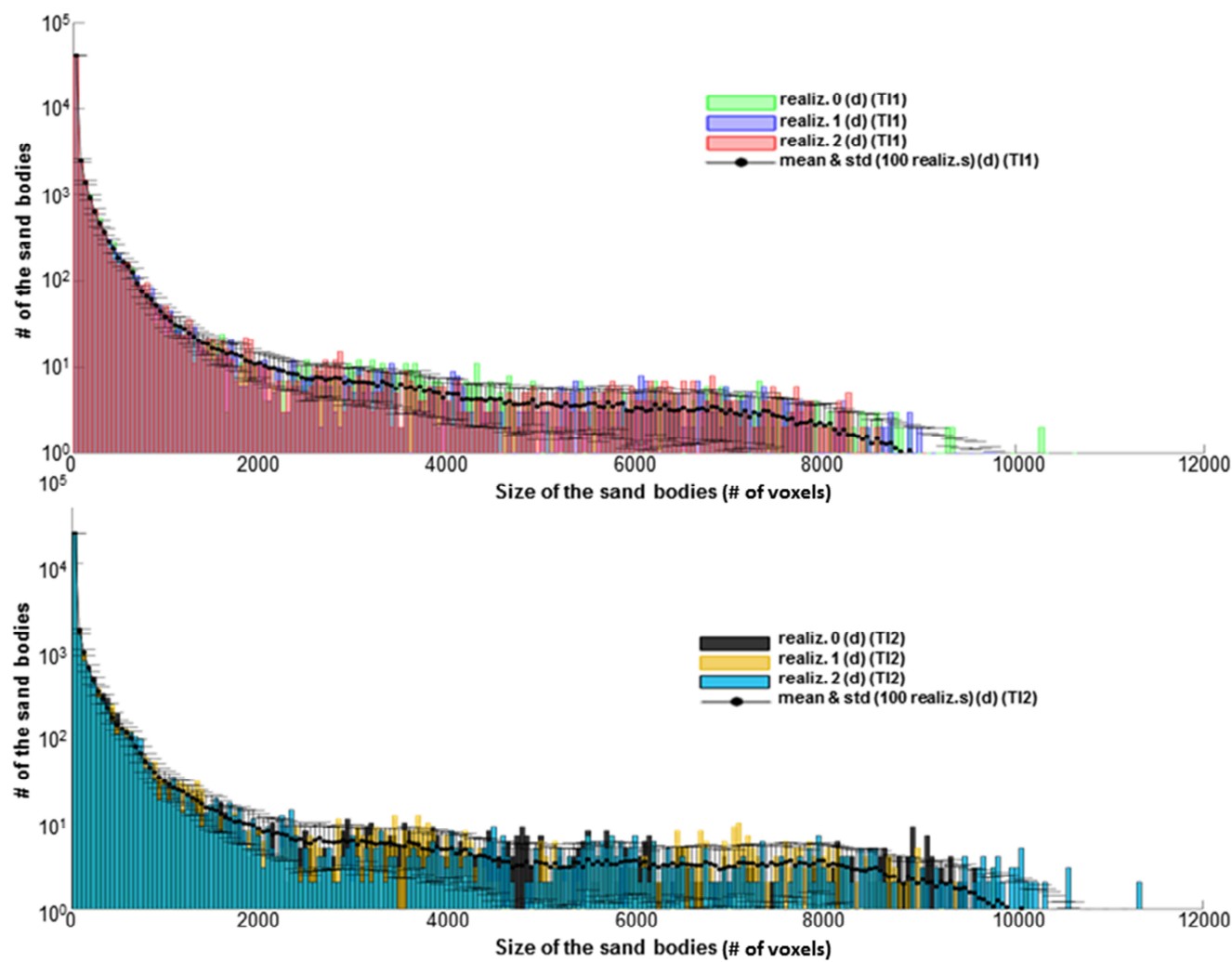

**Figure 20: Sizes of the sand bodies for the realizations obtained by using the conditioning strategy (d) in Table 1 and, respectively, TI1 and TI2 (Fig. 10). While the histograms correspond to the first three realizations, the dots and error bars represent the mean and the standard deviation obtained by using 100 realizations.**

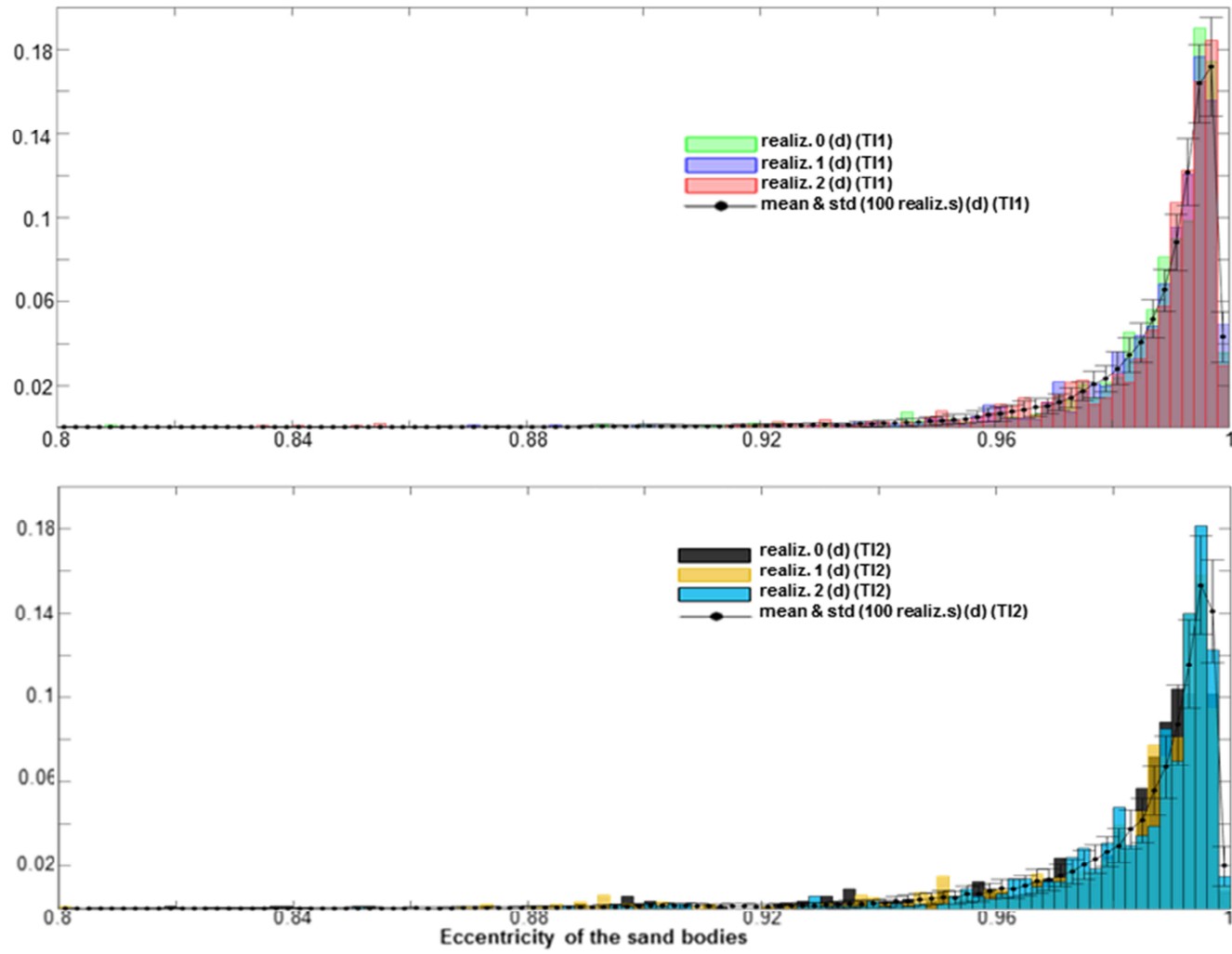

**Figure 21: Probability of the eccentricity of the sand bodies (with a size larger than 1000) for the realizations obtained by using the conditioning strategy (d) in Table 1 and, respectively, TI1 and TI2 (Fig. 10). While the histograms correspond to the first three realizations, the dots and error bars represent the mean and the standard deviation obtained by using 100 realizations.**

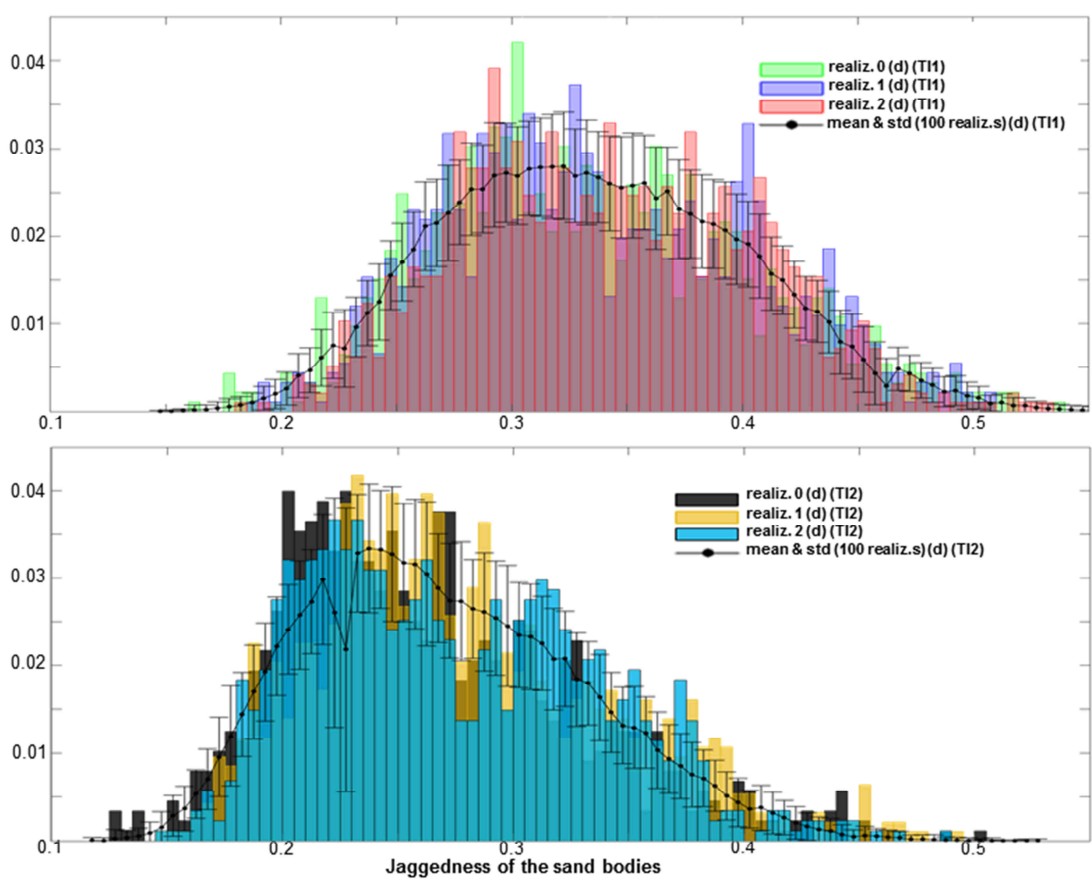

**Figure 22: Probability of the jaggedness of the sand bodies (with a size larger than 1000) for the realizations obtained by using the conditioning strategy (d) in Table 1 and, respectively, TI1 and TI2 (Fig. 10). While the histograms correspond to the first three realizations, the dots and error bars represent the mean and the standard deviation calculated by using 100 realizations.**

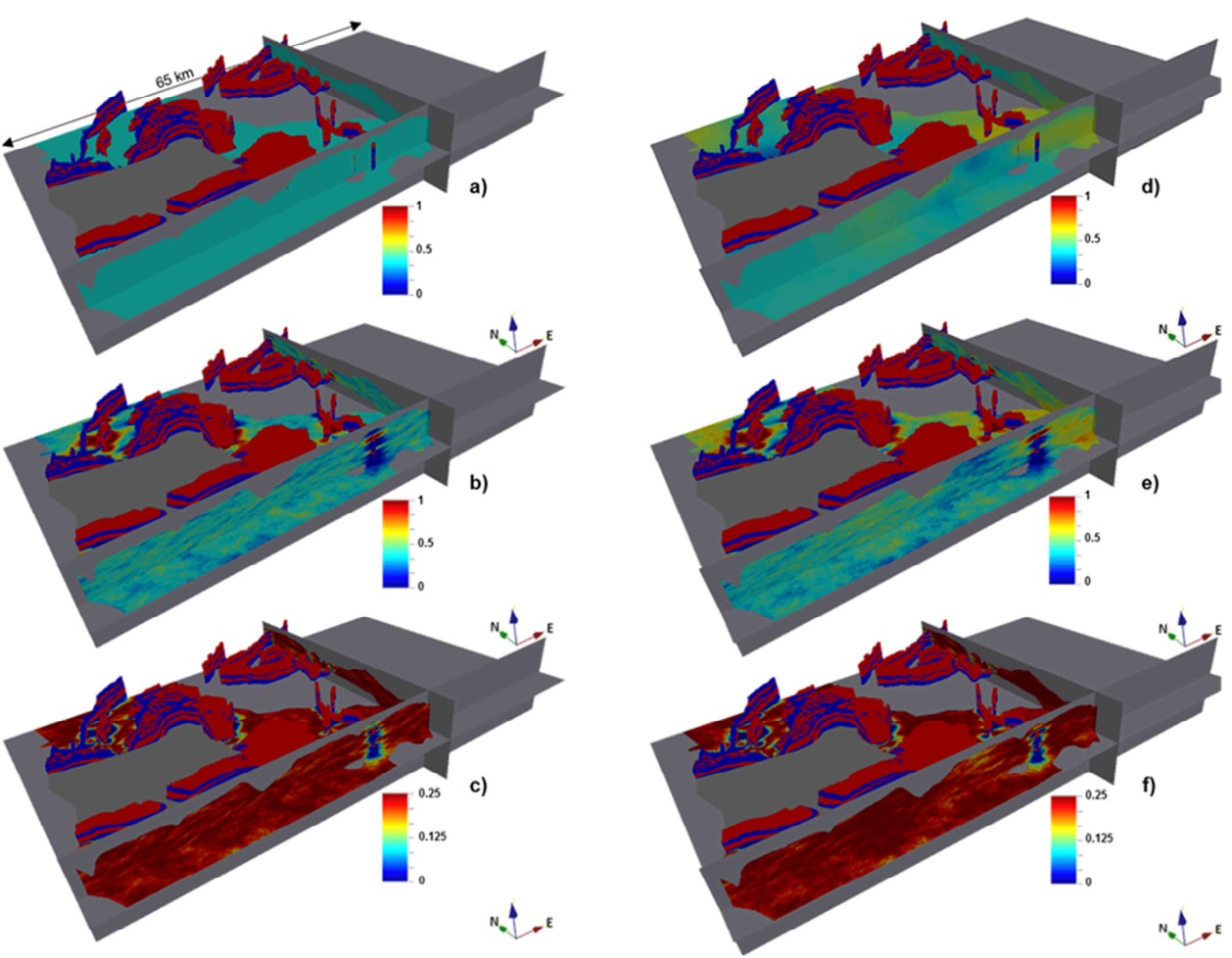

**Figure 23: Comparison of the conditioning approaches (c) and (d) of Table 1 in terms of: soft and hard conditioning - panels (a) and (d); e-type map - panels (b) and (e); variance map - panels (c) and (f). To facilitate the comparison, the probabilities in panels (a) and (d) are presented in a different colour scale with respect to Fig.s 8-9. The e-type and variance maps are based on 100 realizations. In all panels, the interpretation of the seismic data, and the buffer zone around the Tønder model are explicitly shown in terms of sand and clay (the red homogeneous volumes represent the sand bodies, the blue volumes show the clay lenses). Vertical exaggeration = 20x.**

| Strategy | Training image | Soft data | Hard data |
|---|---|---|---|
| a | 2$^{nd}$ (Fig. 10b) | ÷ | ÷ |
| b | 2$^{nd}$ (Fig. 10b) | ÷ | The Tønder model<br>Seismic interpretation<br>Boreholes |
| c | 2$^{nd}$ (Fig. 10b) | Sand probability directly from boreholes | The Tønder model<br>Seismic interpretation |
| d | 2$^{nd}$ (Fig. 10b) | 3D kriged grid of the sand probability distribution from boreholes | The Tønder model<br>Seismic interpretation |

**Table 1: The different conditioning strategies tested in this study. The corresponding realizations are presented in the Figs. 15 - 17.**