# Peer review of "Multiple-point statistical simulation for hydrogeological models: 3D training image development and conditioning strategies"

_Hydrology and Earth System Sciences, 2016_

## Referee Comment (RC1) · Anonymous Referee #1 · 9 Dec 2016

This manuscript is a well-written and clear case study on the application of MPS to a very large domain. As such, it will be valuable for a range of researchers. While I recommend eventual publication, I also have reservations that should be addressed.

Regarding the content of the study, I appreciate the overall methodology and the emphasis, throughout the discussions, on the fact that the training image and the simulation algorithm are all elements structuring the final models, and as such the evaluation should take place on unconditional realizations.

However, I also found that the conclusions would be much better supported by adding a few elements:

1) Currently only a single realization is used for each setting. This is clearly insufficient. On top of p. 12 it is argued that the simulation is considered representative, however I don't agree with this statement. Multiple realizations are needed to quantify uncertainty. It is possible that the single realization is representative, but the only way to find out is to compare with a set of other realizations and decide whether the inter-realization variability is small enough, according to a given criterion (e.g. flow, transport, etc). On p.12 (l.11-12) it is also argued that the methods to use multiple realizations do not exist, which is clearly not the case.

2) The assessment of the results is mostly qualitative, both regarding the patterns produced in the model, and to assess the proportions variability (top of p.8, top of p.9). The tools to do exist and should be used. Also, quantitative comparison of the modeled patterns and the patterns in the conditioning data would be a good validation.

3) The literature review part of the introduction is quite incomplete, missing a number of studies that have looked into 3D MPS models. On p.2 l. 25 it is said that not many studies have looked at 3D TI-based models. I disagree, with for example Ronayne et al (2008), Jha et al (2014), Perez et al (2014) to name a few, and a lot of other studies in reservoir engineering as well. For non-stationarity also, there are Cuhgunova et al (2008), Straubhaar et al (2011), and possibly others, who made important contributions.

Regarding the structure of the document, I also have 2 remarks:

1) There is an imbalance between the description of the data and methods, which is quite short (6 pages), and the discussion/conclusion, which is 5 pages. There is clearly too much material in the discussion, including elements that could be removed or moved to other sections. Here are some suggestions:

- P.7, l.15-20: this could go in section 5.4.

- P.9, from l.19: this could go in the introduction

- P.10, l.22 to p.11, l. 2: This is not related to the purpose of the paper and could be removed.

- P.11, l.3-10: The method for the conversion of boreholes to probabilities should be described in the methodology section, not here.

- P.11, l.11-19: There could be a separate section on non-stationarity because it is mentioned often.

- P.11, l.20-34: This is a long paragraph on something that is not done. It could be removed.

2) Sections 2, 3 and 4 could be grouped together as they all relate to the description of the study site.

Other remark:

Figures 7 and 8: the green-purple color scale is very subjective and seems to highlight values around 0.4. It creates artificial discontinuities. A usual continuous color scale (rainbow or grayscale) would be better.

---

## Referee Comment (RC2) · Anonymous Referee #2 · 21 Dec 2016

This paper presents a case study of multiple-point statistical simulation of sand/clay occurrence. The paper focuses on two aspects of multiple-point statistics: (1) 3D training image development and (2) different conditioning strategies to incorporate borehole data and geophysical data. This is a very relevant topic. Especially the construction of 3D training images is indeed still difficult. There are definitely some very interesting ideas in this paper, such as the different ways of using borehole data as hard or soft conditioning data. I would like to see, however, some more discussion on the following points:

In the title and aims of the paper, the authors stress the importance of realistic 3D training images. They write that they present a workflow to build a training image. The part

about how they build the training image (section 5.4) is however very short. From this short description, it is not clear to me how exactly the training image was constructed. On what data is the TI based? On seismics or on the existing Miocene model? How were the shapes/geometry/position of the clay/sand featureds determined? What was the input in the Geoscene3D model? How exactly does this model work? What are the assumptions? Is this a manual or an automatic method? What are the "interpretation points" and where do they come from? They refer to a methodology in an earlier paper, but this paper uses completely different input data, i.e., airborne EM data?

On page 7, the authors write that "the results are evaluated and compared against the structures expected from the Miocene model"? How exactly? Was this model not already used as a basis for the TI? Is it then fair to use it again to evaluate the resulting structures?

The simulated structures are relatively simple and uniform. Is it really necessary to use MPS? If you would model sand/clay occurrence with a more simple method such as indicator kriging, you would probably get similar results? It would strengthen the paper if you can prove somehow that this relatively complex approach has significant advantages over simpler approaches.

More in general, the conclusions of the paper are only based on visual inspection of the simulated clay/sand patterns. There is not objective or quantitative way of comparing the different results. For example in Figure 11: could you not use cross-validation or something similar to come to a more objective comparison of the different realizations? I also wonder how relevant the differences between the different realizations are, e.g. when you state that "the realizations showed a significant sensitivity to the TI". If you put these different realizations in a groundwater flow model, it is quite plausible that they all give similar results. It would be really interesting to using your geological model for some flow runs to see whether the different realizations based on different conditioning strategies really result in different groundwater flow patterns.

The authors claim that in the study area many of the borehole records are of low quality. Why is that? In what sense are they of low quality? How can users in other study areas determine the quality of boreholes and decide whether they can treat the borehole data as hard or soft data?

The results of the paper are based on visual comparisons of individual realizations using different conditioning strategies. It would be really interesting to see multiple realizations for each conditioning strategy to see the variability and uncertainty of different realizations.

On page 9, the authors write that realization rarely have the same spatial variability as the training image. I find this a really strange statement. The idea of MPS is that you produce realizations with similar spatial patterns as the TI. If the realizations do not show similar patterns, this usually means that the parameters have not been chosen optimally or that the TI was for example not large enough. They also claim that therefore a TI should be chosen together with a specific MPS algorithm? If find this really strange. In my experience, all MPS algorithms can produce realizations with similar patterns to the TI given that they are used in the right way. If the authors really want to claim that different MPS algorithms have different capabilities in reproducing patterns, they should show a comparison of the different algorithms.

On page 12, the authors write that "probabilistic models need to be developed and refined in order to utilize the multiple realizations and the uncertainty the represent". There are however many methods available and applications of MPS using multiple realizations to assess the uncertainty. The methods to do this are available but the authors have chosen to work with single realizations.

The paper is clear and well written. The figures are of good quality.

P4, line 24-25: "along extended profile": error in grammar of sentence?

---

## Referee Comment (RC3) · Anonymous Referee #3 · 27 Dec 2016

Multiple-point simulation (MPS) is a geostatistical simulation technique first developed at Stanford University in the early 2000's. An MPS algorithm is used to reproduce spatial patterns, such as connectivity, that are depicted in a training image (TI), which contains the possible spatial configurations for any given geological object and relationships between objects. A TI contains only spatial patterns and their respective likelihoods. A frequent pattern appears more often in the TI than a rare one. The actual position of a pattern in the TI is largely irrelevant, the MPS algorithm sees a set of patterns and tries to set them together through a randomization process. In order for an MPS simulation to produce a reasonable representation of any given geological system, it must have to honor some conditioning data and have a method of accounting

for spatial trends in the probabilities of selecting patterns from the TI.

The title of this paper suggests that, in this case, these MPS simulations are to guide subsequent hydrogeological applications. The title also suggests the main thrust of the paper is the importance of developing realistic 3-D TI's and strategies for conditioning MPS models. However, this topic is only discussed rather briefly (in 10 lines) in section 5.4 on page 7!

I thus found the paper rather confusing and yet I recognize that the authors and their organizations have considerable experience in 3-D geological modeling for hydrogeological applications, and that there is a considerable body of observational data in southern Denmark that can support the development of multiple-point simulations of the subsurface environment.

In short, while individual sentences and paragraphs use consistently good English, I became uncertain about many important details of their research and objectives. The paper is quite lengthy in its current form, yet it leaves many questions unanswered. If fact, I believe the paper raises more questions than it answers.

I therefore propose that, for publication, the authors undertake to reorganize the current text into something similar to the following:

Section 1: Introduction: This should clearly define the background, objectives, a scope of this project. These topics, I believe, include: • A desire to evaluate the Miocene sediments over a 2810 sq.km. area of southern Denmark where they provide the source of most drinking water. • For about 22% of the area, there is a detailed 3-D stratigraphic model (lithostratigaphic and/or hydrostratigraphic?) developed by deterministic methods (the Tonder model) • Southern Denmark has some high-resolution seismic surveys that can be used as conditioning data for MPS simulations (However, the authors need to provide more information about the spatial adequacy of these surveys, not just that they total 170 km and are shown (rather poorly) on Figure1). • While existing borehole records are available, they are of relatively low quality (WHY?)

and most borehole are relatively shallow, so these can provide only limited-value conditioning data.  ć The project was undertaken to determine if MPS could produce 3-D subsurface information over the entire southern Denmark area more efficiently than deterministic modeling, yet still produce information of value to further hydrogeological models.

Section 2: The Study Area and Available Data Sources: This should summarize its geological character and assess the various data sources. This can be accomplished by revising as necessary existing sections 2 and 3 and Figures 1-5. A discussion about trends should be enhanced.

Section 3: The Experimental Process. This needs considerable expansion from existing section 4. Several questions arise from reading the existing paper. Chief among them: Was the Tonder model the source used to develop the TI? Currently this is unclear. Later the use of two TI's is noted. How were they developed/selected? What are the spatial characteristic patterns desired in the TI?

Section 4: Analysis of Results. This will combine information from existing sections 5 and 6 and some of 7. It also should address several of the limitations defined below.

Section 5: Discussion and Conclusions. This should be relatively short, but include some of the ideas in existing sections 7 and 8. It also should address the need to determine what level of subsurface detail is required to produce an acceptable groundwater management tool for regional and more site-specific applications in southern Denmark (see my final comments).

As I reviewed the current draft, I assembled a list of what I consider to be its current limitations. These include:

1) Apparently, the research so far has focused on examining the role of various hard and soft conditioning data, but only a single realization is used for each setting. This is clearly insufficient. Repeated applications of MPS will produce a sequence of slightly

different realizations even with the same conditioning setup, and it should be possible to quantitatively evaluate their similarities and compare this to the differences introduced by the changed conditioning strategies.

2) Regarding the Training Image aspect of MPS, it seems that two TI's were used. It is unclear how they were developed and what underlying geological concepts or knowledge were used to develop them. Figure 9 does not clearly show the sand/clay layers – the colors are not ideal for this, and the text (line 15 Page 7) merely states one realization has more layers than the other. Does either seem more likely with the geological knowledge available? Are these layers defined as channels or sheets (continuous layers)? How does either TI relate to conditions within the Tonder model?

3) The assessment of the results is mostly qualitative. Quantitative tools to do exist and should be used.

4) The cited literature appears to miss several important more recent studies. Attached are a few representative paper citations.

FINAL COMMENTS

MPS is an interesting and potentially powerful method for developing very useful subsurface geological models. I am aware that at least some of the authors have experimented with other simulation approaches, such as TPROGS to apparently successfully simulate facies heterogeneity in buried valleys. It would be interesting for them to include at least a short comparison between MPS and other simulation approaches. The current paper assumes the reader to be proficient in MPS concepts. This may not be true in many cases, so a short comparison in the introduction might broaden the readership and understanding of the importance of this line of investigation.

I believe the ultimate goal of this research is not to model the Miocene of southern Denmark as a purely academic exercise, but to use this information to guide groundwater management schemes. I wonder what groundwater model sensitivity analysis would

yield in terms of necessary subsurface detail for producing acceptable groundwater resource management at regional or site-specific scales?

Obviously, somewhat less precise spatial definitions are likely to be required for regional assessments. On the other hand, site-specific studies may be unreliable if based only on MPS inputs without careful additional local conditioning. So, the question arises, "Where does MPS fit within this overall objective?" This question also reflects on the limitation noted on page 9 (lines 13-15) the present inability of MPA to handle graben structures and faults.

Despite my several criticisms, I think this is a potential important paper and hope the authors will consider reorganizing it and adding in a few details on some important methodology issues, while at the same time focusing on the TI and conditioning strategies.

SOME SUGGESTED REFERENCES:

Boucher, A. (2011) Strategies for Modeling with Multiple-point Simulation Algorithms. IN: "Closing the Gap", 2011 Gussow Geoscience Conference, Banff, Alberta. 9p.

Kessler, T. (2012) Hydrogeological Characterization of Low-permeability Clayey Tills – the Role of Sand Lenses. PhD Thesis, Department of Environmental Engineering, Technical University of Denmark, Lyngby, Denmark. 80p.

Klenner, R., Braunberger, J.R., Dotzenrod, N.W., Bosshart, N.W., Peck, W.D. & C.D. Gorecki (2014) Training Image Characterization and Multipoint Statistical Modeling of Clastic and Carbonate Formations. PowerPoint Presentation, 2014 Rocky Mountain Section AAPG Annual Meeting, Denver, Colorado. 27 slides.

Meerschman, E., Pirot, G., Mariethoz, G., Straubhaar, J., Van Meirvenne, M. & P. Renard (2013) A Practical Guide to Performing Multiple-Point Statistical Simulations with the Direct Sampling Algorithm. Computers & Geosciences. 52. p. 307-324.

Straubhaar, J., Walgenwitz, A. & P. Renard (2013) Parallel Multiple-Point Statistics

Algorithm Based on List and Tree Structures. Mathematical Geosciences. 45. p. 131-147.

Straubhaar, J., Renard, P. & G. Mariethoz (2016) Conditioning multiple-point statistics simulations to block data. Spatial Statistics. 16. p. 53-71.
* * *

---

## Referee Comment (RC4) · X. He (Referee) · 28 Dec 2016

The study aims to establish a workflow to carry out Multi-Point Statistics modeling for a testing area in Denmark using alternative 3D training images and various conditioning strategies. The research topic is of high relevance to those who work with hydrogeological modeling. The manuscript is well written with accurate language, rational methodology and convincing results. I recommend the manuscript is accepted for publication with minor revision. However, there are several details to be considered which are listed as follows:

As stated in the abstract, the introduction and the conclusion sections, one of the most important steps of the workflow is to develop 3D TIs in an iterative way. However, this

part is only briefly mentioned in the method section and not at all mentioned in the result section. I am curious about how the TIs are evolved gradually with feedback information after each step of adjustment, namely from the initial TI to the final TI. Additionally, is Fig 9 showing the initial or the finally TI?

In the introduction section, I would suggest to add a few sentences indicating the main objectives of the study.

Lin 210-211. The moving window for calculating the borehole uncertainty in the vertical direction is 20 m. Meanwhile, in Fig 7(b), the thickness of the Miocene layer is about 150 m, which in principle corresponds to 7 to 8 intervals in each borehole at maximum. However, as far as I can count, there are usually more than 7 color blocks in each borehole. Am I mistaken for something?

Fig 7, when interpolating the borehole uncertainty, are the borehole data outside the model domain being considered, both horizontally and vertically? If not, would there be extrapolation instead of interpolation towards the edges of the model domain?

In the results section, L259-266, there are two TIs being tested, one clearly has more layers than the other. Do these two TIs have any relation to the iterative approach the authors try to present in the study? Or is it a separate issue here? Moreover, it says the second TI is chosen because it is closer to what has been presented in Kristensen et al., 2015. So maybe it is better to describe very briefly what is in the Kristensen's study, and why that one is used as benchmark.

---

## Author Comment (AC1) · 23 Feb 2017

**Response to Reviewer #1**

We are thankful to Reviewer #1 for his/her valuable comments and suggestions, which will certainly improve the manuscript. The response to the individual comments is given below. The original review is quoted in *italics*, whereas our response is given in **bold** font.

5

*This manuscript is a well-written and clear case study on the application of MPS to a very large domain. As such, it will be valuable for a range of researchers. While I recommend eventual publication, I also have reservations that should be addressed.*

*Regarding the content of the study, I appreciate the overall methodology and the emphasis, throughout the discussions, on*
10 *the fact that the training image and the simulation algorithm are all elements structuring the final models, and as such the evaluation should take place on unconditional realizations.*

*However, I also found that the conclusions would be much better supported by adding a few elements:*

*1) Currently only a single realization is used for each setting. This is clearly insufficient. On top of p. 12 it is argued that the*
15 *simulation is considered representative, however I don't agree with this statement. Multiple realizations are needed to quantify uncertainty. It is possible that the single realization is representative, but the only way to find out is to compare with a set of other realizations and decide whether the inter-realization variability is small enough, according to a given criterion (e.g. flow, transport, etc). On p.12 (l.11-12) it is also argued that the methods to use multiple realizations do not exist, which is clearly not the case.*

20

**Concerning the representativeness of the realization discussed in our manuscript, it is possible to state that, by definition, each individual realization is representative. In fact, every realization is by construction compatible with all the input information (i.e., the statistics formalized by the training image, the hard data, and the soft conditioning).**

25

**Concerning the uncertainty assessment, even if it would be definitely very interesting in principle, we feel it would be out of the scope of the present research that is solely dealing with the development of the optimal strategies for conditioning the simulation and for preparing effective training images.**

30 *2) The assessment of the results is mostly qualitative, both regarding the patterns produced in the model, and to assess the proportions variability (top of p.8, top of p.9). The tools to do exist and should be used. Also, quantitative comparison of the modeled patterns and the patterns in the conditioning data would be a good validation.*

**With respect to a quantitative analysis of the proportions variability, we think that our point (i.e.: the kriged sand probability is effective in enforcing the proper spatial trend) is clearly shown by the comparison between Fig 14b and Fig. 8. The two figures allow a voxel-by-voxel comparison between the soft conditioning distribution and the final corresponding realization. So, it is not clear to us what kind of more quantitative argument we should use. On the other hand, we agree that we should make this comparison more evident in the text and, in the revised version of the manuscript, we will add few lines on this respect.**

**Regarding the quantitative assessment of the produced patterns, we believe that the comparison of Fig.s 10a and 10b, showing the unconstrained realizations associated with the training images TI1 and TI2 (in Fig.s 9a and 9b respectively), demonstrates quite well the large effects on the final realizations caused by relatively small perturbations in the used training image. However, even if the difference between the patterns in Fig.s 10a and 10b is quite evident, if requested, we could specify, in the revised version of manuscript, the value of the mean volume of the sand bodies in the realization generated with TI1 (Fig. 10a) and the corresponding value for the realization associated with TI2 (Fig. 10b).**

*3) The literature review part of the introduction is quite incomplete, missing a number of studies that have looked into 3D MPS models. On p.2 l. 25 it is said that not many studies have looked at 3D TI-based models. I disagree, with for example Ronayne et al (2008), Jha et al (2014), Perez et al (2014) to name a few, and a lot of other studies in reservoir engineering as well. For non-stationarity also, there are Cuhgunova et al (2008), Straubhaar et al (2011), and possibly others, who made important contributions.*

**We acknowledge the relevance of some of the suggested studies. In the revised version of the manuscript, we will include additional references.**

*Regarding the structure of the document, I also have 2 remarks:*
*1) There is an imbalance between the description of the data and methods, which is quite short (6 pages), and the discussion/conclusion, which is 5 pages. There is clearly too much material in the discussion, including elements that could be removed or moved to other sections. Here are some suggestions:*
*- P.7, l.15-20: this could go in section 5.4.*

**The rationale behind our original choice is that, in the initial part, we wish to simply present the different inputs and how we used/prepare them.**
**In the second part (from paragraph "6. Results"), we show the effects of the choices we made and the reasons for these choices. We do this by means of a detailed discussion of the corresponding results.**

So, even if it might seem unbalanced in terms of length of the different parts, we believe that, in this way, the paper is more effectively organized from a logical point of view, with a clear separation between the inputs (and their preparation) and the outputs (and the associated assumptions/choices).

*- P.9, from l.19: this could go in the introduction*

We definitely see the Reviewer's point.

Even if we feel that a few words spent on briefly (re)introducing the concepts can make the paper more easily readable, in the new version of the manuscript, we will follow the suggestion.

*- P.10, l.22 to p.11, l. 2: This is not related to the purpose of the paper and could be removed.*

Also in this case, we understand the Reviewer's point.

In the first place, we decided to add this part to reinforce the discussion on the possible use of seismics via the comparison between the characteristics of the seismic lines collected in this area with those acquired somewhere else for hydrostratigraphic studies.

*- P.11, l.3-10: The method for the conversion of boreholes to probabilities should be described in the methodology section, not here.*

The conversion of the borehole into probability is indeed discussed in the "5.3 Borehole data" section, where we describe the methodology we use to prepare the inputs. At page 11, in the section "7 Discussion", we simply recall that strategy to mention possible, straightforward extensions of the presented approach, for example, to the case where boreholes have varying quality.

*- P.11, l.11-19: There could be a separate section on non-stationarity because it is mentioned often.*

Non-stationarity is tackled by kriging the probability derived from the boreholes. For this reason, we think it is more logical to keep this aspect tightly connected (across the entire paper) with the discussion about the borehole data and the sand probability spatial trend.

*- P.11, l.20-34: This is a long paragraph on something that is not done. It could be removed.*

**Actually, we believe that a discussion on why we made the choice of not to do/show something could be relevant for the community and can contribute to the overall clarity and usefulness of the manuscript.**

*2) Sections 2, 3 and 4 could be grouped together as they all relate to the description of the study site.*

5

**We see the Reviewer's point.**

**In the original manuscript, we decided to keep these sections separate in order to maintain:**

**i) the prior overall geological understanding of the entire area,**

**ii) the observed and utilized data (seismics and boreholes), and**

10   **iii) the description of the specific geological unit to be investigated (the Miocene)**

**well distinct for expositive clarity and logical sequentiality.**

*Other remark:*

*Figures 7 and 8: the green-purple color scale is very subjective and seems to highlight values around 0.4. It creates artificial*

15   *discontinuities. A usual continuous color scale (rainbow or grayscale) would be better.*

**This specific color scale has been selected because 40% is the target marginal distribution value (as it has been derived from the boreholes and as it is consistently formalized in the training images). The details for this choice are described in the methodological section "5.3 Borehole data" where we discuss our approach for dealing with borehole**

20   **uncertainty and for translating the lithological information into probability. For these reasons, we believe that the adopted color scale is not subjective and the value 0.4 has a specific meaning as it corresponds to "no information" about the occurrence of sand or clay.**

---

## Author Comment (AC2) · 23 Feb 2017

**Response to Reviewer #2 comments**

We thank Reviewer #2 for taking the time to review this paper. We sincerely appreciate his/her insightful and constructive comments and suggestions. The response to the individual comments is given below. The original review is quoted in *italics*, whereas our response is given in **bold** font.

*This paper presents a case study of multiple-point statistical simulation of sand/clay occurrence. The paper focuses on two aspects of multiple-point statistics: (1) 3D training image development and (2) different conditioning strategies to incorporate borehole data and geophysical data. This is a very relevant topic. Especially the construction of 3D training images is indeed still difficult. There are definitely some very interesting ideas in this paper, such as the different ways of*

*using borehole data as hard or soft conditioning data. I would like to see, however, some more discussion on the following points:*

*In the title and aims of the paper, the authors stress the importance of realistic 3D training images. They write that they present a workflow to build a training image. The part about how they build the training image (section 5.4) is however very*

*short. From this short description, it is not clear to me how exactly the training image was constructed. On what data is the TI based? On seismics or on the existing Miocene model? How were the shapes/geometry/position of the clay/sand featureds determined? What was the input in the Geoscene3D model? How exactly does this model work? What are the assumptions? Is this a manual or an automatic method? What are the "interpretation points" and where do they come from? They refer to a methodology in an earlier paper, but this paper uses completely different input data, i.e., airborne EM data?*

*On page 7, the authors write that "the results are evaluated and compared against the structures expected from the Miocene model"? How exactly? Was this model not already used as a basis for the TI? Is it then fair to use it again to evaluate the resulting structures?*

**In the manuscript, the TI characteristics are described in "5.4 Training Image", while the details of the strategy we have used during the TI development are provided at the beginning of the section "6. Results", and discussed thorough Fig.s 9 and 10. Along the paper (e.g., at: page 7, lines 15-22, page 9, lines 19-32, and page 10, lines 1-8), we emphasise the importance of unconditional realizations to assess the quality of the selected TI (actually, in combination with the used simulation algorithm). As it is shown by the comparison in Fig. 10, even small**

**perturbations in the TIs (Fig. 9) affect significantly the associated realizations. After reading the Reviewer's comments, we feel that it would be probably more logical to move some of the material from page 7, lines 15-22 to the methodological section "5.4 Training Image". We will do this in the revised version of the manuscript. Moreover, we will add further details to the description of the TI construction: The first TI test was based on the existing 3D geological model (the Tønder model) covering an adjacent area. This first attempt was then manually adjusted**

during several iterations based on the unconditional output (in the paper, for simplicity, only the initial and final TIs are shown). This iterative process stopped when the corresponding unconstrained realization was found satisfactory in terms of its ability to mimic the geological features we expect in the Miocene, across the study area. Those expectations about the geology are based on our prior geological understanding of the area, the available seismic lines, and the few existing deep boreholes. The entire procedure is manual (except, of course, the unconstrained simulation).

Regarding the "interpretation points", they relate to the interpretation of the geophysical data and the consequent construction of the associated (manual) 3D geological models (as described in Kristensen et al., 2015 and, shortly, in Fig. 5 in the manuscript). In the present workflow, in particular, they are used, within Geoscene3D, to efficiently modify the TIs. In fact, the interpretation points define the surfaces delineating the volumes to be populated with, for example, sand/clay voxels. By changing their locations and creating/deleting some of them, we could have a full control over the manual adjustments of the TIs.

Jørgensen et al., 2013 is mentioned simply as an example of research performed by using the voxel modelling tools available in Geoscene3D and specifically designed for (manual) 3D geological modelling.

*The simulated structures are relatively simple and uniform. Is it really necessary to use MPS? If you would model sand/clay occurrence with a more simple method such as indicator kriging, you would probably get similar results? It would strengthen the paper if you can prove somehow that this relatively complex approach has significant advantages over simpler approaches.*

We showed that even small details, which seem to be "irrelevant" in the TIs (that are capturing the available statistical information regarding the object to simulate), have actually significant impacts on the results (Fig.s 9 and 10).

Moreover, this is not meant to be a paper about the comparison of the performances of MPS as a tool to incorporate complex statistical information against other "simpler" approaches. This manuscript deals with the optimal strategies, within the MPS framework, to include the available data into the simulation as soft/hard conditioning and to build the most (geologically) effective TIs.

As a matter of fact, however, we believe that MPS' strength lies also in being quite intuitive: for many users, it is difficult to apply (and deeply understand), for example, indicator kriging and properly choose variogram models, while MPS allows the geologist to provide complex information in form of TIs, which is clearly a more intuitive (and, at same time, generally, a more effective) approach.

*More in general, the conclusions of the paper are only based on visual inspection of the simulated clay/sand patterns. There is not objective or quantitative way of comparing the different results. For example in Figure 11: could you not use cross-validation or something similar to come to a more objective comparison of the different realizations? I also wonder how relevant the differences between the different realizations are, e.g. when you state that "the realizations showed a significant*

*sensitivity to the TI". If you put these different realizations in a groundwater flow model, it is quite plausible that they all give similar results. It would be really interesting to using your geological model for some flow runs to see whether the different realizations based on different conditioning strategies really result in different groundwater flow patterns.*

**We agree that, in principle, it would be very interesting to investigate the results of flow models based on the different**
**realizations. However, we feel that this would be out of the scope of the paper. Adding this kind of considerations would probably make the discussion lengthy and distract the reader from the main focuses of the paper, that are: 1) how to include, in the most effective way, the available data as simulation conditioning, and 2) how to build a TI that is capable to reproduce the geological features we want to see in the realizations. In particular, regarding the latter point, the ability to generate meaningful geological realizations should be seen in a general perspective and, so,**
**relevant per se. Hence, even if, in this specific case, the differences between the two realizations in Fig 10 might be not "significant" for a flow model, the geological differences are evident, and, in principle, could impact the by-products generated by using the realizations. This paper is about the optimal practices to prepare the best possible stochastic geological inputs for subsequent applications.**

**Concerning quantitative analyses of the different realizations across the paper, maybe it would make clearer our point to provide, for the two realizations in Fig. 10, some measurements of, for example, the mean of the volume of the sand units. If this is found relevant, we will add it to the revised version of the paper.**
**In general, with respect to a quantitative analysis of the proportions variability of the different realizations discussed across the paper, we think that our point (i.e.: the kriged sand probability is effective in enforcing the proper spatial**
**trend) is clearly demonstrated by the comparison between Fig 14b and Fig. 8. The two figures allow a voxel-by-voxel comparison between the soft conditioning distribution (derived from the boreholes) and the final corresponding realization. So, sincerely, we do not know which more quantitative argument we should use.**
**On the other hand, we agree that, in the text, we should emphasise more explicitly the relevance of this comparison.**

*The authors claim that in the study area many of the borehole records are of low quality. Why is that? In what sense are they of low quality? How can users in other study areas determine the quality of boreholes and decide whether they can treat the borehole data as hard or soft data?*

**Boreholes, like any other kinds of data, are affected by noise. The level of noise determines the quality/reliability of the measurements (i.e., the quality/reliability of the boreholes). Many factors impact the quality of boreholes. Just to mention a few of them: (i) the drilling methods (e.g.: hydraulic-rotary drilling and air core drillings. In some cases, for example, the finer sediments can be flushed out and the driller could potentially misinterpret, as more sandy, a**
**clay layer); (ii) the drilling purpose (sometimes, if the goal is to reach a specific target, the lithological description can be quite poor since it is not a priority); (iii) the age of the boreholes (nowadays, in Denmark, samples are collected systematically every meter. For sure, this was not the case few years ago); (iv) the presence of simultaneous wireline logging data (these kinds of ancillary information make the geological interpretation definitely more certain). During the geological modelling phase, a skilled geologist goes through all the borehole records and verifies all these different**
**pieces of information and check for inconsistencies.**

**In the paper, we mentioned that, for example, we use the seismic data as hard data because they were considered highly reliable and the scale of the structures they were able to delineate was comparable to the scale of the simulated outputs. Clearly, this is not true for the boreholes.**

*The results of the paper are based on visual comparisons of individual realizations using different conditioning strategies. It would be really interesting to see multiple realizations for each conditioning strategy to see the variability and uncertainty of different realizations.*

**We agree with the reviewer that investigating the variability and uncertainty would be interesting. However, we**
**believe that this would be out of the scope of the present paper.**

*On page 9, the authors write that realization rarely have the same spatial variability as the training image. I find this a really strange statement. The idea of MPS is that you produce realizations with similar spatial patterns as the TI. If the realizations do not show similar patterns, this usually means that the parameters have not been chosen optimally or that the*
*TI was for example not large enough. They also claim that therefore a TI should be chosen together with a specific MPS algorithm? If find this really strange. In my experience, all MPS algorithms can produce realizations with similar patterns to the TI given that they are used in the right way. If the authors really want to claim that different MPS algorithms have different capabilities in reproducing patterns, they should show a comparison of the different algorithms.*

**The meaning of what we wrote at page 9 concerns the fact that, while, ideally, the effects of the TI should not depend on the algorithm choice, these is not true in practice. The different implementations, and the different parameters that can be tuned during the simulation, make the realizations algorithm-dependent. It would not make much sense to compare the realizations generated with different algorithms, not only because different algorithms have different**

parameters to be set, but also because the point of the paper is that the TI must be developed, no matter what, by studying the unconstrained realization and its accordance with our geological expectations.

Basically, we claim that the TI and the simulation algorithm are all interdependent elements structuring the final models, and, as such, the evaluation should take place a-posteriori on unconditional realizations.

*On page 12, the authors write that "probabilistic models need to be developed and refined in order to utilize the multiple realizations and the uncertainty the represent". There are however many methods available and applications of MPS using multiple realizations to assess the uncertainty. The methods to do this are available but the authors have chosen to work with single realizations.*

Writing that, we meant that applications and methods that are making use of the full potential (for example for geological modelling) of probabilistic models (with their large number of realizations and information regarding the uncertainty) are still under development. Those lines in the original manuscript were not about the possibility to assess the uncertainty via analyzing the realization ensemble.

The reasons why we decided not to investigate the uncertainty are largely discussed before in this document. And, accordingly to the future editor's suggestions, we can make these reasons more explicit in the next version of the paper.

*The paper is clear and well written. The figures are of good quality.*

*P4, line 24-25: "along extended profile": error in grammar of sentence?*

We thank once more the reviewer for the positive comments, especially because we invested a lot of time in the preparation of good quality and, hopefully, clear figures.

We believe that "along extended profile" reflects what we mean.

---

## Author Comment (AC3) · 23 Feb 2017

**Response to Referee #3 comments**

We thank Referee #3 for his/her valuable comments and suggestions. The response to the individual comments is given in the following. The original review is quoted in *italics*, whereas the response of the authors is given in **bold** font.

5    *Multiple-point simulation (MPS) is a geostatistical simulation technique first developed at Stanford University in the early 2000's. An MPS algorithm is used to reproduce spatial patterns, such as connectivity, that are depicted in a training image (TI), which contains the possible spatial configurations for any given geological object and relationships between objects. A TI contains only spatial patterns and their respective likelihoods. A frequent pattern appears more often in the TI than a rare one. The actual position of a pattern in the TI is largely irrelevant, the MPS algorithm sees a set of patterns and tries to set*

10    *them together through a randomization process. In order for an MPS simulation to produce a reasonable representation of any given geological system, it must have to honor some conditioning data and have a method of accounting for spatial trends in the probabilities of selecting patterns from the TI.*

*The title of this paper suggests that, in this case, these MPS simulations are to guide subsequent hydrogeological*

15    *applications. The title also suggests the main thrust of the paper is the importance of developing realistic 3-D TI's and strategies for conditioning MPS models. However, this topic is only discussed rather briefly (in 10 lines) in section 5.4 on page 7!*

**In the manuscript, the TI characteristics are described in "5.4 Training Image", while the details of the strategy we**

20    **have used during the TI development are provided at the beginning of the section "6. Results", and discussed thorough Fig.s 9 and 10. Along the paper (e.g., at page 7, lines 15-22; page 9, lines 19-32; and page 10, lines 1-8), we emphasise the importance of unconditional realizations to assess the quality of the selected TI (in combination with the used simulation algorithm).**

**Now, we feel that it would be probably better to move some of the material from page 7, lines 15-22, to the**

25    **methodological section "5.4 Training Image". We will do this in the revised version of the manuscript. Moreover, we will add further details to the description of the TI development.**

**Regarding the conditioning strategies, they are discussed throughout the entire manuscript.**

**In the first place, they are introduced in the methodological section "5 Defining MPS input information". For**

30    **example, sections "5.1 Seismic data" and "5.2 Existing 3D model" are devoted to the reasons for using the seismic data and the boundaries of the pre-existing geological model as hard conditioning, while section "5.3 Borehole data" describes how to translate boreholes into soft conditioning data by means of a moving window approach and discusses the importance of kriging the "localized" borehole information to enforce the necessary spatial trend.**

**The results of the application of the different conditioning strategies - introduced in the section "5 Defining MPS input information" - are then presented and compared in section "6 Results".**

**Section "7 Discussion" examines the assumptions and choices we made and the future possible developments of the presented conditioning strategies and TI development workflow.**

**Thus, we do not agree with the reviewer when s/he writes that we discussed "***realistic 3-D TI's and strategies for conditioning MPS models***" "***rather briefly (in 10 lines) in section 5.4 on page 7!***". We definitely spent a large portion of the manuscript to go through the optimal approaches for conditioning and TI construction.**

10 *I thus found the paper rather confusing and yet I recognize that the authors and their organizations have considerable experience in 3-D geological modeling for hydrogeological applications, and that there is a considerable body of observational data in southern Denmark that can support the development of multiple-point simulations of the subsurface environment.*

15 *In short, while individual sentences and paragraphs use consistently good English, I became uncertain about many important details of their research and objectives. The paper is quite lengthy in its current form, yet it leaves many questions unanswered. If fact, I believe the paper raises more questions than it answers.*

*I therefore propose that, for publication, the authors undertake to reorganize the current text into something similar to the*
20 *following:*

*Section 1: Introduction: This should clearly define the background, objectives, a scope of this project. These topics, I believe, include: • A desire to evaluate the Miocene sediments over a 2810 sq.km. area of southern Denmark where they provide the source of most drinking water. • For about 22% of the area, there is a detailed 3-D stratigraphic model*
25 *(lithostratigaphic and/or hydrostratigraphic?) developed by deterministic methods (the Tonder model) • Southern Denmark has some high-resolution seismic surveys that can be used as conditioning data for MPS simulations (However, the authors need to provide more information about the spatial adequacy of these surveys, not just that they total 170 km and are shown (rather poorly) on Figure1). • While existing borehole records are available, they are of relatively low quality (WHY?) and most borehole are relatively shallow, so these can provide only limited-value conditioning data. • The*
30 *project was undertaken to determine if MPS could produce 3-D subsurface information over the entire southern Denmark area more efficiently than deterministic modeling, yet still produce information of value to further hydrogeological models.*

*Section 2: The Study Area and Available Data Sources: This should summarize its geological character and assess the various data sources. This can be accomplished by revising as necessary existing sections 2 and 3 and Figures 1-5. A*
35 *discussion about trends should be enhanced.*

*Section 3: The Experimental Process. This needs considerable expansion from existing section 4. Several questions arise from reading the existing paper. Chief among them: Was the Tonder model the source used to develop the TI? Currently this is unclear. Later the use of two TI's is noted. How were they developed/selected? What are the spatial characteristic patterns desired in the TI?*

*Section 4: Analysis of Results. This will combine information from existing sections 5 and 6 and some of 7. It also should address several of the limitations defined below.*

*Section 5: Discussion and Conclusions. This should be relatively short, but include some of the ideas in existing sections 7 and 8. It also should address the need to determine what level of subsurface detail is required to produce an acceptable groundwater management tool for regional and more site-specific applications in southern Denmark (see my final comments).*

**We arranged the paper as follows:**

1) **"1 Introduction" - the first section after the abstract - is devoted to a very general and brief discussion of the existing literature on MPS.**

2) **The second section "2 Study area" deals with the overall geological framework of the area. So, this section delineates the general geological setting where the unit investigated in the paper (the Miocene) is embedded. This, at the same time, contextualizes the presented research and better defines its limits.**

3) **Section "3 Data" describes in detail the amount and characteristics of the different kinds of data available (borehole data and seismic measurements). In particular, a large part of the paragraph "3.2 High-resolution seismic data" discusses - to a reasonable level of detail for the purpose of the present study - the different specifications of the seismic data available.**

4) **Section "4 Establishing framework-model constraints" goes into details about the specific geological unit targeted in the research.**

5) **"5 Defining MPS input information" is the most methodological section of the paper. Here, the approaches used in the comparison throughout the manuscript are described. For example, it is discussed how (and why):**

   (i) **the seismic data and the existing manual adjacent model have been incorporated as hard conditioning (paragraphs "5.1 Seismic data" and "5.2 Existing 3D model");**

   (ii) **to translate the boreholes into soft probability to address their uncertainties (paragraph "5.3 Borehole data");**

   (iii) **to build an effective TI based on the outcomes of the unconstrained simulations ("5.4 Training Image").**

6) Section "6 Results" is about the detailed comparisons of the outputs resulting from the application of: (i) the iterative approach for the construction of the TI and (ii) the different conditioning strategies detailed in the previous section.

7) The second last section, "7 Discussion" discusses the assumptions/choices made across the paper and their possible limitations and possible, future, developments.

8) "8 Conclusion" is a very concise section where we simply summarize our results.

We trust this is a reasonable way to present our research.

Clearly, other scientists might think differently. To some extent, it is simply matter of taste, as long as the rationale behind the choice is evident. And we believe that already in the present form, the overall logical organization of the paper is quite clear and effective.

Regarding the specific questions posed by the reviewer:

1) Many factors affect the quality of boreholes: (i) the drilling methods (e.g.: hydraulic-rotary drilling and air core drillings); (ii) the drilling purpose (sometimes, if the goal is simply to reach a specific target, the lithological description is not a priority); (iii) the age of the boreholes (older boreholes are generally less reliable); (iv) the presence of simultaneous wireline logging data. During the geological modelling phase, all the borehole records are (or should be) checked for inconsistencies. In the study area, as stated in the manuscript, the few available deep borehole are characterized by high level of uncertainty.

2) We acknowledge that, even if the procedure for the construction of the TI is discussed in several parts along the paper (not only in the paragraph "5.4 Training Image", but also at: page 7, lines 15-22; page 9, lines 19-32; and page 10, lines 1-8), expanding the associated explanation could make our point clearer. And we will do that in the new version.

3) MPS has not been considered as a way to "*produce 3-D subsurface information over the entire southern Denmark area more efficiently than deterministic modeling*". With the available input data (in terms of types, quality, and amount), MPS is the only way to produce a meaningful estimation of the geological variability within the Miocene at the scale that is reasonable for groundwater investigations.

*As I reviewed the current draft, I assembled a list of what I consider to be its current limitations. These include:*

*1) Apparently, the research so far has focused on examining the role of various hard and soft conditioning data, but only a single realization is used for each setting. This is clearly insufficient. Repeated applications of MPS will produce a sequence of slightly different realizations even with the same conditioning setup, and it should be possible to quantitatively evaluate their similarities and compare this to the differences introduced by the changed conditioning strategies.*

Clearly, the differences we highlighted for each conditioning setup are not realization-dependent. This means that they will appear consistently for every realization obtained with the same conditioning settings. Just to mention an example, all the realizations generated with the "borehole as hard data" (Fig. 12b) will be "perfectly" matching all the borehole samples (page 8, lines 11-18) by construction. The same is true for the "borehole as soft data" (Fig. 12c); in this case, we confirmed empirically that the SNESIM ignores (Hansen et al., submitted) "localized" soft data (e.g.: page 8, lines 4-7).

Of course, considering more realizations would allow uncertainty assessment. And this would be, in principle, very interesting. However, as the reviewer has pointed out, the paper is already quite long and a discussion about the uncertainty would fall out of the original scope of the manuscript. This article is, in fact, meant to be simply about the best conditioning strategy and the optimal preparation of the TI.

*2) Regarding the Training Image aspect of MPS, it seems that two TI's were used. It is unclear how they were developed and what underlying geological concepts or knowledge were used to develop them. Figure 9 does not clearly show the sand/clay layers – the colors are not ideal for this, and the text (line 15 Page 7) merely states one realization has more layers than the other. Does either seem more likely with the geological knowledge available? Are these layers defined as channels or sheets (continuous layers)? How does either TI relate to conditions within the Tonder model?*

One of the main points this manuscript would like to convey is the importance of unconditional realizations to evaluate the effectiveness of the selected TI. In particular, we discuss this in "5.4 Training Image" (where the general TI characteristics are detailed) and in "6 Results" (concerning the strategy for the TI development).

However, for sake of clarity, it seems necessary to add, in the new manuscript, further details. So, in the revised version, we will underline that: (i) the first TI test was based on the existing 3D geological model (the Tønder model) covering an area adjacent to the simulated one; (ii) then, during several iterations, this first TI was manually adjusted based on the unconditional outputs. In the paper, for simplicity, only the initial and final TIs are shown in Fig. 9 together with the associated unconstrained realizations (Fig. 10); (iii) this iterative process was stopped as soon as the corresponding realization was able to mimic the geological features expected in the Miocene.

Those expectations about the geology are based: (i) on our prior geological understanding of the area, (ii) the available seismic lines, and (iii) the few existing deep boreholes.

*3) The assessment of the results is mostly qualitative. Quantitative tools to do exist and should be used.*

We do not agree with this remark. For example, with respect to a quantitative analysis of the proportions variability, we think that our point concerning the effectiveness of using the kriged sand probability to enforce the proper spatial trend is clearly demonstrated via the comparison between Fig 14b and Fig. 8. The two figures allow a voxel-by-voxel

comparison between the soft conditioning distribution and the final corresponding realization. So, it is not clear to us what kind of more quantitative argument we should use instead.

Maybe, regarding the quantitative assessment of the patterns produced by using the two different TIs in Fig. 9, it could be useful to report the values of the mean volume of the sand bodies in the realizations in Fig. 10 generated respectively with TI1 and TI2. So, even if the difference between the patterns in Fig.s 10a and 10b is evident, if requested, we will specify these values in the revised version of manuscript.

*4) The cited literature appears to miss several important more recent studies. Attached are a few representative paper citations.*

We see the reviewer's point. In the new version, we will include additional, relevant references.

*FINAL COMMENTS*

*MPS is an interesting and potentially powerful method for developing very useful subsurface geological models. I am aware that at least some of the authors have experimented with other simulation approaches, such as TPROGS to apparently successfully simulate facies heterogeneity in buried valleys. It would be interesting for them to include at least a short comparison between MPS and other simulation approaches. The current paper assumes the reader to be proficient in MPS concepts. This may not be true in many cases, so a short comparison in the introduction might broaden the readership and understanding of the importance of this line of investigation.*

We believe that the paper (already "*quite lengthy in its current form*") would not benefit from a comparison with other simulation approaches as the manuscript is not about checking the performances of MPS as a tool to incorporate complex statistical information against other "simpler" approaches. This article deals solely with the optimal strategies (within the MPS framework) for the soft/hard conditioning and for the construction of the most (geologically) effective TI.
In addition, the approach of the submitted article is based on TIs for the description of the information about spatial structures, while TPROGS does not make use of TIs and, instead, requires the user to define transition probabilities from which the realizations are then simulated. Therefore, given the topic of the paper, it is not natural to consider the use of TPROGS.

*I believe the ultimate goal of this research is not to model the Miocene of southern Denmark as a purely academic exercise, but to use this information to guide groundwater management schemes. I wonder what groundwater model sensitivity*

*analysis would yield in terms of necessary subsurface detail for producing acceptable groundwater resource management at regional or site-specific scales?*

*Obviously, somewhat less precise spatial definitions are likely to be required for regional assessments. On the other hand, site-specific studies may be unreliable if based only on MPS inputs without careful additional local conditioning. So, the question arises, "Where does MPS fit within this overall objective?" This question also reflects on the limitation noted on page 9 (lines 13-15) the present inability of MPA to handle graben structures and faults.*

**The present paper is about: (i) how to include, in the most effective way, the available data as simulation conditioning, and (ii) how to build a TI that is capable to reproduce the geological features we want to see in the realizations. In particular, regarding point (ii), the ability to generate meaningful geological realizations must be seen in a general perspective and, so, relevant per se.**

**Moreover, MPS is a way to "see" beyond the (strictly speaking) available data. In fact, via the prior geological knowledge formalized by the TI, it is possible to "reconstruct" features compatible, at the same time, with the data (e.g.: the geophysical measurements), but also with the desired prior geostatistical information. This is a very important aspect in the paper: a significant amount of data - but not as much as we desired (as always happens) - was available together with the knowledge of the geological features we wanted to see in the realizations. Thus, we investigate the best workflow to utilize the full potential and flexibility of MPS methodology to jointly exploit those two pieces of information and obtain a realization that is, simultaneously, matching the data (i.e., boreholes, geophysical measurements, and pre-existing geomodels), and having the required spatial characteristics.**

**The scale of the geofeatures necessary for flow modelling purposes is hard to decide before entering in the more hydrological aspects of the problem. In the paper, we show how to get those features in an effective way and we test the proposed approach at the scale we think is the more appropriate one. Further analysis would be definitely necessary, but we think they are not falling in the scope of the present study.**

*Despite my several criticisms, I think this is a potential important paper and hope the authors will consider reorganizing it and adding in a few details on some important methodology issues, while at the same time focusing on the TI and conditioning strategies.*

*SOME SUGGESTED REFERENCES:*

*Boucher, A. (2011) Strategies for Modeling with Multiple-point Simulation Algorithms.IN: "Closing the Gap", 2011 Gussow Geoscience Conference, Banff, Alberta. 9p.*

*Kessler, T. (2012) Hydrogeological Characterization of Low-permeability Clayey Tills – the Role of Sand Lenses. PhD Thesis, Department of Environmental Engineering, Technical University of Denmark, Lyngby, Denmark. 80p.*

*Klenner, R., Braunberger, J.R., Dotzenrod, N.W., Bosshart, N.W., Peck, W.D. & C.D. Gorecki (2014) Training Image Characterization and Multipoint Statistical Modeling of Clastic and Carbonate Formations. PowerPoint Presentation, 2014 Rocky Mountain Section AAPG Annual Meeting, Denver, Colorado. 27 slides.*

*Meerschman, E., Pirot, G., Mariethoz, G., Straubhaar, J., Van Meirvenne, M. & P. Renard (2013) A Practical Guide to Performing Multiple-Point Statistical Simulations with the Direct Sampling Algorithm. Computers & Geosciences. 52. p. 307-324.*

*Straubhaar, J., Walgenwitz, A. & P. Renard (2013) Parallel Multiple-Point Statistics Algorithm Based on List and Tree Structures. Mathematical Geosciences. 45. p. 131-147.*

*Straubhaar, J., Renard, P. & G. Mariethoz (2016) Conditioning multiple-point statistics simulations to block data. Spatial Statistics. 16. p. 53-71*

---

## Author Response (AR1)

**Response to Reviewer #1**

We are thankful to Reviewer #1 for his/her valuable comments and suggestions, which certainly improved the manuscript. The response to the individual comments is given below. The original review is quoted in *italics*, whereas our response is given in **bold** font.

*This manuscript is a well-written and clear case study on the application of MPS to a very large domain. As such, it will be valuable for a range of researchers. While I recommend eventual publication, I also have reservations that should be addressed.*

*Regarding the content of the study, I appreciate the overall methodology and the emphasis, throughout the discussions, on*

10 *the fact that the training image and the simulation algorithm are all elements structuring the final models, and as such the evaluation should take place on unconditional realizations.*

*However, I also found that the conclusions would be much better supported by adding a few elements:*

*1) Currently only a single realization is used for each setting. This is clearly insufficient. On top of p. 12 it is argued that the*

15 *simulation is considered representative, however I don't agree with this statement. Multiple realizations are needed to quantify uncertainty. It is possible that the single realization is representative, but the only way to find out is to compare with a set of other realizations and decide whether the inter-realization variability is small enough, according to a given criterion (e.g. flow, transport, etc). On p.12 (l.11-12) it is also argued that the methods to use multiple realizations do not exist, which is clearly not the case.*

**Concerning the representativeness of the realization discussed in our manuscript, it is possible to state that, by definition, each individual realization is representative. In fact, every realization is, by construction, compatible with all the input information (i.e., the statistics formalized by the training image, the hard data, and the soft conditioning). In the new version of the manuscript, we added a few lines stressing on this, very important, aspect.**

**Concerning the uncertainty assessment, even if it would be definitely very interesting in principle, we feel it would be out of the scope of the present research that is solely dealing with the development of the optimal strategies for conditioning the simulation and for preparing effective training images.**

30 *2) The assessment of the results is mostly qualitative, both regarding the patterns produced in the model, and to assess the proportions variability (top of p.8, top of p.9). The tools to do exist and should be used. Also, quantitative comparison of the modeled patterns and the patterns in the conditioning data would be a good validation.*

With respect to a quantitative analysis of the proportions variability, we think that our point (i.e.: the kriged sand probability is effective in enforcing the proper spatial trend) is clearly shown by the comparison between Fig. 17b (in the new version) and Fig. 8. In fact, the two figures allow a voxel-by-voxel comparison between the soft conditioning distribution and the final corresponding realization. However, in order to meet the reviewer's request, we added, in the new manuscript, Fig. 18 showing the cross-correlations between the soft probability distribution (Fig. 8) and each of the realizations (c) and (d) in Table 1 (and plotted, for example, in Fig. 14c and Fig. 14d).

Regarding the quantitative assessment of the produced patterns, we believe that the comparison of Fig.s 10a and 10b, showing the unconstrained realizations associated with the training images TI1 and TI2 (in Fig.s 9a and 9b respectively), demonstrates quite well the large effects on the final realizations caused by relatively small perturbations in the used training image. But, in order to make our point clearer, we included in the revised version, three more figures: Fig. 11 (showing the sizes of the connected sand bodies for, respectively, the unconstrained realization generated by TI1 and TI2); Fig. 12 (comparing the eccentricities of the unconstrained realization for TI1 and TI2); Fig. 13 (demonstrating that the connected bodies for the unconstrained realization associated with TI1 are more jagged than the others).

*3) The literature review part of the introduction is quite incomplete, missing a number of studies that have looked into 3D MPS models. On p.2 l. 25 it is said that not many studies have looked at 3D TI-based models. I disagree, with for example Ronayne et al (2008), Jha et al (2014), Perez et al (2014) to name a few, and a lot of other studies in reservoir engineering as well. For non-stationarity also, there are Cuhgunova et al (2008), Straubhaar et al (2011), and possibly others, who made important contributions.*

We acknowledge the relevance of the suggested studies. In the revised version of the manuscript, we included the additional references.

*Regarding the structure of the document, I also have 2 remarks:*
*1) There is an imbalance between the description of the data and methods, which is quite short (6 pages), and the discussion/conclusion, which is 5 pages. There is clearly too much material in the discussion, including elements that could be removed or moved to other sections. Here are some suggestions:*
*- P.7, l.15-20: this could go in section 5.4.*

The rationale behind our original choice is that, in the initial part, we wish to simply present the different inputs and how we used/prepare them.

**In the second part (from paragraph "6. Results"), we show the effects of the choices we made and the reasons for these choices. We do this by means of a detailed discussion of the corresponding results.**

**So, even if it might seem unbalanced in terms of length of the different parts, we believe that, in this way, the paper is more effectively organized from a logical point of view, with a clear separation between the inputs (and their preparation) and the outputs (and the associated assumptions/choices).**

*- P.9, from l.19: this could go in the introduction*

**The same concept is already in the "Introduction" and it is (re)discussed in the "Discussion" only for sake of clarity.**

*- P.10, l.22 to p.11, l. 2: This is not related to the purpose of the paper and could be removed.*

**Following one of the original suggestions of the Editor, we added this part to reinforce the discussion on the possible use of seismics via the comparison between the characteristics of the seismic lines collected in this area with those acquired somewhere else for hydrostratigraphic studies.**

*- P.11, l.3-10: The method for the conversion of boreholes to probabilities should be described in the methodology section, not here.*

**The conversion of the borehole into probability is indeed discussed in the "5.3 Borehole data" section, where we describe the methodology we use to prepare the inputs. In the section "7 Discussion", we simply recall that strategy to mention possible, straightforward extensions of the presented approach, for example, to the case where boreholes have varying quality.**

*- P.11, l.11-19: There could be a separate section on non-stationarity because it is mentioned often.*

**Non-stationarity is tackled by kriging the probability derived from the boreholes. For this reason, we think it is more logical to keep this aspect tightly connected (across the entire paper) with the discussion about the borehole data and the sand probability spatial trend.**

*- P.11, l.20-34: This is a long paragraph on something that is not done. It could be removed.*

**Actually, we believe that a discussion on why we made the choice of not to do/show something could be relevant for the community and can contribute to the overall clarity and usefulness of the manuscript.**

*2) Sections 2, 3 and 4 could be grouped together as they all relate to the description of the study site.*

**In the original and new manuscripts, for expositive clarity and logical sequentiality, we decided to keep these sections separate in order to maintain well distinct:**

**i) the prior overall geological understanding of the entire area,**

**ii) the observed and utilized data (seismics and boreholes), and**

**iii) the description of the specific geological unit to be investigated (the Miocene) .**

*Other remark:*

*Figures 7 and 8: the green-purple color scale is very subjective and seems to highlight values around 0.4. It creates artificial discontinuities. A usual continuous color scale (rainbow or grayscale) would be better.*

**This specific color scale has been selected because 40% is the target marginal distribution value (as it has been derived from the boreholes, and as it is consistently formalized in the training images). The details for this choice are described in the methodological section "5.3 Borehole data" where we discuss our approach for dealing with borehole uncertainty and for translating the lithological information into probability. For these reasons, we believe that the adopted color scale is not subjective and the value 0.4 has a specific meaning as it corresponds to "no information" about the occurrence of sand or clay.**

**Response to Reviewer #2 comments**

We thank Reviewer #2 for taking the time to review this paper. We sincerely appreciate his/her insightful and constructive comments and suggestions. The response to the individual comments is given below. The original review is quoted in *italics*, whereas our response is given in **bold** font.

*This paper presents a case study of multiple-point statistical simulation of sand/clay occurrence. The paper focuses on two aspects of multiple-point statistics: (1) 3D training image development and (2) different conditioning strategies to incorporate borehole data and geophysical data. This is a very relevant topic. Especially the construction of 3D training images is indeed still difficult. There are definitely some very interesting ideas in this paper, such as the different ways of*

10 *using borehole data as hard or soft conditioning data. I would like to see, however, some more discussion on the following points:*

*In the title and aims of the paper, the authors stress the importance of realistic 3D training images. They write that they present a workflow to build a training image. The part about how they build the training image (section 5.4) is however very*

15 *short. From this short description, it is not clear to me how exactly the training image was constructed. On what data is the TI based? On seismics or on the existing Miocene model? How were the shapes/geometry/position of the clay/sand featureds determined? What was the input in the Geoscene3D model? How exactly does this model work? What are the assumptions? Is this a manual or an automatic method? What are the "interpretation points" and where do they come from? They refer to a methodology in an earlier paper, but this paper uses completely different input data, i.e., airborne EM data?*

*On page 7, the authors write that "the results are evaluated and compared against the structures expected from the Miocene model"? How exactly? Was this model not already used as a basis for the TI? Is it then fair to use it again to evaluate the resulting structures?*

25 **In the new version of the manuscript, the TI characteristics and the details of the strategy we have used during the TI development are provided in the extended section "5.4 Training image", and discussed thorough Fig.s 9, 10, 11, 12, and 13. Along the paper, we emphasize the importance of unconditional realizations to assess the quality of the selected TI (actually, in combination with the used simulation algorithm). As it is shown by the comparison in the original Fig. 10, and in the new Fig.s 11, 12, and 13, even small perturbations in the TIs (Fig. 9) affect significantly the**

30 **associated realizations.**

**Moreover, in the new version of the paper, we added further details to the description of the TI construction: The first TI test was based on the existing 3D geological model (the Tønder model) covering an adjacent area. This first TI attempt was then manually adjusted during several iterations based on the unconditional output (in the paper, for simplicity, only the initial and final TIs are shown). This iterative process stopped when the corresponding**

unconstrained realization was found satisfactory in terms of its ability to mimic the geological features we expect in the Miocene, across the study area. Those expectations about the geology are based on our prior geological understanding of the area, the available seismic lines, and the few existing deep boreholes. The entire procedure is manual (except, of course, the unconstrained simulation).

Regarding the "interpretation points", they relate to the interpretation of the geophysical data and the consequent construction of the associated (manual) 3D geological models (as described in Kristensen et al., 2015 and, shortly, in Fig. 5 in the manuscript). In the presented workflow, in particular, they are used, within Geoscene3D, to efficiently modify the TIs. In fact, the interpretation points define the surfaces delineating the volumes to be populated with, for example, sand/clay voxels. By manually changing their locations and creating/deleting some of them, we could have a full control over the adjustments of the TIs. In order to make this point clearer, in the revised version, we added few lines with a short explanation on the use of the "interpretation" points for the TI construction.

Jørgensen et al., 2013 is mentioned simply as an example of research performed by using the voxel modelling tools available in Geoscene3D and specifically designed for (manual) 3D geological modelling.

*The simulated structures are relatively simple and uniform. Is it really necessary to use MPS? If you would model sand/clay occurrence with a more simple method such as indicator kriging, you would probably get similar results? It would strengthen the paper if you can prove somehow that this relatively complex approach has significant advantages over simpler approaches.*

We showed that even small details, which seem to be "irrelevant" in the TIs (that are capturing the available statistical information regarding the object to simulate), have actually significant impacts on the results (old Fig.s 9, 10, and new Fig.s 11, 12, and 13).

Moreover, this is not meant to be a paper about the comparison of the performances of MPS as a tool to incorporate complex statistical information against other "simpler" approaches. This manuscript deals with the optimal strategies (within the MPS framework) to include the available data into the simulation as soft/hard conditioning, and to build the most (geologically) effective TIs.

As a matter of fact, however, we believe that MPS' strength lies also in being quite intuitive: for many users, it is difficult to apply (and deeply understand), for example, indicator kriging and properly choose variogram models, while MPS allows the geologist to provide complex information in form of TIs, which is clearly a more intuitive (and, at same time, generally, a more effective) approach.

*More in general, the conclusions of the paper are only based on visual inspection of the simulated clay/sand patterns. There is not objective or quantitative way of comparing the different results. For example in Figure 11: could you not use cross-*

*validation or something similar to come to a more objective comparison of the different realizations? I also wonder how relevant the differences between the different realizations are, e.g. when you state that "the realizations showed a significant sensitivity to the TI". If you put these different realizations in a groundwater flow model, it is quite plausible that they all give similar results. It would be really interesting to using your geological model for some flow runs to see whether the*
5 *different realizations based on different conditioning strategies really result in different groundwater flow patterns.*

**We agree that, in principle, it would be very interesting to investigate the results of flow models based on the different realizations. However, we feel that this would be out of the scope of the paper. Adding this kind of considerations would probably make the discussion lengthy and distract the reader from the main focuses of the paper, that are: 1)**
10 **how to include, in the most effective way, the available data as simulation conditioning, and 2) how to build a TI that is capable to reproduce the geological features we want to see in the realizations. In particular, regarding the latter point, the ability to generate meaningful geological realizations should be seen in a general perspective and, so, relevant per se. Hence, even if, in this specific case, the differences between the two realizations in Fig 10 might be not "significant" for a flow model, the geological differences are evident, and, in principle, could impact the by-products**
15 **generated by using the realizations. This paper is about the optimal practices to prepare the best possible stochastic geological inputs for subsequent applications.**

**Concerning quantitative analyses of the different realizations across the paper, regarding the two realizations in Fig. 10, in the new manuscript, we added three more figures showing the distributions of the size, eccentricity, and**
20 **jaggedness of the connected sand bodies.**
**In general, with respect to a quantitative analysis of the proportions variability of the different realizations discussed across the paper, we think that our point (i.e.: the kriged sand probability is effective in enforcing the proper spatial trend) is clearly demonstrated by the comparison between Fig 17b and Fig. 8. The two figures allow a voxel-by-voxel comparison between the soft conditioning distribution (derived from the boreholes) and the final corresponding**
25 **realization. However, following the reviewer's suggestion, we calculated the cross-correlation between the kriged sand probability (Fig. 8) and the realizations (c) and (d) in Table 1 (and showed, for example, in the new version, in Fig. 14c and Fig. 14d). Not surprisingly, the correlation with the realization (d) has a much pronounced and higher maximum.**

30 *The authors claim that in the study area many of the borehole records are of low quality. Why is that? In what sense are they of low quality? How can users in other study areas determine the quality of boreholes and decide whether they can treat the borehole data as hard or soft data?*

**Boreholes, like any other kinds of data, are affected by uncertainty. The level of uncertainty determines the quality/reliability of the measurements (i.e., the quality/reliability of the boreholes). Many factors impact the quality of boreholes. Just to mention a few of them: (i) the drilling methods (e.g.: rotary drilling and air lift drillings. In some cases, for example, the finer sediments can be flushed out and the driller could potentially misinterpret, as more** 5 **sandy, a clay layer); (ii) the drilling purpose (sometimes, if the goal is to reach a specific target, the lithological description can be poor since it is not a priority); (iii) the age of the boreholes (for example, nowadays, in Denmark, samples are collected systematically every meter, and this was not the case few years ago); (iv) the presence of simultaneous wireline logging data (these kinds of ancillary information make the geological interpretation definitely more certain). During the initial phases of a standard geological modelling, a skilled geologist goes through all the** 10 **borehole records and verifies all these different pieces of information and check for inconsistencies.**

**In the paper, we mentioned that, for example, we use the seismic data as hard data because they were considered highly reliable and the scale of the structures they were able to delineate was comparable to the scale of the simulated outputs. Clearly, this is not true for the boreholes.**

**For sake of completeness, following the reviewer's request, we decided to add, to the new version of the paper, few** 15 **lines to clarify how the borehole data should be, at least in principle, evaluated and prepared for geological modelling.**

*The results of the paper are based on visual comparisons of individual realizations using different conditioning strategies. It would be really interesting to see multiple realizations for each conditioning strategy to see the variability and uncertainty of* 20 *different realizations.*

**We agree with the reviewer that investigating the variability and uncertainty would be interesting. However, we believe that this would be out of the scope of the present paper.**

25 *On page 9, the authors write that realization rarely have the same spatial variability as the training image. I find this a really strange statement. The idea of MPS is that you produce realizations with similar spatial patterns as the TI. If the realizations do not show similar patterns, this usually means that the parameters have not been chosen optimally or that the TI was for example not large enough. They also claim that therefore a TI should be chosen together with a specific MPS algorithm? If find this really strange. In my experience, all MPS algorithms can produce realizations with similar patterns to* 30 *the TI given that they are used in the right way. If the authors really want to claim that different MPS algorithms have different capabilities in reproducing patterns, they should show a comparison of the different algorithms.*

**The meaning of what we wrote in section "7 Discussion" concerns the fact that, while, ideally, the effects of the TI should not depend on the algorithm choice, these is not true in practice. The different implementations, and the**

**different parameters that can be tuned during the simulation, make the realizations algorithm-dependent. It would not make much sense to compare the realizations generated with different algorithms, not only because different algorithms have different parameters to be set, but also because the point of the paper is that the TI must be developed, no matter what, by studying the unconstrained realization and its accordance with our geological expectations.**

**Basically, we claim that the TI and the simulation algorithm are all interdependent elements structuring the final models, and, as such, the evaluation should take place a-posteriori on unconditional realizations.**

*On page 12, the authors write that "probabilistic models need to be developed and refined in order to utilize the multiple realizations and the uncertainty the represent". There are however many methods available and applications of MPS using multiple realizations to assess the uncertainty. The methods to do this are available but the authors have chosen to work with single realizations.*

**Writing that, we meant that applications and methods that are making use of the full potential (for example for geological modelling) of probabilistic models (with their large number of realizations and information regarding the uncertainty) are still under development. Those lines in the original manuscript were not about the possibility to assess the uncertainty via analyzing the realization ensemble.**

**The reasons why we decided not to investigate the uncertainty are largely discussed before in the present document.**

*The paper is clear and well written. The figures are of good quality.*
*P4, line 24-25: "along extended profile": error in grammar of sentence?*

**We thank once more the reviewer for the positive comments, especially because we invested a lot of time in the preparation of good quality, and, hopefully, clear, figures.**

**We believe that "along extended profile" reflects what we mean.**

**Response to Referee #3 comments**

We thank Referee #3 for his/her valuable comments and suggestions. The response to the individual comments is given in the following. The original review is quoted in *italics*, whereas the response of the authors is given in **bold** font.

5 *Multiple-point simulation (MPS) is a geostatistical simulation technique first developed at Stanford University in the early 2000's. An MPS algorithm is used to reproduce spatial patterns, such as connectivity, that are depicted in a training image (TI), which contains the possible spatial configurations for any given geological object and relationships between objects. A TI contains only spatial patterns and their respective likelihoods. A frequent pattern appears more often in the TI than a rare one. The actual position of a pattern in the TI is largely irrelevant, the MPS algorithm sees a set of patterns and tries to set*
10 *them together through a randomization process. In order for an MPS simulation to produce a reasonable representation of any given geological system, it must have to honor some conditioning data and have a method of accounting for spatial trends in the probabilities of selecting patterns from the TI.*

*The title of this paper suggests that, in this case, these MPS simulations are to guide subsequent hydrogeological*
15 *applications. The title also suggests the main thrust of the paper is the importance of developing realistic 3-D TI's and strategies for conditioning MPS models. However, this topic is only discussed rather briefly (in 10 lines) in section 5.4 on page 7!*

**In the new manuscript, the TI characteristics and the details of the strategy we have used during the TI development**
20 **are provided in the extended section "5.4 Training image". Moreover, they are discussed thorough Fig.s 9, 10, 11, 12, and 13. Besides, along the paper, we repeatedly emphasise the importance of unconditional realizations to assess the quality of the selected TI (in combination with the used simulation algorithm).**
**We believe that, in the new form of the paper, the iterative strategy we developed for the optimal construction of the TI is sufficiently clear.**
25
**Regarding the conditioning strategies, they are discussed throughout the entire manuscript.**
**In the first place, they are introduced in the methodological section "5 Defining MPS input information". For example, sections "5.1 Seismic data" and "5.2 Existing 3D model" are devoted to the reasons for using the seismic data and the boundaries of the pre-existing geological model as hard conditioning, while section "5.3 Borehole data"**
30 **describes how to translate boreholes into soft conditioning data by means of a moving window approach and discusses the importance of kriging the "localized" borehole information to enforce the necessary spatial trend.**
**The results of the application of the different conditioning strategies - introduced in the section "5 Defining MPS input information" - are then compared in section "6 Results".**

**Section "7 Discussion" examines the assumptions and choices we made and the future possible developments of the presented conditioning strategies and TI development workflow.**

**So, a large portion of the manuscript is devoted to the detailed descriptions of the optimal approaches for conditioning and TI construction, which are the focus of the present study.**

*I thus found the paper rather confusing and yet I recognize that the authors and their organizations have considerable experience in 3-D geological modeling for hydrogeological applications, and that there is a considerable body of observational data in southern Denmark that can support the development of multiple-point simulations of the subsurface environment.*

*In short, while individual sentences and paragraphs use consistently good English, I became uncertain about many important details of their research and objectives. The paper is quite lengthy in its current form, yet it leaves many questions unanswered. If fact, I believe the paper raises more questions than it answers.*

15 *I therefore propose that, for publication, the authors undertake to reorganize the current text into something similar to the following:*

*Section 1: Introduction: This should clearly define the background, objectives, a scope of this project. These topics, I believe, include: • A desire to evaluate the Miocene sediments over a 2810 sq.km. area of southern Denmark where they*

20 *provide the source of most drinking water. • For about 22% of the area, there is a detailed 3-D stratigraphic model (lithostratigaphic and/or hydrostratigraphic?) developed by deterministic methods (the Tonder model) • Southern Denmark has some high-resolution seismic surveys that can be used as conditioning data for MPS simulations (However, the authors need to provide more information about the spatial adequacy of these surveys, not just that they total 170 km and are shown (rather poorly) on Figure1). • While existing borehole records are available, they are of relatively low quality*

25 *(WHY?) and most borehole are relatively shallow, so these can provide only limited-value conditioning data. • The project was undertaken to determine if MPS could produce 3-D subsurface information over the entire southern Denmark area more efficiently than deterministic modeling, yet still produce information of value to further hydrogeological models.*

*Section 2: The Study Area and Available Data Sources: This should summarize its geological character and assess the*

30 *various data sources. This can be accomplished by revising as necessary existing sections 2 and 3 and Figures 1-5. A discussion about trends should be enhanced.*

*Section 3: The Experimental Process. This needs considerable expansion from existing section 4. Several questions arise from reading the existing paper. Chief among them: Was the Tonder model the source used to develop the TI? Currently this*

35 *is unclear. Later the use of two TI's is noted. How were they developed/selected? What are the spatial characteristic patterns desired in the TI?*

*Section 4: Analysis of Results. This will combine information from existing sections 5 and 6 and some of 7. It also should address several of the limitations defined below.*

*Section 5: Discussion and Conclusions. This should be relatively short, but include some of the ideas in existing sections 7 and 8. It also should address the need to determine what level of subsurface detail is required to produce an acceptable groundwater management tool for regional and more site-specific applications in southern Denmark (see my final comments).*

**We arranged the paper as follows:**

1) **"1 Introduction" - the first section after the abstract - is devoted to a very general and brief discussion of the existing literature on MPS.**

2) **The second section "2 Study area" deals with the overall geological framework of the area. So, this section delineates the general geological setting where the unit investigated in the paper (the Miocene) is embedded. This, at the same time, contextualizes the presented research and better defines its limits.**

3) **Section "3 Data" describes in detail the amount and characteristics of the different kinds of data available (borehole data and seismic measurements). In particular, a large part of the paragraph "3.2 High-resolution seismic data" discusses - to a reasonable level of detail for the purpose of the present study - the different specifications of the seismic data available.**

4) **Section "4 Establishing framework-model constraints" goes into details about the specific geological unit targeted in the research.**

5) **"5 Defining MPS input information" is the most methodological section of the paper. Here, the approaches used in the comparison throughout the manuscript are described. For example, it is discussed how (and why):**

   (i) **the seismic data and the existing manual adjacent model have been incorporated as hard conditioning (paragraphs "5.1 Seismic data" and "5.2 Existing 3D model");**

   (ii) **to translate the boreholes into soft probability to address their uncertainties (paragraph "5.3 Borehole data");**

   (iii) **to build an effective TI based on the outcomes of the unconstrained simulations ("5.4 Training Image").**

6) **Section "6 Results" is about the detailed comparisons of the outputs resulting from the application of: (i) the iterative approach for the construction of the TI and (ii) the different conditioning strategies detailed in the previous section.**

7) **The second last section, "7 Discussion" discusses the assumptions/choices made across the paper and their possible limitations and possible, future, developments.**

8) **"8 Conclusion" is a very concise section where we simply summarize our results.**

**We trust this is a reasonable way to present our research.**

**Clearly, other scientists might think differently. To some extent, it is simply matter of taste, as long as the rationale behind the choice is evident. And we believe that already in the present form, the overall logical organization of the paper is quite clear and effective. Nevertheless, in the new manuscript, we further expanded the "1 Introduction" to describe the organization of the paper.**

**Regarding the specific questions posed by the reviewer:**

1) **Many factors affect the quality of boreholes: (i) the drilling methods (e.g.: rotary drilling and air lift drillings); (ii) the drilling purpose (sometimes, if the goal is simply to reach a specific target, the lithological description is not a priority); (iii) the age of the boreholes (older boreholes are generally less reliable); (iv) the presence of simultaneous wireline logging data. During the geological modelling phase, all the borehole records are (or should be) checked for inconsistencies. In the study area, as stated in the manuscript, the few available deep boreholes are characterized by high level of uncertainty.**

2) **The procedure for the construction of the TI was already discussed in several parts along the original paper, however, to make our point clearer in the new manuscript, we largely expanded the section "5.4 Training image" and added three new figures (Fig.s 11, 12, and 13).**

3) **MPS has not been considered as a way to "***produce 3-D subsurface information over the entire southern Denmark area more efficiently than deterministic modeling***". With the available input data (in terms of types, quality, and amount), MPS is the only way to produce a meaningful estimation of the geological variability within the Miocene at the scale that is reasonable for groundwater investigations.**

*As I reviewed the current draft, I assembled a list of what I consider to be its current limitations. These include:*

*1) Apparently, the research so far has focused on examining the role of various hard and soft conditioning data, but only a single realization is used for each setting. This is clearly insufficient. Repeated applications of MPS will produce a sequence of slightly different realizations even with the same conditioning setup, and it should be possible to quantitatively evaluate their similarities and compare this to the differences introduced by the changed conditioning strategies.*

**Clearly, the differences we highlighted for each conditioning setup are not realization-dependent. This means that they will appear consistently for every realization obtained with the same conditioning settings. Just to mention an example, all the realizations generated with the "borehole as hard data" (Fig. 15b, in the new version) will be "perfectly" matching all the borehole samples by construction. The same is true for the "borehole as soft data" (Fig.**

15c, in the new version); in this case, we confirmed empirically that the SNESIM ignores (Hansen et al., submitted) "localized" soft data.

Of course, considering more realizations would allow uncertainty assessment. And this would be, in principle, very interesting. However, as the reviewer has pointed out, the paper is already quite long and a discussion about the uncertainty would fall out of the original scope of the manuscript. This article is, in fact, meant to be simply about the best conditioning strategy and the optimal preparation of the TI.

*2) Regarding the Training Image aspect of MPS, it seems that two TI's were used. It is unclear how they were developed and what underlying geological concepts or knowledge were used to develop them. Figure 9 does not clearly show the sand/clay layers – the colors are not ideal for this, and the text (line 15 Page 7) merely states one realization has more layers than the other. Does either seem more likely with the geological knowledge available? Are these layers defined as channels or sheets (continuous layers)? How does either TI relate to conditions within the Tonder model?*

One of the main points this manuscript would like to convey is the importance of unconditional realizations to evaluate the effectiveness of the selected TI. In particular, we discuss this in the new "5.4 Training Image" and by using three additional figures (Fig.s 11, 12, 13).

In addition, in the extended version of section "5.4 Training Image", we provide further details and underline that: (i) the first TI test was based on the existing 3D geological model (the Tønder model) covering an area adjacent to the simulated one; (ii) then, during several iterations, this first TI was manually adjusted based on the unconditional outputs (however, in the paper, for simplicity, only the initial and final TIs are shown in Fig. 9 together with the associated unconstrained realizations in Fig. 10); (iii) this iterative process was stopped as soon as the corresponding realization was able to mimic the geological features expected in the Miocene.

Those expectations about the geology are based: (i) on our prior geological understanding of the area, (ii) the available seismic lines, and (iii) the few existing deep boreholes.

*3) The assessment of the results is mostly qualitative. Quantitative tools to do exist and should be used.*

With respect to a quantitative analysis of the proportions variability, we think that our point concerning the effectiveness of using the kriged sand probability to enforce the proper spatial trend is clearly demonstrated via the comparison between Fig 17b and Fig. 8. In fact, the two figures allow a voxel-by-voxel comparison between the soft conditioning distribution and the final corresponding realization. However, to meet the requests of the reviewer, we included an additional figure (Fig. 18) with the cross-correlation between the kriged sand probability (Fig. 8) and the realizations (c) and (d) in Table 1 (showed, for example, in Fig. 14c and Fig. 14d). Not surprisingly, the correlation with the realization (d) has a much more pronounced and higher maximum.

**Regarding the quantitative assessment of the patterns produced by using the two different TIs in Fig. 9, in the new version, the distributions of the size, eccentricity, and jaggedness of the connected bodies in the realizations in Fig. 10 are compared in the new Fig.s 11, 12, and 13, respectively. Also in this case, the new figures further confirm what is evident from Fig. 10a and Fig. 10b.**

*4) The cited literature appears to miss several important more recent studies. Attached are a few representative paper citations.*

**In the revised version, we added several, relevant, references.**

*FINAL COMMENTS*

*MPS is an interesting and potentially powerful method for developing very useful subsurface geological models. I am aware that at least some of the authors have experimented with other simulation approaches, such as TPROGS to apparently successfully simulate facies heterogeneity in buried valleys. It would be interesting for them to include at least a short comparison between MPS and other simulation approaches. The current paper assumes the reader to be proficient in MPS concepts. This may not be true in many cases, so a short comparison in the introduction might broaden the readership and understanding of the importance of this line of investigation.*

**We believe that the paper (already "*quite lengthy in its current form*") would not benefit from a comparison with other simulation approaches as the manuscript is not about checking the performances of MPS as a tool to incorporate complex statistical information against other "simpler" approaches. This article deals solely with the optimal strategies (within the MPS framework) for the soft/hard conditioning and for the construction of the most (geologically) effective TI.**

**In addition, the approach of the submitted article is based on TIs for the description of the information about spatial structures, while TPROGS does not make use of TIs and, instead, requires the user to define transition probabilities from which the realizations are then simulated. Therefore, given the topic of the paper, it is not natural to consider the use of TPROGS.**

*I believe the ultimate goal of this research is not to model the Miocene of southern Denmark as a purely academic exercise, but to use this information to guide groundwater management schemes. I wonder what groundwater model sensitivity analysis would yield in terms of necessary subsurface detail for producing acceptable groundwater resource management at regional or site-specific scales?*

*Obviously, somewhat less precise spatial definitions are likely to be required for regional assessments. On the other hand, site-specific studies may be unreliable if based only on MPS inputs without careful additional local conditioning. So, the question arises, "Where does MPS fit within this overall objective?" This question also reflects on the limitation noted on page 9 (lines 13-15) the present inability of MPA to handle graben structures and faults.*

**The present paper is about: (i) how to include, in the most effective way, the available data as simulation conditioning, and (ii) how to build a TI that is capable to reproduce the geological features we want to see in the realizations. In particular, regarding point (ii), the ability to generate meaningful geological realizations must be seen in a general perspective and, so, relevant per se.**

10 **Moreover, MPS is a way to "see" beyond the (strictly speaking) available data. In fact, via the prior geological knowledge formalized by the TI, it is possible to "reconstruct" features compatible, at the same time, with the data (e.g.: the geophysical measurements), but also with the desired prior geostatistical information. This is a very important aspect in the paper: a significant amount of data - but not as much as we desired (as always happens) - was available together with the knowledge of the geological features we wanted to see in the realizations. Thus, we**

15 **investigate the best workflow to utilize the full potential and flexibility of MPS methodology to jointly exploit those two pieces of information and obtain a realization that is, simultaneously, matching the data (i.e., boreholes, geophysical measurements, and pre-existing geo-models), and having the required spatial characteristics.**

**The scale of the geo-features necessary for flow modelling purposes is hard to decide before entering in the more hydrological aspects of the problem. In the paper, we show how to get those features in an effective way and we test**

20 **the proposed approach at the scale we think is the more appropriate one. Further analysis would be definitely necessary, but we think they are not falling in the scope of the present study.**

*Despite my several criticisms, I think this is a potential important paper and hope the authors will consider reorganizing it and adding in a few details on some important methodology issues, while at the same time focusing on the TI and*

25 *conditioning strategies.*

*SOME SUGGESTED REFERENCES:*

*Boucher, A. (2011) Strategies for Modeling with Multiple-point Simulation Algorithms.IN: "Closing the Gap", 2011 Gussow Geoscience Conference, Banff, Alberta. 9p.*

30 *Kessler, T. (2012) Hydrogeological Characterization of Low-permeability Clayey Tills – the Role of Sand Lenses. PhD Thesis, Department of Environmental Engineering, Technical University of Denmark, Lyngby, Denmark. 80p.*

*Klenner, R., Braunberger, J.R., Dotzenrod, N.W., Bosshart, N.W., Peck, W.D. & C.D. Gorecki (2014) Training Image Characterization and Multipoint Statistical Modeling of Clastic and Carbonate Formations. PowerPoint Presentation, 2014 Rocky Mountain Section AAPG Annual Meeting, Denver, Colorado. 27 slides.*

*Meerschman, E., Pirot, G., Mariethoz, G., Straubhaar, J., Van Meirvenne, M. & P. Renard (2013) A Practical Guide to Performing Multiple-Point Statistical Simulations with the Direct Sampling Algorithm. Computers & Geosciences. 52. p. 307-324.*

*Straubhaar, J., Walgenwitz, A. & P. Renard (2013) Parallel Multiple-Point Statistics Algorithm Based on List and Tree Structures. Mathematical Geosciences. 45. p. 131-147.*

*Straubhaar, J., Renard, P. & G. Mariethoz (2016) Conditioning multiple-point statistics simulations to block data. Spatial Statistics. 16. p. 53-71*

**Response to Reviewer #4 comments**

We thank Dr. Xin He for his valuable comments and suggestions. They certainly improved the manuscript. In the following, we provide the response to the individual comments. The original review is quoted in *italics*, while our response is given in **bold** font.

*The study aims to establish a workflow to carry out Multi-Point Statistics modeling for a testing area in Denmark using alternative 3D training images and various conditioning strategies. The research topic is of high relevance to those who work with hydrogeological modeling. The manuscript is well written with accurate language, rational methodology and convincing results. I recommend the manuscript is accepted for publication with minor revision. However, there are several*

10  *details to be considered which are listed as follows:*

*As stated in the abstract, the introduction and the conclusion sections, one of the most important steps of the workflow is to develop 3D TIs in an iterative way. However, this part is only briefly mentioned in the method section and not at all mentioned in the result section. I am curious about how the TIs are evolved gradually with feedback information after each*

15  *step of adjustment, namely from the initial TI to the final TI.*
*Additionally, is Fig 9 showing the initial or the finally TI?*

**We definitely see the reviewer's point and, in the new version of the manuscript, we added further details regarding the development of the TI, from the initial guess, TI1, in Fig. 9a, to the final result, TI2, in Fig. 9b. In particular, in**

20  **the revised paper, we stressed the fact that: (i) the first TI simply consists in a portion of the adjacent, pre-existing, geological model (the Tønder model); (ii) then, this initial attempt has been iteratively and manually adjusted based on the output of the associated unconstrained simulation; (iii) this iterative process ended when the final unconstrained realization was found satisfactory in terms of its ability to mimic the geological features we expect in the Miocene across the study area.**

*In the introduction section, I would suggest to add a few sentences indicating the main objectives of the study.*

**In the revised version of the manuscript, we followed the reviewer's suggestion.**

30  *Lin 210-211. The moving window for calculating the borehole uncertainty in the vertical direction is 20 m. Meanwhile, in Fig 7(b), the thickness of the Miocene layer is about 150 m, which in principle corresponds to 7 to 8 intervals in each borehole at maximum. However, as far as I can count, there are usually more than 7 color blocks in each borehole. Am I mistaken for something?*

**20 m is simply the width of the moving window. This means that the information about the categories (the lithologies) is averaged across a 20 m wide interval. This does not necessarily imply that only 7 or 8 samples remain after the application of the moving window. Actually, the size of sampling interval is unchanged, and the only minor**

5 **modification on this respect consists in a loss of a certain amount of samplings at the top and bottom of the boreholes due to the fact that, in our specific implementation, we considered a window with always the same width.**

*Fig 7, when interpolating the borehole uncertainty, are the borehole data outside the model domain being considered, both horizontally and vertically? If not, would there be extrapolation instead of interpolation towards the edges of the model*

10 *domain?*

**We do not interpolate the uncertainty of the borehole. Instead, we krig the sand probability of the portions of the boreholes lying within the Miocene. So, if this was the question of reviewer, generally, towards the edges of the model domain, extrapolation does occur.**

*In the results section, L259-266, there are two TIs being tested, one clearly has more layers than the other. Do these two TIs have any relation to the iterative approach the authors try to present in the study? Or is it a separate issue here? Moreover, it says the second TI is chosen because it is closer to what has been presented in Kristensen et al., 2015. So maybe it is better to describe very briefly what is in the Kristensen's study, and why that one is used as benchmark.*

**The two TIs are, respectively, the first and final test along the iterative TI development process. In particular, the second TI (TI2 - Fig. 9b) was selected to run all the subsequent conditioned simulations as the associated unconditioned realization (and not the TI2 itself) was fund able to mimic the geological structures characterizing the Miocene in the study area. The characteristics we expect for the Miocene structures are described in Kristensen et al.,**

25 **2015 and discussed in the manuscript in the dedicated section "4 Establishing framework-model constraints" (and in the associated Fig.s 2, 3, 4, 5).**

[revised manuscript text omitted]

---

## Referee Report (RR1)

General comments

The paper is globally well written and interesting. However, the title and introduction are too general, which is misleading as the reader could expect a methodological framework rather than a specific application of 3D MPS with a conditioning sensitivity analysis. The motivations, novelty and specificity of this work should be better stated in the introduction. As this paper investigates soft conditioning, it seems strange not to perform multiple MPS realizations (until the appendix) and not to compute probability maps to compare to the soft data. The authors should justify strongly and properly the use of a single realization to assess the quality of soft data conditioning in the main part of the paper.

I do not understand why the 100 generated realizations are only used in the appendix, and why the authors do not state in the text that they produced 100 realizations per strategy (it is only written in the caption of figure 22). In addition, it reduces the validity of the performed analysis to display the histogram of the sand body size, eccentricity and jaggedness for 3 specific realizations rather than showing the mean plus minus 1 or 2 variance(s) of the histograms, to account for the 100 available realizations. These results should be incorporated in the paper, and the discussion could eventually be enriched by a comparison of quality assessment using 1 realization versus 100.

Details

Introduction : the motivations and objectives need to be clarified. For instance, page 2 line 28-29, you could as described in Pirot (2017) give a few examples of ways to produce 3D training images (by object-based or process-based algorithm), or by enhanced tools (Comunian, 2012 ; Rezae, 2015).

Page 2 line 30, it should be indicated that it is applied to a specific dataset.

Page 3 regarding the main objectives, it seems important to precise that the iterative development is performed by an expert user, and that it is applied to a specific dataset. The actual formulation might lead the reader to think that you propose a general framework to develop 3D TIs, which is not really the case. The TI is adapted iteratively such that the resulting prior (MPS realizations influenced by algorithm parameters) fits qualitatively your expectations (a quantitative criteria would be an argument to claim a new methodology). For the second objective, it also seems to be an application of a recent strategy proposed by Hansen et al. (submitted).

Introduction/Study area : would it be a good place to define your voxel dimensions (grid resolution) and justify it ? It could also be the right place to explain why you are not interested by multiple realizations and justify your choice, which comes too late in the current version of the manuscript.

Page 5 line 31 : the bottom surface has a grater dip than the top one, as the thickness increases from east to west.

Page 6 line 12 : please justify the high reliability of the seismic data interpretations.

Page 7 line 12: as the paper by Hansen et al. (submitted) is not yet published, it is necessary to explain in more details what is meant by 'ignore such localized soft data'.

Page 8 line 16: it would be could to recall the definition of your indicators displayed in Figures 11 to 13, and explain why they are significant when computed on a single realization, and why they are not rather computed on the TIs. Are these indicators useful as they are not considered in a quantitative criteria to select the TI? If they are not used, they might be removed, as well as the corresponding figures.

Page 9 line 5: 'which further supports our final choice for the TI2'. Is this really a good argument? As you display horizontal slices and as the TIs mainly differ by the layer thickness, I am not sure sections in realizations generated with TI1 would have been less good.

Page 9 line 20: can you give some reasons for this (linked to the algorithm? …)

Page 9 line 22 / Figure 15c&d: how do you justify the quality of your realization when using soft conditioning if you can not compute probability maps with a single realization?

Page 10 line 12 / Figure 18: why is the cross correlation shaped like a pyramid? Is it due to voxel indexing? Which voxel indexes correspond to borehole information? Should you recall the definition of the cross-correlation to explain that shape?

Page 10 line 30: do you mean 'geo time ad yz-axis'?

Page 14 line 9: could you precise 'This study investigates conditioning strategies' ?

Page 19 line 2: can you reformulate the beginning of your sentence?

Figure 4 : could you super-impose the graben structure like in Fig 1b, it would facilitate the interpretation.

Figure 5 : missing legend for the colored lines

Figures 11,12,13 : legend on the figure should state realization (TI#) and not only TI# to be coherent with the caption.

Figure 15 : missing color legend for the boreholes. It is not straightforward to localize the mismatch. Would a white line around the boreholes + same colors as on the simulation when it matches and yellow or black when it does not match and outside the simulation grid help ?

Figure 22: what is 'e-type map' ? The presented results are not clear. The caption should describe clearly what is displayed

---

## Author Response (AR2)

**Response to Editor's comments**

The original prof. Gudici's review is quoted in *italics*, while our response is given in **bold** font.

*The two Referees gave opposite assessments of the manuscript (accept vs. very major revision) and therefore I read it again*
5 *very carefully. Also, I deeply reconsidered the comments to the original submission. After this examination, I agree with Referee #1, and I stress that some of his comments were raised also by other Referees during the first stage of the manuscript assessment and found limited answer in the revised manuscript. Therefore, the acceptance of the paper for publication requires that the manuscript's revision properly and fully complies with the Referee's comments. I expect that the Authors' efforts will yield a revised version which enhances the paper quality:*

**In the revised version of the manuscript, to meet the requests of the Editor and of one of the Reviewers, we added an entire new section and four figures dealing with the variability within the probabilistic models. In the present form, we believe that our paper answers all the questions raised by the Editor and Reviewer #1.**

**Response to Reviewer #1**

The response to the individual comments of Reviewer #1 is given below. The original review is quoted in *italics*, whereas our response is given in **bold** font.

5   *This paper considers data-driven training image construction and soft data conditioning with MPS. Reading the authors replies, it seems clear to me that the authors are not willing to analyze the result by considering more than one realization. However, I still feel that using MPS with only a single realization and no uncertainty quantification falls short of the very purpose of geostatistics. I do not understand this resistance because generating supplementary realizations should in principle not represent significant work. p.13, l. 29: a "probabilistic model" is mentioned, but I do not agree because no*
10   *uncertainty is investigated. Without it, it is mostly glorified 3D drawing.*

**To meet the requests of Reviewer #1, we generated 100 conditional realizations for each of the cases (c) and (d) in Table 1, and added to the revised version of the manuscript an entire section (together with four new figures).**
**Thus, in the new "Appendix A", we further discuss the variability within each probabilistic model and the effects on it of the different conditioning strategies and TIs.**

*Regarding validation of the realizations, I agree that flow simulations might not be necessary. However, the uncertainty between realizations needs to be assessed. This variability is an inherent property of the "probabilistic model" the authors are mentioning, to the same degree as the type of geobodies produced or their connectivity. It may be the case that all realizations are the same (underestimating uncertainty, it happens in some cases) or that some realizations do not have the*
20   *right properties. Not investigating this is not acceptable in my opinion.*

**In the new section "Appendix A", we address this point and get additional confirmations of the validity of our conclusions.**

*If the paper is only about TI construction and not about MPS algorithms, as the authors claim, then they should only focus*
25   *on that and would, without the need to show any realization, but this might become a thin paper.*

**The paper is about:**
**(1) the TI construction (and the coupling between the TI and the implementation of the used MPS algorithm), and**
**(2) the optimal conditioning strategy.**
**Very often both TI and the available data are not constructed and handle in the proper way. This is the problem we**
30   **wish to tackle. And we believe it is a crucial issue within the MPS community.**

*Regarding the structure of the paper, I agree with the rationale for the current structure, however I do not think it is optimal to use a large section entitled "results" to present all sorts of validation tests. These tests could be detailed in sub-sections.*

*Replies to reviewers comments, p.1, l.21-22: This shows a fundamental misunderstanding of the distinction between one realization and a random function model. Moreover, with MPS the realizations are not by definition representative of the TI,*

5     *as the algorithm and its parameters can result in large differences between realizations and TI.*

**One of the main point in the paper concerns the coupled effects of the TI and the specific implementation (and settings) of the used MPS algorithm. Each considered realization is a realization from a random function model defined by the TI, conditioning data, and the specific choice of the algorithm (and its associated settings). This is exactly why we focus a significant part of the paper on both unconditional simulations (which show the results of the**

10     **combination of TI and (settings) of the considered MPS algorithm) and conditioning strategies.**

*p.10, l.27-29: This sentence is interesting but unfortunately not demonstrated or argued.*

**If the Reviewer is referring to "*In future studies, faults should be handled, such that the MPS simulation results are affected across the faults. A possible solution could be to use the geochron formalism (Mallet, 2004), in which the simulation could be performed in a regular grid, with geo-time as y-axis*", it is simply a hypothesis on a reasonable way**

15     **to address the problem in the future.**

*figure 18 is hard to interpret as it only shows the contour of a joint distribution, and not the density itself. As such, it is very non-informative and appears to show a relatively poor correlation (or there may be a problem with the way the figure is displayed).*

20     **Fig. 18 compares the spatial 3D cross-correlations:**
**(1) between the 3D probability grid (Fig. 17b) and the realization obtained by applying the conditioning strategy (d) of Table 1 (in black in Fig. 18), and**
**(2) between the same 3D probability grid (Fig. 17b) and the other realization obtained by applying the conditioning strategy (c) of Table 1 (in green in Fig. 18).**

25     **Obviously, the normalization is the same, so, the green and black curves in Fig. 18 are immediately and directly comparable. Moreover, the cross-correlation - especially for the case (d) – is not poor; in particular, if we keep in mind the nature of the objects we are cross-correlating.**

**From the figure below (similar to Fig. 18 in the original paper, but, now, concerning several realizations of the probabilistic models), it is clear that cross-correlations associated with the realizations from conditioning strategy (d):**

30     **(1) show similar behaviours, and**

**(2) are consistently higher than those associated with the realizations from strategy (c).**

[revised manuscript text omitted]

---

## Author Response (AR3)

**Response to Editor's comments**

The original "Comments to the Authors" of prof. Giudici are quoted in *italics*, while our response is given in **bold**.

*The paper has been improved during the revisions, but I agree with the basic complaints from one of the Referees and*
5 *therefore I think that the Authors can provide a final revision, which properly accounts for those comments:*

**In the revised version of the manuscript, we took into account the comments from Reviewer #1 (in the opinion of the Reviewer #2 the article was already suitable for publication in its original form).**
**In particular, we modified Figures 19, 20 and 21, and, now they include the analysis of 100 realizations for each of the**
10 **unconstrained simulations performed respectively with TI1 and TI2.**

**It seems that the number of realizations has been of concern during this review. For this reason a further clarification is probably important: We completely agree that, if some decisions should be based on the geostatistical model, then, a larger number of realizations would be useful, and, indeed, needed. However, the main focus in this manuscript is**
15 **not the _use_ of the model, but the _construction_ of the geostatistical model consistent with all the information. This is why we base much of the reasoning/validation on a single realization, and summarize statistics for multiple realizations in the appendix for reference.**

**As requested, in the following, the Editor can find a detailed list of the relevant, most recent, modifications of the**
20 **manuscript (the Line and Page numbers refer to the revised version):**

- **Line 30, Page 2 ["1 Introduction"]: the original "**In the present study, we describe a strategy to develop effective and realistic 3D TIs based on the a posteriori analysis of the associated unconstrained realizations." **has been changed into: "**In the present study, we describe a strategy to develop effective and realistic 3D TIs based on the
25 analysis of the associated unconstrained realizations. This analysis can be performed by using a few unconstrained realizations per tested TI. In fact, only general features induced by the coupled effects of the investigated TI and the used implementation of the MPS algorithm are evaluated. These general features are common to all the realizations (as it is confirmed also by the analysis of the results generated by using hundreds of realizations and discussed at the end of the paper)."
30 - **Line 4, Page 3 ["1 Introduction"]: the original "**In this study, available sources of information are seismic lines, boreholes, and a pre-existing, manually constructed geo-model. Depending on their scale and uncertainty, we developed a practical way to incorporate all of them into the Single Normal Equation Simulation (SNESIM) workflow as it is implemented in SGEMS (Remy et al., 2009). The approach is general and can be readily extended

to other data types." **has been changed into: "**In this study, available sources of information are seismic lines, boreholes, and a pre-existing, manually constructed geo-model. Depending on their scale and uncertainty, we developed a practical way to incorporate all of them into the Single Normal Equation Simulation (SNESIM) workflow as it is implemented in SGEMS (Remy et al., 2009). For example, the data from the borehole are used in a non-standard way as soft conditioning and only after an appropriate pre-processing aiming at: (i) removing the effects of scale mismatches, (ii) properly accounting for the data uncertainty, and (iii) effectively migrating the information between the borehole locations. The results of this novel approach are compared against more traditional strategies along the article. The proposed workflow is general and can be readily extended to other data types (for example, other kinds of geophysical data).**"**

- **Line 1, Page 9 ["5.4 Training image"]: the original "**Not surprisingly, the sand lenses in the realization associated to TI1 are more jagged than those in the other realization obtained by using TI2. Because of the higher accordance between our geological expectation of the Miocene in the area and the unconstrained realization in Fig. 10b, TI2 is selected for all MPS simulations discussed in the rest of the paper.**" has been changed into: "**Not surprisingly, the sand lenses in the realization associated to TI1 are more jagged than those in the other realization obtained by using TI2. Because of the higher accordance between our geological expectation of the Miocene in the area and the unconstrained realization in Fig. 10b, TI2 is selected for all MPS simulations discussed in the rest of the paper. These conclusions are drawn on a single unconstrained realization per TI. Nevertheless, their validity is general and they do not depend on the specific realization considered. In fact, only the features induced by the specific choice of the TI (coupled with the actually used implementation of the MPS algorithm) are taken into account in the TI selection process. An in depth discussion of multiple realizations and their mutual coherence in terms of the proposed analysis is presented in the Appendix A (and in the associated Fig.s 19-21). Naturally, after the full model has been set up conditional to all the data and the selected TI, and during its use for, e.g., risk analysis or as input for hydrological modelling, a large (as large as possible) collection of realizations of this model would be useful. However, during the construction of the optimal TI to be utilized as input to the geostatistical model (as we do here) a few realizations suffice.**"**

- **Line 7, Page 14 ["7 Discussion"]: the original: "**And each individual realization (e.g., in Fig. 17) is, by construction, compatible with all simulation inputs (thus, the statistics from the TI, the hard data, and the soft conditioning).**" has been changed into: "**And each individual realization (e.g., in Fig. 17) is, by construction, compatible with all simulation inputs (thus, the statistics from the TI, the hard data, and the soft conditioning). So, to set up the proposed workflow, only the analysis of single realizations is necessary and, in the study of the performances of the different conditioning strategy, only the features present, by definition, in each realization have been taken into consideration.**"**

- **Figure 19 ["Appendix A"]: The original Figure 19 has been modified and, now, includes the mean number of sand bodies (together with the associated standard deviation) generated by using 100 realizations.**

- **Figure 20 ["Appendix A"]: The original Figure 20 has been modified and, now, includes the eccentricity mean (together with the associated standard deviation) generated by using 100 realizations.**
- **Figure 21 ["Appendix A"]: The original Figure 21 has been modified and, now, includes the jaggedness mean (together with the associated standard deviation) generated by using 100 realizations.**

**After these modifications, we believe we provided complete responses to the comments of the Reviewer #1 and the Editor.**

**Response to Reviewer #1**

The response to the individual comments of Reviewer #1 is given below. The original review is quoted in *italics*, whereas our response is given in **bold**.

5    *The paper is globally well written and interesting. However, the title and introduction are too general, which is misleading as the reader could expect a methodological framework rather than a specific application of 3D MPS with a conditioning sensitivity analysis. The motivations, novelty and specificity of this work should be better stated in the introduction.*

**This paper is about: (i) the development of a methodological framework, and (ii) its application/validation on a real, large-scale, problem. For these reasons, we believe that the title properly reflects the content of the manuscript.**
10   **The motivations and novelties of the research were already emphasized in several parts of the original manuscript; however, to accommodate the Reviewer's requests, in the revised version, we further expanded the "1 Introduction" by adding several new lines on this respect.**

   *As this paper investigates soft conditioning, it seems strange not to perform multiple MPS realizations (until the appendix)*
15   *and not to compute probability maps to compare to the soft data. The authors should justify strongly and properly the use of a single realization to assess the quality of soft data conditioning in the main part of the paper.*

**We completely agree that, if some decisions should be based on this model, then, of course, a larger number of realizations would be useful, and, indeed, needed. But, the main focus of this manuscript is not the use of the model, but the construction of the geostatistical model consistent with all the information. This is why we base much of the**
20   **reasoning/validation on a single realization, and summarize statistics for multiple realizations for reference in the Appendix A. This was discussed in the last version of the paper (e.g., lines 16-19, page 10, of the previous version, at the end of the "6 Results" section; lines 18-29, page 13, in the previous version). In the present revised manuscript, we further elaborate on this: (i) in the "1 Introduction"; (ii) at the end of the "5.4 Training image" section; (iii) in the "7 Discussion".**

   *I do not understand why the 100 generated realizations are only used in the appendix, and why the authors do not state in the text that they produced 100 realizations per strategy (it is only written in the caption of figure 22). In addition, it reduces the validity of the performed analysis to display the histogram of the sand body size, eccentricity and jaggedness for 3 specific realizations rather than showing the mean plus minus 1 or 2 variance(s) of the histograms, to account for the 100*
30   *available realizations. These results should be incorporated in the paper, and the discussion could eventually be enriched by a comparison of quality assessment using 1 realization versus 100.*

The manuscript is about constructing a realistic sized geostatistical model conditional to different data types. In the proposed strategy, the set-up of such a simulation is based/validated on a few realizations (as we demonstrate for example when we build the TI). Of course, after the model is set up, it will make perfect sense to generate a very large collection of realization for subsequent analysis and risk assessment. The latter is however not the key part of the article, and, therefore, we believe it is better to keep the analysis as an appendix for reference to the reader.

As a matter of fact, the results in the Appendix A (deduced from the analyses of hundreds of realizations) completely confirm the conclusions drawn in the other sections of the paper (and based on single realizations).

In particular, this is evident when we look at the updated Figures 19, 20, 21. In fact, following the Reviewer's suggestion, in the revised manuscript, we modified those Figures that now include the estimations of the mean and the standard deviation obtained from 100 realizations per tested TI.

We feel confident that, in its present form, the manuscript satisfies all the requests from the Reviewer #1.

[revised manuscript text omitted]

---

## Author Response (AR4)

**Response to Editor's comments**

The original "Comments to the Authors" of prof. Giudici are quoted in *italics*, while our response is given in **bold**.

*In their answer to one of the comments by Reviewer #2, the Authors claim that "This paper is about: (i) the development of a methodological framework...". However, such a methodological framework is not clearly described and discussed. They added some sentences in the introductory section, where the proposed methodology is sketched in a couple of paragraphs (from page 2, line 30, to page 3, line 13). The rest of the paper is strictly related to the specific test, apart from some methodological aspects which are somehow hidden in section "5 Defining MPS input information".*

*My basic suggestion to the Authors is to rewrite those sentences and to add a section, immediately after "1 Introduction", specifically devoted to the clear and thorough description of the methodological framework. This section should include all the methodological material about the proposed approach which is of general validity and not strictly related to the specific example.*

**In its original version the manuscript is pervaded by the general description of the proposed approach: not only in the "Introduction", but also in the sub-sections "Borehole data" and "Training image" (almost entirely devoted to the discussion of the methodology), and in the sections "Discussion" and "Conclusions". Nevertheless, in the revised version, to address the Editor's requests, we included a completely new section, "2 The methodological framework" together with the corresponding additional Figure 1 showing the flowcharts of our strategy.**

**Through section 2, we also underline that the case study that follows exemplifies an actual implementation of the proposed strategy.**

*Moreover, section "4 Establishing framework-model constraints" could be inserted as a sub-section in one of the other sections.*

**In the revised version, we moved the original section "Establishing framework-model constraints" into the new, expanded, "3 Study area".**

*A final technical remark: in figures 11 and 19 the measurement units of the x-axes ("size of the sand bodies") are missing; meters, I suppose.*

**In the new manuscript, the measurement units of Fig.s 11 and 19 are now explicit (number of voxels).**

[revised manuscript text omitted]

| b | 2$^{nd}$ (Fig. 109b) | ÷ | The Tønder model
Seismic interpretation
Boreholes |
| c | 2$^{nd}$ (Fig. 109b) | Sand probability directly from boreholes | The Tønder model
Seismic interpretation |
| d | 2$^{nd}$ (Fig. 109b) | 3D kriged grid of the sand probability distribution from boreholes | The Tønder model
Seismic interpretation |

**Table 1: The different conditioning strategies tested in this study. The corresponding realizations are presented in the Figs. 154 - 176.**

---

## Author Response (AR5)

**Response to Editor's comments**

The original "Comments to the Authors" of prof. Giudici are quoted in *italics*, while our response is given in **bold**.

*The paper has been revised and slightly reorganized by the Authors in such a way as to conform with the expected quality*

5 *standard for publication on HESS.*

*In particular, the new section "2 Methodological framework" and the workflow in Figure 1 clarify the proposed method, which was originally hidden in the description of the specific case study, and will permit the reader to test the method with other data sets.*

10 *I provide below a few technical comments.*

*1) Page 2, lines 27 to 31. Substitute this paragraph, possibly with the following:*

*"In the present study, we describe a strategy to develop effective and realistic 3D TIs based on iterative modification of the TI images, so that general features of the unconstrained realizations meet the expected properties. Such an approach permits to analyse the coupled effects of the investigated TI and of the specific implementation of the MPS algorithm. Moreover, the*

15 *tests performed in this paper show that few unconstrained realizations per tested TI are sufficient to obtain a correct assessment of the general features common to all the realizations."*

*2) Page 3, line 3. Substitute "the borehole" with "boreholes".*

*3) Page 4, lines 3 to 4. Susbtitute "in our way to proceed", possibly with "in this framework".*

*4) Page 4, line 6. Substitute "an initial TI attempt", possibly with "an initial tentative TI". Moreover, it is necessary (i) to*

20 *give an idea of the procedure to be adopted for the iterative adjustment of the TI and/or (ii) to clarify if such an adjustment can be done in an objective way or with a trial-and-error approach.*

*5) Page 4, line 8. I suggest to erase the sentence in parentheses, because the effect that some modifications on the TI have on conditioned realizations, depends on the conditioning data, above all on their density, and therefore it would be necessary to explicitly state under which consditions this statement is correct.*

25 *6) Page 4, lines 10 to 11. Please, rephrase the sentence "For example,... sand bodies", possibly as "For example, in the present article, updates were based on the size, elongation, and compactness of the resulting sand bodies and on the visual analysis of a single realization per considered TI."*

*7) Page 4, line 12. Please rephrase "visible to the modeller building the TI and studying the realizations".*

*8) Page 4, lines 15 and 16; page 5, line 9; page 7, lines 1 and 3; page 8, line 26; page 11, line 23; page 12, line 4; page 16,*

30 *lines 14, 29 and 31. Substitute "borehole", possibly with "borehole data", "borehole logs", etc.*

**The manuscript has been modified to address the requests from the Editor.**

*9) Page 4, line 19. Is "adjoining" the right word?*

**Yes, we believe it is.**

5 *10) Page 4, line 25. Erase "of".*

*11) Page 5, line 3. Add "at large scale" after "are spatially varying", or provide a similar modification.*

**Following the Editor suggestions, we made these modifications to the original manuscript.**

[revised manuscript text omitted]